# DoubleGen: Debiased Generative Modeling of Counterfactuals

## Abstract

Generative models for counterfactual outcomes face two key sources of bias. Confounding bias arises when approaches fail to account for systematic differences between those who receive the intervention and those who do not. Misspecification bias arises when methods attempt to address confounding through estimation of an auxiliary model, but specify it incorrectly. We introduce DoubleGen, a doubly robust framework that modifies generative modeling training objectives to mitigate these biases. The new objectives rely on two auxiliaries—a propensity and outcome model—and successfully address confounding bias even if only one of them is correct. We provide finite-sample guarantees for this robustness property. We further establish conditions under which DoubleGen achieves oracle optimality—matching the convergence rates standard approaches would enjoy if interventional data were available—and minimax rate optimality. We illustrate DoubleGen with three examples: diffusion models, flow matching, and autoregressive language models.

## 1 Introduction

Generative models have achieved remarkable success at imitating real-world data. This capability underlies recent advances in image generation software (Ramesh et al., 2021; Rombach et al., 2021) and large language models (Radford, 2018; Touvron et al., 2023). But sometimes, there is a need to go beyond the world as it is, and instead imitate counterfactual data—that which would have arisen had the world been intervened on in some way (Komanduri et al., 2023).

For example, suppose a policymaker wants to see how medical records would look if a new treatment were universally adopted. The available records come from a time when resources were limited, and so the treatment was given preferentially to sicker patients. These patients also tend to have worse outcomes on treatment than others. A generative model trained on only treated patients' records would internalize this confounding, suggesting overly pessimistic outcomes. What the policymaker wants instead is to generate records as they *would have appeared* had treatment been given to everyone.

Confounding can also arise in many other settings. More active internet users are more likely to be exposed to web campaigns (Chan et al., 2010). Students with greater family resources are more likely to get extracurricular tutoring (Zhang et al., 2023). Celebrities with certain attributes are more likely to smile—see Tab. 1 (Liu et al., 2015). In this work, we use this celebrity setting as an illustration, seeking to answer the question: what would celebrity photos look like if everyone smiled?

Table 1: Selected attributes possessed by a larger percentage of smiling ($n = 78,080$) than nonsmiling ($n = 84,690$) CelebA faces. When trained on only smiling faces, traditional generative models overrepresent these attributes, failing to reflect how the population would look if everyone smiled.

|            | Lipstick | Makeup | Female* | Earrings | No Beard | Blonde |
|------------|----------|--------|---------|----------|----------|--------|
| Smiling    | 56%      | 47%    | 65%     | 26%      | 88%      | 18%    |
| Not Smiling| 38%      | 30%    | 52%     | 12%      | 79%      | 12%    |
| **Overall**| **47%**  | **38%**| **58%** | **19%**  | **83%**  | **15%**|

\* *Perceived binary sex, as labeled by human annotators (Liu et al., 2015).*

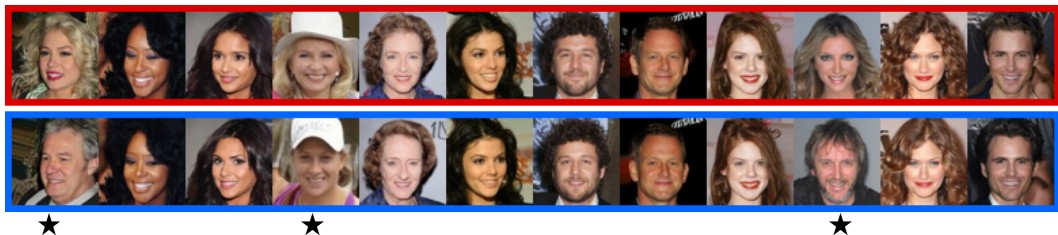

Figure 1: Counterfactual smiling celebrities generated by a **traditional diffusion model** trained on only smiling faces (top) and a **DoubleGen diffusion model** (bottom). Columns contain coupled samples, with the random seed set to the same value before generation. The ★'s mark the most qualitatively different pairs. See Fig. S2 for 200 random samples.

Formally, we aim to generate samples from the distribution $P^\star$ of a counterfactual outcome $Y^\star \in \mathcal{Y}$ under intervention $a^\star$. To accomplish this, we approximate a transport map $\phi^\star$ that takes as input simulated noise $U \sim \Pi$ and outputs $\phi^\star(U) \sim P^\star$—in other words, its pushforward satisfies $\phi^\star_\sharp \Pi = P^\star$. If independent samples from $P^\star$ were available, a standard generative modeling approach such as variational autoencoders (Kingma, 2013), diffusion models (Sohl-Dickstein et al., 2015), or flow matching (Lipman et al., 2022) could be used. However, such samples are not available: $P^\star$ represents the distribution of outcomes if everyone had received $a^\star$, but in reality not everyone did. Instead, a sample of independent copies of $Z = (X, A, Y) \sim P$ is available, and the intervention $A$ that someone received may depend on confounders $X$ of its effect on the outcome $Y$. Under a no-unmeasured confounders condition, $P^\star$ can be learned using data from $P$—see Sec. 3.

**Contributions.** We introduce DoubleGen, the first doubly robust framework for adapting standard generative modeling frameworks to generate counterfactuals (Sec. 3). We illustrate how DoubleGen can adapt three frameworks: flow matching, diffusion models, and autoregressive language models (Sec. 4). We establish high-probability statistical divergence guarantees (Secs. 5.1 and 5.2) and develop a method to evaluate minimax rate optimality, which we use to give conditions for DoubleGen diffusion models to be rate optimal (Sec. 5.3). Finally, we conduct experiments (Sec. 6 and Fig. 1).

## 2 RELATED WORK AND MOTIVATION

Previous works have provided ways to learn a counterfactual transport map $\phi^\star$ from observational data. These methods use one of three generation strategies: iterative, joint, or direct.

Iterative approaches generate data according to a known causal ordering. In our setting, they first generate features $X$, then intervene to set $A = a^\star$, and finally generate an outcome. These strategies have been proposed using generative adversarial networks (Kocaoglu et al., 2017), normalizing flows (Pawlowski et al., 2020), variational autoencoders (Karimi et al., 2020), and diffusion models (Chao et al., 2023). A disadvantage of these approaches is that they can suffer from error propagation, which is especially problematic if $X$ is high-dimensional (e.g., an image) (Javaloy et al., 2024).

Joint generation approaches avoid error propagation by simultaneously learning the distribution of $(X, Y^\star)$. This has been done with autoregressive flows (Khemakhem et al., 2021; Javaloy et al., 2024), variational graph autoencoders (Sanchez-Martin et al., 2021), and diffusion models (Sanchez & Tsaftaris, 2022). However, these methods, too, solve a harder problem than is necessary: generating $(X, Y^\star)$ even when only $Y^\star$ is of interest. This can worsen sample efficiency and computation time.

Direct approaches generate only the counterfactual outcome $Y^\star$. Wu et al. (2024) describes how to use inverse probability weighting with (approximate) likelihood approaches—such as classifier-free guided diffusion models (Ho & Salimans, 2022) and conditional variational autoencoders (Sohn et al., 2015). Our proposed approach—DoubleGen—is also a direct approach. It applies to any loss-based generative modeling strategy, of which likelihood approaches are a special case.

A limitation of all existing counterfactual generation approaches is that they are only singly robust. For iterative and joint generation approaches, this means they can only correctly generate samples of $Y^\star$ if they correctly model the distribution of $(X, Y^\star)$. For inverse probability weighted approaches (Robins, 1986), this means that they must correctly specify the propensity, $P\{A = a^\star \mid X = \cdot\}$

(Wu et al., 2024). When these auxiliary models are incorrect, the resulting misspecification bias can lead to incorrect counterfactual distributions. Doubly robust estimators are less prone to this bias by remaining valid if either a propensity or outcome model is correct (Robins et al., 1994). These estimators have been used previously to estimate counterfactual distributions (Fawkes et al., 2022; Kennedy et al., 2023; Martinez Taboada et al., 2023; Luedtke & Chung, 2024), but not to generate samples from them.

Another limitation of existing works is that, with the exception of (Wu et al., 2024), they only show how to adapt individual generative modeling approaches, and so cannot be immediately applied when new ones are developed. For instance, despite the recent success of flow matching, this approach has not yet been adapted to generate counterfactuals.

From a theoretical perspective, existing counterfactual generation methods are not well understood. This is in contrast to non-counterfactual settings, where generalization upper bounds and minimax lower bounds have been developed (Lee et al., 2022; De Bortoli, 2022; Oko et al., 2023; Chen et al., 2023; Lotfi et al., 2023; Fukumizu et al., 2024; Holk et al., 2024). This theoretical gap is significant: without performance guarantees, practitioners cannot know when these methods will succeed or fail.

## 3 PROPOSED APPROACH

Our proposed approach leverages observational data from $P$ to learn to generate samples from the counterfactual distribution $P^\star$. It does so under a no-unmeasured confounders assumption (Robins, 1986; Pearl, 2009), which makes it possible to identify $P^\star$ with $P$ through the relation

$$P^\star\{Y^\star \in \mathcal{Y}'\} = \int P\{Y \in \mathcal{Y}' \mid A = a^\star, X = x\}\, P_X(dx), \quad \text{for all measurable sets } \mathcal{Y}' \subseteq \mathcal{Y}, \quad (1)$$

with $P_X$ the marginal distribution of $X$ under $P$. Formally, (1) holds if $Z$ has a corresponding counterfactual outcome $Y^\star$ and the following hold: no unmeasured confounders ($Y^\star \perp\!\!\!\perp A \mid X$), consistency ($Y = Y^\star$ whenever $A = a^\star$), and positivity ($P\{A = a^\star \mid X\} > 0$ $P_X$-a.s.).

Our method builds upon existing generative modeling strategies—those that practitioners would use in idealized scenarios where draws from $P^\star$ are available. We restrict our focus to loss-based strategies expressible as in Alg. 1 (OracleGen), which impose that the final transport map $\phi_n^\star$ is defined as a transformation of a hypothesis $\theta_n^\star$ selected by optimizing an empirical risk $R_n^\star$. This hypothesis may, for example, be a global empirical risk minimizer or a neural network whose weights were learned by stochastic gradient descent. As illustrated in the next subsection, flow matching, diffusion models, and autoregressive language models can all be expressed as in OracleGen.

Alg. 2 displays our proposed approach, DoubleGen. In it, the oracle risk $R_n^\star$ is replaced by a risk $R_n$ that can be computed using data drawn from $P$. Evaluating this risk requires estimating two auxiliary nuisances: an inverse propensity $1/P(A = a^\star \mid X = \cdot)$ and a conditional transport map $\psi_P$ that returns draws of $Y \mid A = a^\star, X$. We learn these nuisances with 2-fold cross-fitting, so that every observation is used for estimation and for risk evaluation but never for both at the same time (Newey & Robins, 2018). This allows the nuisances to be constructed using any preferred approach, such as Riesz regression for the inverse propensity (Chernozhukov et al., 2021) or a diffusion model for the outcome model (Batzolis et al., 2021). The nuisance estimates are used to construct an augmented inverse probability weighted estimator of the risk (Robins et al., 1994). The resulting estimator, $R_n$, is doubly robust (Scharfstein et al., 1999), in the sense that, for each $\theta \in \Theta$, $R_n(\theta)$ is a consistent estimator of the counterfactual population risk $E_{P^*}[\ell(\theta, Y^\star)]$ if either nuisance is estimated consistently (van der Laan et al., 2003; Bang & Robins, 2005). Such double robustness has

---

**Algorithm 1** OracleGen: Oracle counterfactual generative modeling

⚠ *Cannot be used in practice — requires access to draws from the counterfactual distribution $P^\star$*

**Require:** • counterfactual data $Y_1^\star, Y_2^\star, \ldots, Y_n^\star \overset{\text{iid}}{\sim} P^\star$
          • choice of generative modeling framework, defined by a hypothesis space $\Theta$, loss $\ell :$
             $\Theta \times \mathcal{Y} \to \mathbb{R}$, and sampling map transformation $\tau : \Theta \to \mathcal{Y}^{\mathcal{U}}$
  1: **Risk minimization:** using any preferred approach, define $\theta_n^\star$ via $R_n^\star(\theta) := \frac{1}{n} \sum_{i=1}^n \ell(\theta, Y_i^\star)$
  2: **return** transport map $\phi_n^\star := \tau(\theta_n^\star)$

---

---

**Algorithm 2** DoubleGen: Doubly robust counterfactual generative modeling
☑ *Can be used in practice — only requires draws from the factual distribution $P$*

---

**Require:** ● data $Z_1, Z_2, \ldots, Z_n \overset{\text{iid}}{\sim} P$, partitioned into multisets $\mathcal{Z}_n^1, \mathcal{Z}_n^2$ of sizes $\lfloor n/2 \rfloor$ and $\lceil n/2 \rceil$
   ● choice of generative modeling framework, as in OracleGen

1: **Nuisance estimation:** for $j \in \{1, 2\}$, use observations in $\mathcal{Z}_n^j$ to obtain estimates $\psi_n^j$ and $\alpha_n^j$ of

 (i) **Outcome model:** a conditional transport map with $\psi_P(\,\cdot\,|\,x)_\sharp \Pi = P_{Y|A=a^\star, X=x}\, P_X$-a.s.

 (ii) **Inverse propensity:** $\alpha_P(x) := 1/P(A = a^\star \,|\, X = x)$

2: **Risk minimization:** using any preferred approach, define $\theta_n$ based on

$$R_n(\theta) := \frac{1}{n} \sum_{j=1}^{2} \sum_{z \in \mathcal{Z}_n^{3-j}} \int \big[ 1(a = a^\star) \alpha_n^j(x) \big\{ \ell(\theta, y) - \ell(\theta, \psi_n^j(u|x)) \big\} + \ell(\theta, \psi_n^j(u|x)) \big] \Pi(du)$$

 ▷ *unbiased gradients can be obtained in the usual way, via random sampling of $(j, z, u)$*
3: **return** $\phi_n := \tau(\theta_n)$

---

previously been leveraged when deriving causal machine learning guarantees (e.g., Rotnitzky et al., 2006; Laan et al., 2006; Rubin & van der Laan, 2007; Luedtke et al., 2017; Rotnitzky et al., 2017; Nie & Wager, 2021; Kennedy, 2023; Morzywolek et al., 2023; Foster & Syrgkanis, 2023; van der Laan et al., 2024).

# 4 EXAMPLES OF FRAMEWORKS EXPRESSIBLE AS IN ORACLEGEN

**Example 1** (Flow matching). Flow matching estimates a transport map $\phi^\star$ that solves an ordinary differential equation (Lipman et al., 2022). Taking $\mathcal{U} = \mathcal{Y} = \mathbb{R}^d$, a simple version of this approach takes $\phi^\star(u) := y_1$, with the value of $y_1$ implied by the differential equation on $[0, 1]$ satisfying initial condition $y_0 = u$ with $y_t' := dy_t/dt$ satisfying

$$y_t' = E\left[Y^\star - U \mid (1 - t)U + tY^\star = y_t\right],$$

where $(U, Y^\star) \sim \Pi \times P^\star$ with $\Pi = N(0_d, I_d)$ (Liu et al., 2022). This can be estimated as in OracleGen by letting $\Theta$ denote an appropriately restricted (e.g., bounded, smooth) set of $\mathcal{Y} \times [0, 1] \to \mathcal{Y}$ functions, taking $\ell(\theta, y) = \int_0^1 E_\Pi[\|y - U - \theta([1-t]U + ty, t)\|^2] dt$, and letting $\tau(\theta)(u)$ denote the value of $y_1$ implied by the differential equation with initial condition $y_0 = u$ and $dy_t/dt = \theta(y_t, t)$.

**Example 2** (Diffusion model). Diffusion models seek to reverse an iterative process that gradually converts data from $P^\star$ into Gaussian noise (Song et al., 2020). The noising process is described via a stochastic differential equation (SDE) on $\mathbb{R}^d \supseteq \mathcal{Y}$. A common framing takes $Y_0 \sim P^\star$ and then, for $W$ a $d$-dimensional Wiener process on $[0, \bar{t}]$, evolves as $dY_t = -\beta_t Y_t\, dt + \sqrt{2\beta_t}\, dW_t$, with $\bar{t} < \infty$ a truncation time at which the noising process is terminated and $t \mapsto \beta_t$ a smooth map from $\mathbb{R}$ into $[\underline{\beta}, \overline{\beta}] \subset (0, \infty)$. A strong solution exists provided $E_{P^\star}[\|Y_0\|^2] < \infty$ (Oksendal, 2013, Thm. 5.2.1).

An approximate transport map $\phi^\star$ from $\text{Law}(Y_{\bar{t}}) \times \text{Law}(\widetilde{W})$ to $P^\star$ can be obtained by reversing this SDE (Anderson, 1982), where $\widetilde{W}$ denotes a $d$-dimensional Wiener process with time flowing backward from $\bar{t}$ to $\underline{t}$. The truncation time $\underline{t}$ can be chosen to be slightly larger than $0$ to avoid instability (Oko et al., 2023). To generate a sample using $\phi^\star$, first independently draw $\widetilde{Y}_{\bar{t}} \sim \text{Law}(Y_{\bar{t}})$ and $\widetilde{W}$, and then let $\phi^\star(\widetilde{Y}_{\bar{t}}, \widetilde{W})$ be a strong solution $\widetilde{Y}_{\underline{t}}$ to the reverse-time SDE

$$d\widetilde{Y}_t = -\beta_t\big[\widetilde{Y}_t + 2\theta_{P^\star}(\tilde{Y}_t, t)\big] dt + \sqrt{2\beta_t}\, d\widetilde{W}_t, \tag{2}$$

where $\theta_{P^\star}(y, t) := \nabla_y \log \mathrm{p}_t(y)$ is the score of $\mathrm{p}_t$, the density of $Y_t$. This score rewrites as $\theta_{P^\star}(y, t) = \{\mu_t E[Y_0 \,|\, Y_t = y] - y\}/\sigma_t^2$ with $\mu_t := \exp(-\int_0^t \beta_v\, dv)$ and $\sigma_t^2 := 1 - \exp(-2\int_0^t \beta_v\, dv)$, and belongs to the set $\Theta$ of maps from $\mathbb{R}^d \times [\underline{t}, \bar{t}]$ to $\mathbb{R}^d$.

The approximate transport map $\phi^\star$ cannot be used to generate samples in practice, because using it relies on knowing two $P^\star$-dependent quantities: $\text{Law}(Y_{\bar{t}})$ and $\theta_{P^\star}$. Diffusion modeling replaces both by approximations. For the first, it uses that $\text{Law}(Y_{\bar{t}})$ is approximately Gaussian

Table 2: Examples of generative modeling paradigms that can be expressed as in OracleGen.

|  | Outcome type ($Y^\star$) | Hypothesis ($\theta_{P^\star}$) | Loss ($\ell$) | Sampler ($\tau$) |
|---|---|---|---|---|
| Flow matching | $\mathbb{R}^d$ (e.g., image) | vector field | velocity matching | ODE solver |
| Diffusion model | $\mathbb{R}^d$ (e.g., image) | score | denoising score matching | SDE solver |
| Autoreg. model | $[k]^d$ (token seq.) | next-token prob. | cross-entropy | ancestral |

noise, $N(0_d, I_d)$, provided the hyperparameter $\beta$ is chosen so $(\mu_{\bar{t}}, \sigma_{\bar{t}}^2) \approx (0, 1)$; this holds since $Y_t \mid Y_0 \sim N(\mu_t Y_0, \sigma_t^2 I_d)$. Second, it uses that $\theta_{P^\star}$ can be estimated using the denoising score matching loss (Vincent, 2011)

$$\ell(\theta, y) = \int_{\underline{t}}^{\bar{t}} E\big[\big\|(\mu_t Y_0 - Y_t)/\sigma_t^2 - \theta(Y_t, t)\big\|^2 \mid Y_0 = y\big] dt.$$

The final map $\tau(\theta)$ is defined a.s. over $(\widetilde{Y}_{\bar{t}}, \widetilde{W}) \sim N(0_d, I_d) \times \mathrm{Law}(\widetilde{W}) =: \Pi$ so that $\tau(\theta)(\widetilde{Y}_{\bar{t}}, \widetilde{W})$ is the value of $\widetilde{Y}_{\underline{t}}$ under a strong solution of an SDE that evolves in as (2), but with $\theta_{P^\star}$ replaced by $\theta$.

**Example 3** (Unsupervised pretraining of autoregressive language model)**. Autoregressive language models generate sequences $Y(1), Y(2), \ldots, Y(d)$ whose elements belong to a token dictionary $[k]$ (Graves, 2013; Radford et al., 2019; Touvron et al., 2023). Token $k$ marks the end of content and token 1 is used for padding thereafter, meaning $Y(j) = k$ implies $Y(i) = 1$ for all $i > j$; removing these tokens in post-processing allows generation of variable-length outputs.

Samples from a token-sequence distribution $P^\star$ can be obtained via ancestral sampling, which recursively draws the next token from a categorical distribution conditional on previous ones (Bishop & Nasrabadi, 2006). Expressing this procedure via inverse transform sampling yields a transport map $\phi^\star(\cdot) = (\phi_j^\star(\cdot))_{j=1}^d$ from $\Pi = \mathrm{Unif}[0, 1]^d$ to $P^\star$. This map is defined recursively through the conditional quantile function of $Y^\star(j)$ as $\phi_j^\star(u) = Q_{P^\star, j}(u_j \mid Y^\star(i) = \phi_i^\star(u) : i < j)$, $j = 1, 2, \ldots, d$.

The transport map $\phi^\star$ can be estimated as in OracleGen. To do this, $\Theta$ is taken to be the set of functions mapping from $[k]^{d-1}$ to the $(k-1)$-simplex. The cross-entropy loss

$$\ell(\theta, y) = -\sum_{j: y(j) \neq 1} \log \theta_{y(j)}(1, 1, \ldots, 1, y(1), y(2), \ldots, y(j-1))$$

is used (Graves, 2013), with the input to each $\theta_{y(j)}$ left-padded with 1s to make them $(d-1)$-dimensional. The risk minimization step of OracleGen may, for example, train a transformer model that masks the padded tokens (Vaswani, 2017). The inverse transform sampling map $\tau(\theta)(\cdot) = (\tau_j(\theta)(\cdot))_{j=1}^d$ is defined recursively as $\tau_j(\theta)(u) = Q_{\theta, j}(u_j \mid \tau_i(\theta)(u) : i < j)$, with the conditional quantile function $Q_{\theta, j}$ defined as the left-continuous generalized inverse of the distribution function

$$F_{\theta, j}(\cdot \mid \tau_i(\theta)(u) : i < j) = \sum_{m=1}^k 1\{m \leq \cdot\}\, \theta_m\big(1, 1, \ldots, 1, \tau_1(\theta)(u), \tau_2(\theta)(u), \ldots, \tau_{j-1}(\theta)(u)\big).$$

The transport map $\phi^\star$ defined earlier equals $\tau(\theta_{P^\star})$ with $\theta_{P^\star} \in \arg\min_{\theta \in \Theta} E_{P^\star}[\ell(\theta, Y^\star)]$.

## 5 THEORETICAL GUARANTEES

### 5.1 OVERVIEW

Our guarantees for the transport map $\phi_n$ are measured in terms of a divergence $D$, defined as a nonnegative function of two distributions satisfying $D(P_1^\star, P_2^\star) = 0$ if and only if $P_1^\star = P_2^\star$. Giving these guarantees requires relating the divergence of a pushforward $\tau(\theta)_\sharp \Pi$ from $P^\star$ to the generalization error $\mathcal{G}_{P^\star}(\theta) := \inf_{\theta^\star \in \Theta} \int [\ell(\theta, y) - \ell(\theta^\star, y)] \, P^\star(dy)$ and ensuring that the output of DoubleGen makes this generalization error small with high probability.

**C1)** *Divergence dominated by generalization error:* There exist $b, C_1, \epsilon > 0$ such that

$$D(P^\star, \tau(\theta)_\sharp \Pi) \leq C_1 \mathcal{G}_{P^\star}(\theta)^b + \epsilon \quad \text{for all } \theta \in \Theta. \tag{3}$$

**C2)** *Generalization bound:* For $s, r > 0$, $\mathcal{G}_{P^\star}(\theta_n) \leq r$ with probability (w.p.) at least $1 - e^{-s}$.

**Proposition 1** (Divergence bound)**.** *Under C1 and C2, $D(P^\star, \phi_{n\sharp}\Pi) \leq C_1 r^b + \epsilon$ w.p. at least $1 - e^{-s}$.*

The proof is immediate. This result is important because it yields exactly the sort of guarantee desired for a generative modeling algorithm, provided its conditions are satisfied. The first has already been established for all our examples (Appx. A). We provide a means to establish the second in Sec. 5.2. A visual overview of how this result can be used to provide divergence guarantees in our three examples is given in Fig. S1 in the appendix.

The minimax optimality of $\phi_n$ follows under conditions, as we show in Sec. 5.3. The main insight is simple: because DoubleGen must learn from factual data alone while OracleGen has access to counterfactuals, the oracle problem is no harder. We formalize this in Thm. 2 by showing that minimax lower bounds for OracleGen also apply to DoubleGen. Hence, optimality can be assessed by importing existing lower bounds from the non-counterfactual generative modeling literature.

## 5.2 GENERALIZATION BOUNDS FOR DOUBLEGEN

We now give a generalization bound for DoubleGen when implemented via an empirical risk minimizer over $\underline{\Theta} \subseteq \Theta$. While we focus on empirical risk minimizers here, generalization bounds can also be derived for other frameworks—see Appx. H. Our guarantee will rely on several conditions.

**C3)** *Existence of risk minimizer:* there exists $\theta_{P^\star} \in \arg\min_{\theta \in \Theta} \int \ell(\theta, y) \, P^\star(dy)$.

The remaining conditions are assumed to hold for a common choice of risk minimizer $\theta_{P^\star}$. We also define the centered losses $\ell_{P^\star}(\theta)(y) := \ell(\theta, y) - \ell(\theta_{P^\star}, y)$ and loss class $\ell_{P^\star}(\underline{\Theta}) := \{\ell_{P^\star}(\theta) : \theta \in \underline{\Theta}\}$.

**C4)** *Bounded loss:* there exists $C_4 < \infty$ such that $\sup_{\theta \in \underline{\Theta}} \|\ell_{P^\star}(\theta)\|_{L^\infty(P^\star)} \leq C_4$.

**C5)** *Curvature of risk:* there exists $C_5 < \infty$ such that, for all $\theta \in \underline{\Theta}$, $\|\ell_{P^\star}(\theta)\|_{L^2(P^\star)}^2 \leq C_5 \mathcal{G}_{P^\star}(\theta)$.

**C6)** *Uniform entropy integral bound (see Appx. B):* $J(\delta, \ell_{P^\star}(\underline{\Theta})) < \infty$ for some $\delta > 0$.

The above conditions yield a generalization bound for OracleGen (Appx. F). Similar conditions are commonly used to derive rate guarantees in statistical learning problems (Bartlett & Mendelson, 2006).

The remaining conditions account for DoubleGen's reliance on nuisances. The first imposes smoothness of the loss. For collections of conditional transport maps $\Psi := \mathcal{Y}^{\mathcal{U} \times \mathcal{X}}$ and $\Psi_P := \{\psi : \psi(\cdot \,|\, x)_\sharp \Pi = P_{Y|A=a^\star, X=x} \; P_X\text{-a.s.}\}$, this condition involves a function $d_\Psi(\cdot, \Psi_P) : \Psi \to [0, \infty)$. Our bound will be tightest when $d_\Psi(\psi_n^j, \Psi_P)$ is small. The measure $d_\Psi$ need not arise from a metric, but it must satisfy $d_\Psi(\psi_P, \Psi_P) = 0$ for all $\psi_P \in \Psi_P$—see Lem. S16 for one possible choice of $d_\Psi$.

**C7)** *Mixed-Lipschitz loss:* there exists $C_7 < \infty$ such that, for all $\theta \in \underline{\Theta}$, $\psi \in \Psi$, and $\psi_P \in \Psi_P$,

$$\int \left\{ \int [\ell_{P^\star}(\theta)(\psi(u|x)) - \ell_{P^\star}(\theta)(\psi_P(u|x))] \, \Pi(du) \right\}^2 P_X(dx) \leq C_7 \mathcal{G}_{P^\star}(\theta) \, d_\Psi^2(\psi, \Psi_P).$$

**C8)** *Strong positivity:* there exists $C_8 > 0$ and a version of $\alpha_P$ such that $\alpha_P \in [1, C_8]^{\mathcal{X}}$. Moreover, the estimates of $\alpha_P$ respect this bound, in that $\alpha_n^j \in [1, C_8]^{\mathcal{X}}$ for all $j \in [2]$.

**Theorem 1** (DoubleGen generalization bound, informal statement)**.** *Suppose there exists $\theta_n \in \arg\min_{\theta \in \underline{\Theta}} R_n(\theta)$ and C3–C8. If $s > 0$ and $\delta_n$ satisfies $J(\delta_n, \ell_{P^\star}(\underline{\Theta})) \leq n^{1/2}\delta_n^2$, then*

$$\mathcal{G}_{P^\star}(\theta_n) \lesssim \inf_{\theta \in \underline{\Theta}} \mathcal{G}_{P^\star}(\theta) + \delta_n^2 + s/n + \max_{j \in [2]} \|\alpha_n^j - \alpha_P\|_{L^2(P_X)}^2 d_\Psi^2(\psi_n^j, \Psi_P) \qquad (4)$$

*w.p. at least $1 - e^{-s}$. The final term is **doubly robust**, vanishing if $\alpha_n^j = \alpha_P$ or $\psi_n^j \in \Psi_P$ for $j \in [2]$.*

Above, '$\lesssim$' hides a multiplicative constant that does not depend on $n$. The bound reveals a tradeoff: increasing the size of $\underline{\Theta}$ will decrease the approximation error, $\inf_{\underline{\theta} \in \underline{\Theta}} \mathcal{G}_{P^\star}(\underline{\theta})$, but increase the complexity term. This informal statement captures these main dependencies and is valid when

$\max_j d_\Psi^2(\psi_n^j, \Psi_P)$ is almost surely bounded. See Thm. S1 in Appx. D for the formal statement, which gives explicit constants and also holds when $\max_j d_\Psi^2(\psi_n^j, \Psi_P)$ is unbounded.

Appendix C interprets and discusses the above generalization bound: the complexity term $\delta_n^2$, conditions under which the doubly robust term will be negligible, a localized version of Thm. 1 that can be used to establish oracle-optimal rates of convergence, and connections to existing approaches.

## 5.3 Minimax optimality

**Upper bound on worst-case divergence of an empirical risk minimizer.** When paired with Prop. 1, Thm. 1 provides a means to provide divergence guarantees for the transport map $\phi_n = \tau(\theta_n)$. In particular, if C1 and the conditions of Thm. 1 hold, the doubly robust term is no more than $\delta_n^2$ with sufficient probability, and $\epsilon \leq \delta_n^2$, then Prop. 1 shows that $D(P^\star, \phi_{n\sharp}\Pi) \lesssim [\inf_{\theta \in \Theta} \mathcal{G}_{P^\star}(\theta) + \delta_n^2 + s/n]^b$ w.p. at least $1 - e^{-s}$, where '$\lesssim$' denotes inequality up to constants that only depend on the $C_j$ constants indexing the conditions for Prop. 1 and Thm. 1. If $\delta_n = \Omega(\log(n)/\sqrt{n})$—as is typical for nonparametric hypothesis classes $\underline{\Theta}$—then Thm. 1 in Mey (2020) yields the in-expectation bound $E_{P^n}[D(P^\star, \phi_{n\sharp}\Pi)] \lesssim [\inf_{\theta \in \underline{\Theta}} \mathcal{G}_{P^\star}(\theta) + \delta_n^2]^b$. This upper bound only depends on $P^\star$ and $P$ through the constants indexing the conditions for Prop. 1 and Thm. 1. Hence, it holds uniformly over all $(P^\star, P)$ in any collection $\mathcal{M}$ over which these constants are uniformly bounded and $P^\star$ is identified through (1). This in turn provides a way to upper bound DoubleGen's worst-case expected divergence over $\mathcal{M}$.

Interesting models $\mathcal{M}$ arise by defining a model $\mathcal{P}^\star$ for the counterfactual distribution $P^\star$ and then, for each $P^\star$, letting $\mathcal{P}(P^\star)$ denote a collection of $P$ satisfying (1). Both $\mathcal{P}^\star$ and $\mathcal{P}(P^\star)$ may be subject to local or global smoothness constraints so that the relevant constants from C1–C8 are uniformly bounded over $(P^\star, P)$ in $\mathcal{M} := \{(P^\star, P) : P^\star \in \mathcal{P}^\star, P \in \mathcal{P}(P^\star)\}$. We then have the following bound on the worst-case performance of the transport map $\phi_n$ in terms of expected divergence:

$$\sup_{P^\star \in \mathcal{P}^\star} \sup_{P \in \mathcal{P}(P^\star)} E_{P^n}[D(P^\star, \phi_{n\sharp}\Pi)] \lesssim \left[\inf_{\theta \in \underline{\Theta}} \mathcal{G}_{P^\star}(\theta) + \delta_n^2\right]^b. \tag{5}$$

**Minimax lower bound for any generative modeling algorithm.** We now show that any minimax lower bound for OracleGen is also a lower bound for DoubleGen. This provides a simple path for establishing the minimax optimality of $\phi_n$: show that an existing lower bound from the non-causal generative modeling literature matches the rate of decay of (5) in $n$.

In the following result, $\mathcal{T}^\star$ and $\mathcal{T}$ denote unrestricted sets of oracle and non-oracle generative modeling procedures returning transport maps in $\mathcal{Y}^\mathcal{U}$. Procedures in $\mathcal{T}^\star$ take as input counterfactuals $Y_{[n]}^\star := \{Y_i^\star\}_{i \in [n]}$ drawn from an $n$-fold product distribution $P^{\star n} := P^\star \times \ldots \times P^\star$, while those in $\mathcal{T}$ take factual data $Z_{[n]} := \{Z_i\}_{i \in [n]}$ drawn from $P^n := P \times \ldots \times P$. They also take in exogenous noise $V \sim \nu := \mathrm{Unif}[0,1]$; this allows, for example, the procedures to train a neural network using stochastic gradient descent or use a stopping criterion based on a training-validation split.

**Theorem 2.** *Let $\mathbb{E}$ and $E$ being expectations under sampling from $P^{\star n} \times \nu$ and $P^n \times \nu$. It holds that*

$$\inf_{T^\star \in \mathcal{T}^\star} \sup_{P^\star \in \mathcal{P}^\star} \mathbb{E}\Big[D\Big(P^\star, T^\star(Y_{[n]}^\star, V)_\sharp \Pi\Big)\Big]$$

$$\leq \inf_{T \in \mathcal{T}} \sup_{P^\star \in \mathcal{P}^\star} \sup_{P \in \mathcal{P}(P^\star)} E\Big[D\Big(P^\star, T(Z_{[n]}, V)_\sharp \Pi\Big)\Big],$$

*Hence, any lower bound on the oracle minimax risk also lower bounds the factual minimax risk.*

**Example 2 (cont.): rate optimality of DoubleGen diffusion modeling.** We build on a recent work that showed non-counterfactual diffusion modeling nearly achieves the minimax rate for estimating a Besov-smooth density with respect to the total variation distance (Oko et al., 2023). We show DoubleGen achieves the same guarantee provided the nuisances are estimated well enough.

We begin by deriving a minimax lower bound for our counterfactual generative modeling problem. When doing this, we let $\mathcal{P}^\star$ denote the set of distributions on $[-1,1]^d$ whose densities belong to a fixed-radius ball in the Besov space $B_{p,q}^s([-1,1]^d)$, with $p, q \in [1, \infty)$ and $s > 0 \vee d(1/p - 1/2)$

(van der Vaart & Wellner, 2023, Sec. 2.7.2). Proposition D.4 of Oko et al. (2023) gives a minimax lower bound for any deterministic estimator of a distribution in $\mathcal{P}^\star$, and the convexity of the total variation distance (TV) implies that any randomized estimator is dominated by a deterministic one. Hence, for a constant $C$ not depending on $n$, the oracle minimax risk with $D = \mathrm{TV}$ lower bounds as

$$\inf_{T^\star \in \mathcal{T}^\star} \sup_{P^\star \in \mathcal{P}^\star} E_{P^{\star n} \times \nu} \left[ \mathrm{TV}\left(P^\star, T^\star(Y^\star_{[n]}, V)_\sharp \Pi\right) \right] \geq C\, n^{-\frac{s}{2s+d}}.$$

By Thm. 2, the same **minimax lower bound** is valid for our counterfactual generation problem.

The instance of DoubleGen diffusion we study is nearly the same as the one used in Oko et al. (2023), differing only in the choice of loss function and the need to estimate the nuisances used to define it. An empirical risk minimizer is run over a deep, sparse neural network class $\underline{\Theta}$ that grows with $n$. This class is rich enough to approximate the true score well (Lem. S17), while also being small enough so that the entropy of $\ell_{P^\star}(\underline{\Theta})$ is not too large (Lem. S20).

The conditions of our generalization bound are satisfied when $\underline{\Theta}$ is specified in this way (Appx. J.2). Combining this with Prop. 1 gives a TV bound, which we state below. In this result, '$\lesssim$' denotes an inequality up to a multiplicative constant that may depend on the constants from the conditions but may not depend on $n$ or the particular instance of $(P^\star, P)$.

**Theorem 3** (TV bound for DoubleGen diffusion). *Suppose $\phi_n := \tau(\theta_n)$ for $\theta_n \in \mathrm{argmin}_{\theta \in \underline{\Theta}} R_n(\theta)$, with $\underline{\Theta}$ the neural network class from Lem. S17. Let $d_\Psi$ be as in Lem. S16 and suppose $\max_j d_\Psi(\psi_n^j, \Psi_P)$ is a.s. bounded uniformly in $n$. If C8 and C9–C13, $r > 0$, and $n$ is large enough, then, w.p. at least $1 - e^{-r}$,*

$$\mathrm{TV}(P^\star, \phi_{n\sharp}\Pi) \lesssim \log^{\frac{17}{2}}(n)\, n^{\frac{-s}{2s+d}} + \sqrt{r/n} + \max_{j \in [2]} \|\alpha_n^j - \alpha_P\|_{L^2(P_X)} d_\Psi(\psi_n^j, \Psi_P).$$

If the first term on the right dominates with high probability uniformly over $P$, then this shows that DoubleGen diffusion achieves the **minimax rate** over a Besov smoothness class, up to polylogarithmic factors. This is the same guarantee as Oko et al. (2023) established for traditional diffusion models.

Both our analysis and that of Oko et al. (2023) invoke two additional regularity assumptions beyond Besov smoothness—boundedness away from zero and smoothness on the boundary (C11 and C12). Ours further assumes that the nuisances can be estimated at a suitable rate. Because the available minimax lower bound is stated for the larger Besov class, a matching lower bound for the slightly more restrictive class considered here (and in Oko et al.) is still needed to claim full minimax optimality. Closing this gap is an interesting direction for future work.

In Appxs. K and L, we similarly combine Prop. 1 and Thm. S1 to establish divergence bounds for our other two examples. In future work, it would be interesting to show they are are rate optimal.

## 6 NUMERICAL EXPERIMENTS

### 6.1 GENERATING COUNTERFACTUAL FACES

We evaluated DoubleGen's performance on generating synthetic celebrity faces, under the intervention that they are all smiling. We trained diffusion models using 162,770 images from CelebA (Liu et al., 2015), withholding the other 39,829 for evaluation. Each image $Y$ is accompanied by a binary smiling indicator $A$ and 31 attributes that serve as potential baseline confounders. Tab. 1 presents the percentages of instances in the training set with some of these attributes.

We compare to three baselines. The first ignores potential confounding by fitting a traditional diffusion model with only smiling instances; Tab. 1 suggests this is inadvisable, and so we call it the 'naïve' approach. The second uses a marginal structural generative model (MSGM) (Wu et al., 2024), equivalent to Alg. 2 with an inverse inverse probability weighted (IPW) loss—that is, with $\ell(\theta, \psi_n^j)$ replaced by 0. The third uses plug-in estimation, equivalent to Alg. 2 with $\alpha_n^j$ replaced by 0.

To ensure fair comparison, the same nuisance estimates were supplied to all methods. We estimated each nuisance twice: once in a well-specified setting using all available training data, and again in a misspecified setting where nuisance models were trained to overrepresent dark-haired instances. Further details on our experiment can be found in Appx. M.1.1.

Table 3: Performance under different misspecification settings. $\downarrow$ = lower better, $\uparrow$ = higher better.

| | | Diffusion model (Sec. 6.1)[1] | | | | Language model (Sec. 6.2)[2] | | | |
| | | FAD$\downarrow$ | KAD$\downarrow$ | Prec$\uparrow$ | Rec$\uparrow$ | PPL$\downarrow$ | Mve$\uparrow$ | FI$\downarrow$ | $W_1\downarrow$ |
|---|---|---|---|---|---|---|---|---|---|
| | Naïve | 1.00 | 1.00 | 0.38 | 0.65 | 1.99 | 0.72 | 1.22 | 0.36 |
| *Both right* | Plug-in | 0.87 | 0.68 | 0.34 | 0.76 | 1.98 | 0.81 | 0.91 | 0.17 |
| | MSGM | 0.88 | 0.71 | 0.36 | 0.73 | 1.98 | 0.82 | 0.90 | 0.05 |
| | DoubleGen | **0.86** | **0.68** | 0.35 | 0.74 | **1.98** | **0.83** | **0.87** | 0.17 |
| *Outcome wrong* | Plug-in | 1.90 | 2.17 | 0.34 | 0.34 | 1.98 | 0.83 | 0.87 | 0.17 |
| | DoubleGen | **0.86** | **0.68** | **0.35** | **0.73** | **1.98** | 0.82 | 0.88 | **0.17** |
| *Propensity wrong* | MSGM | 0.93 | 0.71 | 0.34 | 0.74 | 1.98 | 0.80 | 0.96 | 0.17 |
| | DoubleGen | **0.85** | **0.56** | 0.32 | **0.79** | **1.98** | **0.82** | **0.89** | **0.12** |
| *Both wrong* | DoubleGen | 1.01 | 0.79 | 0.32 | 0.77 | 1.99 | 0.82 | 0.88 | 0.17 |

Blue: DoubleGen better than Naïve. Bold: **at least as good as best baseline** (in misspec. category).

[1] FAD/KAD=Fréchet/kernel ArcFace distance (rescaled so Naïve $= 1$), Prec=precision, Rec=recall. Details on these metrics can be found in Appx. M.1.1.

[2] PPL=perplexity$\div 10$, Mve=MAUVE, FI=frontier int$\div 10$, $W_1$=rating distribution Wass. error$\times 10$. Details on these metrics can be found in Appx. M.2.1.

The left half of Tab. 3 displays the results. When both nuisances were well-specified, DoubleGen achieved better Fréchet and kernel distances than the naïve approach. Its precision was slightly lower, suggesting the naïvely-generated faces more often resembled real faces. However, its recall was considerably higher, suggesting counterfactual smiling faces are more likely to be represented among DoubleGen's samples. Overall, DoubleGen performed comparably to plug-in estimation and MSGM when both nuisances were well-specified. When nuisances were misspecified, DoubleGen was more robust. Tab. S3 in the appendix shows similar results with respect to other metrics.

Tab. S4 in the appendix shows estimates of the attribute distributions of the synthetic smiling faces generated by each method. As anticipated in Tab. 1, the naïve model yields distributions similar to those of the population of factual smiling faces, whereas DoubleGen yields distributions more similar to those of the overall population.

## 6.2 GENERATING COUNTERFACTUAL PRODUCT REVIEWS

We next conducted a semi-synthetic experiment using the Amazon Reviews 2023 dataset (Hou et al., 2024), which consists of roughly 570 million reviews about 48 million products. We used real baseline features and review texts and a synthetic intervention sampled from a known propensity $\pi$. This semi-synthetic setup provides two key advantages: it gives us access to ground truth for evaluation, and it ensures that the identifiability condition in (1) is satisfied by construction.

We used low-rank adaptation (LoRA) (Hu et al., 2022) to finetune Llama-3.2-1B (Dubey et al., 2024). We compare performance using the DoubleGen loss to the same three baselines as in the previous experiment, under the same set of misspecification scenarios. Further details are in Appx. M.2.1.

The right half of Tab. 3 displays the results. When at least one nuisance was well-specified, DoubleGen outperformed the naïve approach across all metrics. DoubleGen also performed similarly to or better than other methods when both nuisances were correct. Under propensity misspecification, DoubleGen outperformed MSGM . Under outcome model misspecification, both DoubleGen and plug-in estimation maintained similar performance to what they achieved under correct specification, suggesting outcome model misspecification was mild. Fig. S3 in the appendix provides examples of reviews generated by DoubleGen, while Fig. S4 provides generated samples illustrating how the naïve model severely underrepresents book-related content, whereas DoubleGen does not.

## 7 DISCUSSION OF IMPLICATIONS AND EXTENSIONS

**Scope of empirical study.** Our experiments were designed to evaluate the applicability of our theoretical guarantees to realistic data-generating processes and sample sizes, not to provide an exhaustive benchmark. Given that training a single diffusion model in our setup required roughly 300 NVIDIA A6000 GPU hours, we leave systematically exploring performance on additional datasets and random seeds to future work.

**Equivalence between causal and missing data problems.** Generative modeling problems with outcomes missing at random can be tackled using DoubleGen. To do this, $A$ can be taken to indicate whether the outcome is observed ($A = a^\star$) or missing (Ding & Li, 2018). The cross-fitted augmented inverse probability weighted risk estimator DoubleGen uses allows predictions of missing outcomes to be obtained from any algorithm, including a pretrained foundation model (van der Laan et al., 2011; Chernozhukov et al., 2018).

**Minimax optimality when nuisances are hard to estimate.** We showed DoubleGen diffusion achieves the optimal rate under conditions, including that the nuisances are estimated well enough. When they cannot be estimated well—for example, because they are nonsmooth—the order $n^{-s/(2s+d)}$ minimax lower bound we gave may be loose. It would be interesting to derive a sharper bound that reflects this difficulty, as was done by Kennedy et al. (2024) in a different problem.

**Reduced-entropy sampling for language models.** Language models often generate qualitatively better text when they oversample high-probability tokens (Holtzman et al., 2019)—e.g., via temperature scaling (Caccia et al., 2018), top-$k$ sampling (Fan et al., 2018), or nucleus sampling (Holtzman et al., 2019). Each of these approaches redefines $\tau$ to yield a lower-entropy sampling scheme than the one from Example 3 (Nadeem et al., 2020). While reducing entropy can improve sample quality, it also makes the resulting $\tau(\theta_{P^\star})$ a transport map to an entropy-reduced variant of $P^\star$, rather than $P^\star$ itself. Regardless, DoubleGen can be applied to estimate these transport maps by simply redefining $\tau$.

**Extensions to joint and conditional sampling.** DoubleGen can be naturally extended to generate counterfactuals jointly with or conditionally on a subvector $V$ of the features $X$. Joint generation is straightforward: run DoubleGen with the modified outcome $Y' = (V, Y)$. Conditional generation requires allowing the oracle loss $\ell$ to depend on $y' = (v, y)$, rather than just $y$. For example, some image restoration and text-to-image diffusion models use $\ell(\theta, y') = \int 1\{t \in [\underline{t}, \overline{t}]\} E[\|(\mu_t Y_0 - Y_t)/\sigma_t^2 - \theta((v, Y_t), t)\|^2 \mid Y_0 = y] \, dt$ (Saharia et al., 2022; Rombach et al., 2022). With such losses, DoubleGen proceeds as in Alg. 2, but with $y$ and $\psi_n^j(u|x)$ on line 2 replaced by $y'$ and $(v, \psi_n^j(u|x))$. The analysis is nearly identical, yielding a generalization bound like Thm. 1.

## 8 ETHICS STATEMENT

Any system for generating synthetic data carries risks for misuse in generating deepfakes. While the focus of this work is on advancing methods and theory, future work should study responsible deployment practices to mitigate these risks. While DoubleGen is designed to address potential confounding between the intervention and outcome, it is not designed to address other sorts of bias, such as an underrepresentation of people from some groups. For example, CelebA reflects a particular western-focused dataset that may not adequately represent the full diversity of celebrity faces globally and certainly does not represent that of all human faces.

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

# APPENDICES

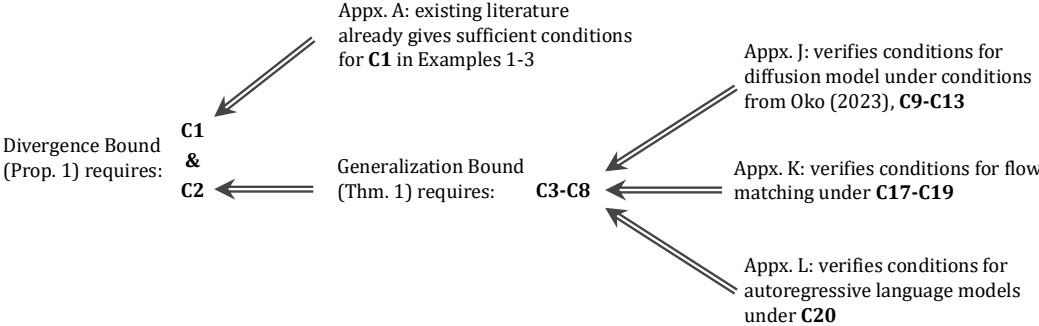

Figure S1: Summary of our main theoretical guarantees and the conditions they rely on. Moving from left to right, results go from high-level guarantees that can be applied broadly to more specialized results that verify the conditions of those to their left in specific settings.

*Not shown: A generalization bound for OracleGen that requires conditions similar to but slightly weaker than those of Thm. 1 (see Appx. F), a localized variant of Prop. S1 (Appx. E), and a minimax lower bound that can be used to establish that DoubleGen is minimax rate optimal (Thm. 2).*

# A   PLAUSIBILITY OF C1 IN OUR EXAMPLES

As summarized in Tab. S1, C1 holds in each of our examples under conditions, under regularity conditions that we give below.

**Example 1** (Flow matching, continued). When $D$ is the 2-Wasserstein distance ($W_2$), Thm. 1 in Benton et al. (2023) gives conditions under which (3) holds with $b = 1/2$, $\epsilon = 0$, and $C_1$ a constant depending on the smoothness of the vector field $\theta$. These conditions are satisfied if each vector field $\theta \in \Theta$ is sufficiently smooth and corresponds to a unique, smooth flow—see Benton et al. (2023).

**Example 2** (Diffusion model, continued). We follow the arguments used in Oko et al. (2023) to establish C1. These arguments rely on several regularity conditions on $P^\star$ and its density $\mathbb{p}$.

   **C9)** *Supported on hypercube:* $P^\star$ has support $\mathcal{Y} = [-1, 1]^d$.

   **C10)** *Smooth density:* $\|\mathbb{p}\|_{s|p,q} \leq C_{10}$, with $\|\cdot\|_{s|p,q}$ the usual norm on the Besov space $B_{p,q}^s(\mathcal{Y})$ (van der Vaart & Wellner, 2023, Sec. 2.7.2).

   **C11)** *Density bounded away from 0 and infinity:* $\mathbb{p}$ is bounded in $[1/C_{11}, C_{11}]$ on $\mathcal{Y}$.

   **C12)** *Density smooth at the boundary:* For some $\varepsilon > 0$, the restriction $\mathbb{p}_\varepsilon$ of $\mathbb{p}$ to $[-1, 1]^d \backslash [-1 + \varepsilon, 1 - \varepsilon]^d$ is infinitely differentiable with $\sum_{k=0}^\infty 2^{-k} \|D^k \mathbb{p}_\varepsilon\|_\infty / (1 + \|D^k \mathbb{p}_\varepsilon\|_\infty) \leq C_{12}$.

The above conditions are derived from Assumptions 2.4 and 2.6 of Oko et al. (2023). All our results will also hold if C12 is weakened so that $\varepsilon$ decays with $n$ at an appropriate rate—see Assumption 2.6 from that work for details. Following Oko et al. (2023), we also make the following requirement on the truncation times:

   **C13)** *Truncation times change appropriately with $n$:* $\bar{t} = s \log n / (\beta(2s + d))$ and $\underline{t} = n^{-\gamma}$ for any $\gamma > 0$ sufficiently large so that $\mathrm{TV}(P^\star, \mathrm{Law}(Y_{\underline{t}})) = O(n^{-s/(2s+d)})$.

By Thm. D.2 in Oko et al. (2023), such a $\gamma$ necessarily exists under C9–C12, and it can be chosen uniformly over all $P^\star$ satisfying C10–C12 for fixed values of the conditions' constants.

To ensure each reverse-time SDE we consider has a strong solution, we further require there to exist $L < \infty$ such that each $\theta \in \Theta$ is $L$-Lipschitz in its first argument. We will show C1 holds with $b = 1/2$, $C_1 = 1/\underline{\beta}$, and $\epsilon := O(n^{-s/(2s+d)})$, where showing $\epsilon$ is small requires C13.

We will write write $P^\star_{\theta, N(0,I)}$ as shorthand for $\tau(\theta)_\sharp \Pi$ and $P^\star_{\theta, Y_{\bar{t}}}$ to represent the law of the solution to the same reverse-time SDE as $\tau(\theta)$, but with the $N(0_d, I_d)$ distribution used in the initial condition at time $\bar{t}$ replaced by $\mathrm{Law}(Y_{\bar{t}})$. By the triangle inequality,

$$\mathrm{TV}(P^\star, P^\star_{\theta, N(0,I)}) \leq \mathrm{TV}(\mathrm{Law}(Y_{\underline{t}}), P^\star_{\theta, Y_{\bar{t}}}) + \mathrm{TV}(P^\star_{\theta, Y_{\bar{t}}}, P^\star_{\theta, N(0,I)}) + \mathrm{TV}(P^\star, \mathrm{Law}(Y_{\underline{t}})).$$

We bound the three terms on the right separately. For the first term, Pinsker's inequality and Girsanov's theorem [Oko et al., 2023, Prop. D.1; Karatzas & Shreve, 1991] together imply the following bound:

$$\mathrm{TV}(\mathrm{Law}(Y_{\underline{t}}), P^\star_{\theta, Y_{\bar{t}}}) \leq \left[\frac{1}{2} \mathrm{D_{KL}}(P^\star \parallel P^\star_{\theta, Y_{\bar{t}}})\right]^{1/2} \leq \left(\int_{\underline{t}}^{\bar{t}} \beta_t^{-2} E\left[\|\theta(Y_t, t) - \theta_{P^\star}(Y_t, t)\|^2\right] dt\right)^{1/2}.$$

The right-hand side is upper bounded by $\underline{\beta}^{-1} \mathcal{G}_{P^\star}(\theta)^{1/2}$. The second term is upper bounded by a constant $C$ depending only on $C_{11}$ times $e^{-\bar{t}\underline{\beta}}$ (Oko et al., 2023, Lem. D.3), which is $O(n^{-s/(2s+d)})$ by the choice of $\bar{t}$ in C13. The choice of $\underline{t}$ in C13 ensures the third term is $O(n^{-s/(2s+d)})$. Putting these three bounds together shows that $\mathrm{TV}(\mathrm{Law}(Y_{\underline{t}}), P^\star_{\theta, Y_{\bar{t}}}) \leq \underline{\beta}^{-1} \mathcal{G}_{P^\star}(\theta)^{1/2} + \epsilon$ for $\epsilon := \mathrm{TV}(P^\star, \mathrm{Law}(Y_{\underline{t}})) = O(n^{-s/(2s+d)})$ and an appropriately defined $C > 0$, and inspecting the second and third terms above shows that the additive $\epsilon$ term is needed on the right due to the truncation of the forward diffusion process at time $\bar{t} < \infty$ and reverse diffusion process at time $\underline{t} > 0$.

**Example 3** (Autoregressive model, continued). Let $\mathrm{D_{KL}}$ denote the Kullback-Leibler (KL) divergence. To ensure $\mathrm{D_{KL}}(P^\star \parallel \tau(\theta)_\sharp \Pi)$ is finite, suppose $P^\star$ is dominated by $\tau(\theta)_\sharp \Pi$ for each $\theta \in \Theta$. By the definition of the KL divergence (Cover, 1999, Eq. 2.26), $\mathrm{D_{KL}}(P^\star \parallel \tau(\theta)_\sharp \Pi) = \mathcal{G}_{P^\star}(\theta)$. Hence, C1 trivially holds with $D = \mathrm{D_{KL}}$, $b = C_1 = 1$, and $\epsilon = 0$.

Table S1: In our examples, C1 holds for the below choices of $(D, b, \epsilon)$.

| Framework | $D$ | $b$ | $\epsilon$ | Key results used in proof |
|---|---|---|---|---|
| Flow matching | 2-Wasserstein | 1/2 | 0 | Alekseev-Gröbner (Benton et al., 2023) |
| Diffusion model | Total variation | 1/2 | Trunc. error | Girsanov (Oko et al., 2023); Pinsker (Tsybakov, 2009) |
| Autoregressive language model | KL divergence | 1 | 0 | Definition of $D_{\mathrm{KL}}$ (Cover, 1999) |

# B  REVIEW: COVERING NUMBERS, ENTROPY INTEGRALS, AND A LOCAL MAXIMAL INEQUALITY

This appendix reviews concepts from empirical process theory: covering numbers, entropy integrals, and a local maximal inequality. We focus on collections of functions $\mathcal{F} \subseteq \mathbb{R}^{\mathcal{Z}}$ and study how well an empirical distribution $P_n$ approximates its population counterpart $P$. Specifically, we examine conditions under which $E\|P_n - P\|_{\mathcal{F}} := E \sup_{f \in \mathcal{F}} |(P_n - P)f|$ is small. While we present these concepts for functions and probability distributions on $\mathcal{Z}$, the definitions and results apply analogously to those defined on $\mathcal{Y}$, such as $\ell_{P^\star}(\theta)$ and $P^\star$.

For a class of functions $\mathcal{F} \subseteq \mathbb{R}^{\mathcal{Z}}$ and a probability measure $Q$ on $\mathcal{Z}$, the (external) covering number $N(\epsilon, \mathcal{F}, L^2(Q))$ is defined as the smallest cardinality of a $\mathcal{F}_\epsilon \subset L^2(Q)$ satisfying the following: for all $f \in \mathcal{F}$, there exists $g \in \mathcal{F}_\epsilon$ such that $\|f - g\|_{L^2(Q)} \leq \epsilon$. The uniform entropy integral is given by

$$J(\delta, \mathcal{F}) := \sup_Q \int_0^\delta \sqrt{1 + \log N(\epsilon, \mathcal{F}, L^2(Q))}\, d\epsilon, \tag{S1}$$

where the supremum is over all finitely supported measures $Q$ on $\mathcal{Z}$.

In the following lemma, the class $\mathcal{F} \subseteq \mathbb{R}^{\mathcal{Z}}$ is called 'suitably measurable' if $(z_i)_{i=1}^n \mapsto \sup_{f \in \mathcal{F}} \left| \sum_{i=1}^n e_i f^k(z_i) \right|$ is measurable for all $(e_i)_{i=1}^n \in \{-1, 1\}^n$ and $k \in \{1, 2\}$.

**Lemma S1** (Variant of Thm. 2.1 in Van Der Vaart & Wellner, 2011). *There exists a universal constant $K > 0$ such that, for all probability measures $P$ on some space $\mathcal{Z}$, suitably measurable $\mathcal{F} \subset [-b, b]^{\mathcal{Z}}$ bounded by some $b > 0$, and $\delta^2 \geq \sup_{f \in \mathcal{F}} \|f\|_{L^2(P)}^2$,*

$$E\|P_n - P\|_{\mathcal{F}} \leq Kn^{-1/2} J(\delta, \mathcal{F}) \left( 1 + \frac{bJ(\delta, \mathcal{F})}{\delta^2 n^{1/2}} \right).$$

We defer this and all proofs to the end of this appendix. The above can be used to compute a rate of convergence (van der Vaart & Wellner, 2023, Theorem 3.2.5) or finite-sample generalization bound (van der Vaart & Wellner, 2023, Lemma 3.5.9) for an empirical risk minimizer. This is the case, for example, in our Thms. S1 and S2. In special cases where $J(\delta, \mathcal{F}/b)$ is upper bounded by a constant times $\delta^{1-1/(2\rho)}$ for $\rho > 1/2$, such guarantees are determined by the solution to (S2) in the lemma below, for appropriate choices of constants $c, d$.

**Lemma S2** (Helpful result for determining complexity term in generalization bounds). *Fix $c, d \geq 0$ and $\rho > 1/2$. If $\delta \geq \left( 1 \vee [(c + d/4)^2/n]^{1/(4\rho+2)} \right) [(c + d/4)^2/n]^{\rho/(2\rho+1)}$, then*

$$c(\delta^{1-1/(2\rho)} \vee \delta) \left( 1 + \frac{d(\delta^{-1-1/(2\rho)} \vee \delta^{-1})}{n^{1/2}} \right) \leq n^{1/2} \delta^2. \tag{S2}$$

The following lemma is also useful when computing a critical radius $\delta_n$ such that, for all $\delta \geq \delta_n$, the right-hand side of Lem. S1 is no more than $\delta^2 n^{1/2}$. Specifically, this lemma shows that $\delta_n$ can be computed by just finding the $\delta$ such that the bound from Lem. S1 equals $\delta^2 n^{1/2}$.

**Lemma S3** (Monotonicity of bound from Lem. S1 when divided by $\delta^2$). *For any function class $\mathcal{F}$ and constants $c, d > 0$, the following function is strictly decreasing over $(0, \infty)$:*

$$\delta \mapsto [cJ(\delta, \mathcal{F})/\delta^2] \left[ 1 + dJ(\delta, \mathcal{F})/\delta^2 \right].$$

The following lemma lower bounds the complexity term in Thm. 1.

**Lemma S4** (Complexity term is $\Omega(n^{-1})$). *If $\delta_n$ satisfies $J(\delta_n, \ell_{P^\star}(\underline{\Theta})) \leq n^{1/2}\delta_n^2$, then $\delta_n^2 \geq n^{-1}$.*

We conclude with proofs.

*Proof of Lem. S1.* First suppose $\delta < b$. Since $\mathcal{F}/b \subset [-1,1]^{\mathcal{Z}}$ is suitably measurable and $\sup_{g \in \mathcal{F}/b} \|g\|_{L^2(P)}^2 \leq (\delta/b)^2 < 1$, Thm. 2.1 in Van Der Vaart & Wellner (2011) with constant envelope function 1 shows there exists a universal constant $K'$ such that

$$E\|P_n - P\|_{\mathcal{F}/b} \leq K' n^{-1/2} J(\delta/b, \mathcal{F}/b)\left(1 + \frac{b^2 J(\delta/b, \mathcal{F}/b)}{\delta^2 n^{1/2}}\right).$$

Now suppose $\delta \geq b$. In this case, Thm. 2.14.1 of van der Vaart & Wellner (2023) shows there exists a universal constant $K''$ such that $E\|P_n - P\|_{\mathcal{F}/b} \leq K'' n^{-1/2} J(1, \mathcal{F}/b)$. Combining these two cases, letting $K = K' \vee K''$, and using that $J(\cdot, \mathcal{F}/b)$ is monotone nondecreasing and nonnegative shows that

$$E\|P_n - P\|_{\mathcal{F}/b} \leq K n^{-1/2} J(\delta/b, \mathcal{F}/b)\left(1 + \frac{b^2 J(\delta/b, \mathcal{F}/b)}{\delta^2 n^{1/2}}\right)$$

for all $\delta > 0$. Since $N(\epsilon, \mathcal{F}/b, L^2(Q)) = N(b\epsilon, \mathcal{F}, L^2(Q))$ for any $Q$, $J(\delta/b, \mathcal{F}/b) = b^{-1} J(\delta, \mathcal{F})$. Combining this with $E\|P_n - P\|_{\mathcal{F}/b} = b^{-1} E\|P_n - P\|_{\mathcal{F}}$ gives the desired bound. □

*Proof of Lem. S2.* We first show that

$$\delta \geq n^{-\rho/(2\rho+1)} 2^{-2\rho/(2\rho+1)}\left[c + c^{1/2}(c+d)^{1/2}\right]^{2\rho/(2\rho+1)}, \tag{S3}$$

implies

$$c\delta^{1-1/(2\rho)}\left(1 + \frac{d\delta^{-1-1/(2\rho)}}{n^{1/2}}\right) \leq n^{1/2}\delta^2. \tag{S4}$$

When $\delta$ equals the right-hand side of (S3), (S4) is an equality. Divide both sides of (S4) by $\delta^2$ and note that the resulting left-hand side and right-hand side are monotone decreasing and constant functions of $\delta$, respectively. Hence, (S4) holds whenever (S3) holds, that is, (S3) implies (S4). A similar argument shows $\delta \geq n^{-1/2} 2^{-1}\left[c + c^{1/2}(c+d)^{1/2}\right]$ implies

$$c\delta^{1-1/(2\rho)}\left(1 + \frac{d\delta^{-1-1/(2\rho)}}{n^{1/2}}\right) \leq n^{1/2}\delta^2.$$

Putting these two results together shows (S2) holds whenever

$$\delta \geq \left(n^{-1/2} 2^{-1}\left[c + c^{1/2}(c+d)^{1/2}\right]\right) \vee \left(n^{-\rho/(2\rho+1)} 2^{-2\rho/(2\rho+1)}\left[c + c^{1/2}(c+d)^{1/2}\right]^{2\rho/(2\rho+1)}\right).$$

By Young's inequality, $c^{1/2}(c+d)^{1/2} \leq c + d/2$, and so the right-hand side above is upper bounded by $[n^{-1/2}(c+d/4)] \vee [n^{-1/2}(c+d/4)]^{2\rho/(2\rho+1)}$, which gives the result. □

*Proof of Lem. S3.* Fix a finitely supported distribution $Q$. Note that $\delta \mapsto J_Q(\delta, \mathcal{F}) := \int_0^\delta \sqrt{1 + \log N(\epsilon, \mathcal{F}, L^2(Q))}\, d\epsilon$ is the integral of a nonincreasing function, and so is concave, and hence has nonincreasing average derivative $\delta \mapsto J_Q(\delta, \mathcal{F})/\delta$. As $Q$ was arbitrary and the pointwise supremum of a collection of nonincreasing functions is itself nonincreasing, $\delta \mapsto \sup_Q J_Q(\delta, \mathcal{F})/\delta = J(\delta, \mathcal{F})/\delta$ is nonincreasing. Hence, $\delta \mapsto J(\delta, \mathcal{F})/\delta^2$ is strictly decreasing. The result follows since this function is also positive and the product of two strictly decreasing positive functions is strictly decreasing. □

*Proof of Lem. S4.* By (S1), $J(\delta_n, \ell_{P^\star}(\underline{\Theta})) \geq \int_0^{\delta_n} d\epsilon = \delta_n$. Combining this with $J(\delta_n, \ell_{P^\star}(\underline{\Theta})) \leq n^{1/2}\delta_n^2$ gives the result. □

## C  Discussion of DoubleGen generalization bound (Thm. 1)

We begin by discussing the behavior of terms in the finite-sample bound in (4). To simplify the discussion, we do this under an asymptotic regime where $\underline{\Theta}$ may vary as $n \to \infty$.

The complexity term $\delta_n^2$ never decays to 0 faster than $1/n$, and will typically decay slower unless unless $\underline{\Theta}$ is finite-dimensional and fixed with sample size. Upper bounds on the uniform entropy integral $J(\,\cdot\,, \ell_{P^\star}(\underline{\Theta}))$, and therefore $\delta_n$, can be obtained provided $\underline{\Theta}$ is small enough so that $\ell_{P^\star}(\underline{\Theta})$ is smooth (e.g., van der Vaart & Wellner, 2023, Chaps. 2.7.1 and 2.7.2), sparsely indexed by some parameter (e.g., Bühlmann & Van De Geer, 2011, Chap. 14.2), or otherwise structured in some way (e.g., van der Vaart & Wellner, 2023, Chaps. 2.7.3 and 2.7.4). An example is given Thm. 3, where $\underline{\Theta}$ is a neural network class and $\ell_{P^\star}$ is the denoising score matching loss.

The doubly robust term $\max_{j \in [2]} \|\alpha_n^j - \alpha_P\|_{L^2(P_X)}^2 d_\Psi^2(\psi_n^j, \Psi_P)$ is a product of squared errors, making it fourth order. In contrast, the $\Omega(1/n)$ complexity term $\delta_n^2$ is only second-order. Hence, $n^{-1/4}$ rates for $\|\alpha_n^j - \alpha_P\|_{L^2(P_X)}$ and $d_\Psi(\psi_n^j, \Psi_P)$ suffice to ensure negligibility of the fourth-order term. Even slower rates can suffice when the complexity term decays more slowly. If $X$ is low-dimensional relative to $Y$ (e.g., because $Y$ is an image or text and $X$ is not), the negligibility of the fourth-order term is even more plausible. Indeed, under reasonable smoothness conditions, the mean-squared prediction error $\|\alpha_n^j - \alpha_P\|_{L^2(P_X)}^2$ from estimating the low-dimensional function $\alpha_P$ will be of smaller order than the complexity term, even before multiplying by the additional quadratic term $d_\Psi^2(\psi_n^j, \Psi_P)$.

The entropy integral $J(\delta_n/(8C_8), \ell_{P^\star}(\underline{\Theta}))$ in the definition of $\delta_n$ measures the size of the loss class $\ell_{P^\star}(\underline{\Theta})$ over all of $\underline{\Theta}$. In Thm. S2 in Appx. E, an alternative, localized version of this result is given, which instead considers the entropy of $\ell_{P^\star}(\underline{\Theta}_{\delta_n})$, where $\underline{\Theta}_{\delta_n} := \{\theta \in \underline{\Theta} : \|\ell_{P^\star}(\theta)\|_{L^2(P^\star)} \le \delta_n\}$. That result yields a similar conclusion to Thm. 1 and replaces C6 by a weaker condition, though makes additional requirements on the nuisance estimators. Under conditions, this localized generalization bound yields the same rate as one for OracleGen derived using the same techniques—see Appx. F. This oracle optimality property is notable given that OracleGen has access to all $n$ counterfactuals, whereas DoubleGen only sees a biased subset of them.

An alternative approach to deriving generalization bounds would be to apply Thm. 3 from Foster & Syrgkanis (2023), which uses local Rademacher complexity (Bartlett & Mendelson, 2006) instead of entropy integrals. While their framework applies to general risk minimization problems with nuisances, including ours, it does not explicitly characterize the fourth-order term in (4) or establish its double robustness. As Bonvini & Kennedy (2022) noted in a different setting, tailored analyses such as ours yield faster convergence guarantees when nuisances are estimated at different rates.

## D  Formal statement of DoubleGen generalization bound (Thm. 1)

In the following theorem, we let $K$ denote the universal constant from Lem. S1 for a review. We further let

$$K_0 := 2 \max_{j \in [2]} \min \left\{ C_5 C_8 \left\|\alpha_n^j/\alpha_P\right\|_{L^\infty(P_X)} + C_7(1 + C_8)C_8 d_\Psi^2(\psi_n^j, \Psi_P), \right. \tag{S5}$$

$$\left. C_8 \left[C_5 + C_7 d_\Psi^2(\psi_n^j, \Psi_P)\right] + 2\|\alpha_n^j - \alpha_P\|_{L^\infty(P_X)}^2 \left[C_5 + C_7 d_\Psi^2(\psi_n^j, \Psi_P)\right] \right\}.$$

Though we do not require this in the theorem below, if $d_\Psi^2(\psi_n^j, \Psi_P)$ is a.s. bounded, then $K_0$ is bounded by a constant under C8. If $\psi_n^j$ and $\alpha_n^j$ were perfect estimators, so that $\psi_n^j \in \Psi_P$ and $\alpha_n^j = \alpha_P$, $K_0$ would simplify to $2C_5 C_8$.

**Theorem S1** (Formal statement of Thm. 1). *Suppose C3–C8 and that there exists an empirical risk minimizer $\theta_n \in \arg\min_{\theta \in \underline{\Theta}} R_n(\theta)$. Fix a $\delta_n$ satisfying*

$$8KC_8 J(\delta_n/(8C_8), \ell_{P^\star}(\underline{\Theta})) \left[1 + 16C_4 C_8^2 J(\delta_n/(8C_8), \ell_{P^\star}(\underline{\Theta}))/(\delta_n^2 n_{\bar{j}}^{1/2})\right] \le \lfloor n/2 \rfloor^{1/2} \delta_n^2. \tag{S6}$$

*Fix $s > 0$. With probability at least $1 - e^{-s}$,*

$$\mathcal{G}_{P^\star}(\theta_n) \leq 4 \inf_{\theta \in \underline{\Theta}} \mathcal{G}_{P^\star}(\theta) + K_1 \delta_n^2 + K_2(s+2)/n + \mathbf{K_3} \max_{j \in [2]} \|\alpha_n^j - \alpha_P\|_{L^2(P_X)}^2 d_\Psi^2(\psi_n^j, \Psi_P),$$

(S7)

*where $K_1 = 1350(1 \vee 135 K_0)$, $K_2 = 2700\left[C_4 C_8 + (1 \vee 135 K_0)\right]$, and $K_3 = 13 C_7$. The final term is **doubly robust**, vanishing if $\alpha_n^j = \alpha_P$ or $\psi_n^j \in \Psi_P$ for $j \in [2]$.*

Though beyond the scope of this work, a more careful analysis could sharpen the constants.

## E    A LOCALIZED DOUBLEGEN GENERALIZATION BOUND

In this appendix, we derive a generalization bound under an alternative to the global entropy bound from the main text, C6. The bound we present here only requires the local entropy condition given below—which is weaker than the global one assumed in Thm. S1—but will also make two additional requirements. In the local entropy condition, $\ell_{P^\star}(\underline{\Theta}_\delta) := \{\ell_{P^\star}(\theta) : \theta \in \underline{\Theta}_\delta\}$ with $\underline{\Theta}_\delta := \{\theta \in \underline{\Theta} : \|\ell_{P^\star}(\theta)\|_{L^2(P^\star)} \leq \delta\}, \delta > 0$.

    **C14)** *Local uniform entropy bound:* $J(\delta, \ell_{P^\star}(\underline{\Theta}_\delta)) < \infty$ for some $\delta > 0$.

Using concentration inequalities based on the local (rather than global) entropy can sometimes yield sharper generalization bounds—see Van Der Vaart & Wellner (2011) and Chapter 14 of Wainwright (2019) for a discussion.

The first additional requirement compared to Thm. S1 is the following:

    **C15)** *At least one nuisance not estimated poorly:* it is almost surely true that, for each $j \in \{1, 2\}$, at least one of the following holds:

        (i) $d_\Psi^2(\psi_n^j, \Psi_P) \leq \left[4(1 \vee \|\alpha_P/\alpha_n^j\|_{L^\infty(P_X)}) C_8 C_{16} \|1 + \alpha_n^j/\alpha_P\|_{L^\infty(P_X)}\right]^{-1}$;

        (ii) $\|\alpha_n^j - \alpha_P\|_{L^\infty(P_X)}^2 \leq \left(8\left[1 + C_{16} d_\Psi^2(\psi_n^j, \Psi_P)\right]\right)^{-1}$.

The almost-sure requirement made in the condition can be straightforwardly relaxed to a 'w.p. at least $1 - \varepsilon$' requirement, at the cost of a slightly more involved theorem statement. The two inequalities respectively impose conditions on the quality of the estimator $\psi_n$ and $\alpha_n$. Under C8, a sufficient condition for the first inequality to hold is that $d_\Psi^2(\psi_n^j, \Psi_P) \leq [8 C_8^3 C_{16}]^{-1}$.

We will need an alternative version of C7 that measures the discrepancy of $\theta$ from $\theta_{P^\star}$ by $\|\ell_{P^\star}(\theta)\|_{L^2(P^\star)}^2$, rather than $\mathcal{G}_{P^\star}(\theta)$.

    **C16)** *Mixed-Lipschitz loss (stronger version):* there exists $C_{16} < \infty$ such that, for all $\theta \in \underline{\Theta}$, $\psi \in \Psi$, and $\psi_P \in \Psi_P$,

$$\int \left\{ \int \left[\ell_{P^\star}(\theta)(\psi(u|x)) - \ell_{P^\star}(\theta)(\psi_P(u|x))\right] \Pi(du) \right\}^2 P_X(dx)$$
$$\leq C_{16} \|\ell_{P^\star}(\theta)\|_{L^2(P^\star)}^2 d_\Psi^2(\psi, \Psi_P).$$

The above is stronger than C7 in the sense that, when it and C5 hold, C7 holds with $C_7 = C_5 C_{16}$.

In the following theorem, $K$, $K_1$, $K_2$, and $K_3$ are as defined in and above Thm. S1.

**Theorem S2** (Localized version DoubleGen generalization bound). *Suppose C3–C5, C8, and C14–C16 and that there exists an empirical risk minimizer $\theta_n \in \operatorname{argmin}_{\theta \in \underline{\Theta}} R_n(\theta)$. Fix $s > 0$ and $\delta_n > 0$ such that*

$$8 K C_8 J(2 C_8^{1/2} \delta_n, \ell_{P^\star}(\underline{\Theta}_{2 C_8^{1/2} \delta_n})) \left( 1 + \frac{16 C_4 C_8^2 J(2 C_8^{1/2} \delta_n, \ell_{P^\star}(\underline{\Theta}_{2 C_8^{1/2} \delta_n}))}{\delta_n^2 \lfloor n/2 \rfloor^{1/2}} \right) \leq \lfloor n/2 \rfloor^{1/2} \delta_n^2.$$

(S8)

*With probability at least $1 - e^{-s}$, (S7) holds.*

Since the generalization bound in (S7) has a doubly robust remainder of order $\max_j \|\alpha_n^j - \alpha_P\|_{L^2(P_X)}^2 d_\Psi^2(\psi_n^j, \Psi_P)$, the localized result is asymptotically non-vacuous (i.e. the right-hand side of Eq. S7 converges to zero in probability as $n, s \to \infty$ and $s/n \to 0$) whenever, for each $j$, $\|\alpha_n^j - \alpha_P\|_{L^2(P_X)}^2$ or $d_\Psi^2(\psi_n^j, \Psi_P)$ converges to $0$ in probability. This imposes that at least one nuisance must be estimated increasingly well as $n$ grows on all but a vanishing sequence of events. Condition C15 complements this condition by further requiring good behavior on the events in that vanishing sequence: on those events, the error of a consistent nuisance estimator must be appropriately bounded. When this condition is not satisfied, Thm. S1 can be applied instead to derive a generalization bound.

## F BENCHMARK: GENERALIZATION ERROR OF ORACLEGEN

We now establish a generalization bound for OracleGen when implemented via an empirical risk minimizer over $\underline{\Theta} \subseteq \Theta$. The resulting guarantee—which is proved using the same techniques as for the DoubleGen generalization bounds from Thms. S1 and S2—serves as a benchmark for those bounds. Despite having access only to factual rather than counterfactual outcomes, we show that DoubleGen can match this benchmark up to constant factors. When it achieves this, we call DoubleGen oracle optimal.

In what follows, $K$ is the same universal constant appearing in Thms. S1 and S2.

**Proposition S1** (Generalization bound for OracleGen). *Suppose C3–C5 and C14 and there exists an empirical risk minimizer $\theta_n^\star \in \mathrm{argmin}_{\theta \in \underline{\Theta}} R_n^\star(\theta)$. Fix $s > 0$ and $\delta_n > 0$ satisfying*

$$KJ(\delta_n, \ell_{P^\star}(\underline{\Theta}_{\delta_n})) \left[1 + C_4 J(\delta_n, \ell_{P^\star}(\underline{\Theta}_{\delta_n}))/(\delta_n^2 n^{1/2})\right] \leq n^{1/2} \delta_n^2. \tag{S9}$$

*With probability at least $1 - e^{-s}$,*
$$\mathcal{G}_{P^\star}(\theta_n^\star) \leq 3 \inf_{\theta \in \underline{\Theta}} \mathcal{G}_{P^\star}(\theta) + K_1^\star \delta_n^2 + K_2^\star (s+1)/n, \tag{S10}$$

*where $K_1^\star := 540(2 \vee 270 C_5)$ and $K_2^\star := 540 \left[C_4 + (2 \vee 270 C_5)\right]$.*

The bound for OracleGen resembles the ones for DoubleGen in Thms. S1 and S2, but requires fewer conditions and replaces the doubly robust error term by zero. This is not surprising given that OracleGen does not rely on any nuisances. The additional conditions for DoubleGen require a mixed Lipschitz loss, strong positivity, and either a stronger uniform entropy condition (Thm. S1) or the nuisances to be estimated well enough (Thm. S2). When these conditions hold and the doubly robust term is negligible, the bounds for DoubleGen can be of the same order as the one for OracleGen, making DoubleGen oracle optimal. This is the case in our study of DoubleGen diffusion modeling in Sec. 5.3. There, we also show something stronger: the rate of the generalization upper bound for DoubleGen not only matches that for OracleGen, but also, under regularity conditions, matches a minimax lower bound for all oracle algorithms—that is, those with access to counterfactual data.

*Proof of Prop. S1.* By Lem. S1, the following holds for any $\delta > 0$:

$$n^{1/2} E \|P_n^\star - P^\star\|_{\ell_{P^\star}(\underline{\Theta}_\delta)} \leq KJ(\delta, \ell_{P^\star}(\underline{\Theta}_\delta)) \left[1 + \frac{C_4 J(\delta, \ell_{P^\star}(\underline{\Theta}_\delta))}{\delta^2 n^{1/2}}\right].$$

By the choice of $\delta_n$, the right-hand side is no more than $n^{1/2} \delta^2$ when $\delta = \delta_n$ and, by Lem. S3, the same holds true for all $\delta \geq \delta_n$.

Let $\eta := 1/\max\{2, 270 C_5\} \in (0, 1)$. Applying the consequence of Talagrand's inequality given in Lem. 3.5.9 of van der Vaart & Wellner (2023) to the class $\{-\ell_{P^\star}(\theta) : \theta \in \underline{\Theta}\}$ then shows that, w.p. at least $1 - e^{-[s + \log 2]}$,

$$P^\star \ell_{P^\star}(\theta) \leq P_n^\star \ell_{P^\star}(\theta) + 135 \eta P^\star \ell_{P^\star}^2(\theta) + 135 \frac{\delta_n^2}{\eta} + 135 \left(C_4 + \frac{1}{\eta}\right) \frac{s + \log 2}{n} \quad \forall \theta \in \underline{\Theta}. \tag{S11}$$

Similarly, applying Lem. 3.5.9 of van der Vaart & Wellner (2023) to $\{\ell_{P^\star}(\theta) : \theta \in \underline{\Theta}\}$ shows that, w.p. at least $1 - e^{-[s + \log 2]}$,

$$P_n^\star \ell_{P^\star}(\theta) \leq P^\star \ell_{P^\star}(\theta) + 135 \eta P^\star \ell_{P^\star}^2(\theta) + 135 \frac{\delta_n^2}{\eta} + 135 \left(C_4 + \frac{1}{\eta}\right) \frac{s + \log 2}{n} \quad \forall \theta \in \underline{\Theta}, \tag{S12}$$

and a union bound shows that both (S11) and (S12) hold w.p. at least $1 - e^{-s}$. We assume that both of these inequalities hold hereafter and study their consequences.

Fix $\underline{\theta} \in \underline{\Theta}$. Since $\theta_n^\star$ is an empirical risk minimizer over $\underline{\Theta}$, $P_n^\star \ell_{P^\star}(\theta_n^\star) \leq P_n^\star \ell_{P^\star}(\underline{\theta})$, and so combining (S11) with $\theta = \theta_n^\star$ and (S12) with $\theta = \underline{\Theta}$ yields

$$P^\star \ell_{P^\star}(\theta_n^\star) \leq P^\star \ell_{P^\star}(\underline{\theta}) + 135\eta[P^\star \ell_{P^\star}^2(\theta_n^\star) + P^\star \ell_{P^\star}^2(\underline{\theta})] + 270\frac{\delta_n^2}{\eta} + 270\left(C_4 + \frac{1}{\eta}\right)\frac{s + \log 2}{n}.$$

Recognizing that $P^\star \ell_{P^\star}(\theta) = \mathcal{G}_{P^\star}(\theta)$ for any $\theta$, noting that $135\eta = 135/\max\{2, 270C_5\} \leq 1/(2C_5)$, leveraging C5 to upper bound $P^\star \ell_{P^\star}^2(\theta_n^\star) + P^\star \ell_{P^\star}^2(\underline{\theta})$, and subtracting $\frac{1}{2}\mathcal{G}_{P^\star}(\theta_n^\star)$ from both sides yields

$$\frac{1}{2}\mathcal{G}_{P^\star}(\theta_n^\star) \leq \frac{3}{2}\mathcal{G}_{P^\star}(\underline{\theta}) + 270\frac{\delta_n^2}{\eta} + 270\left(C_4 + \frac{1}{\eta}\right)\frac{s + \log 2}{n}.$$

Multiplying both sides by 2 and recalling the definitions of $K_1^\star$, $K_2^\star$, $\eta$, and $\delta_n$ shows that

$$\mathcal{G}_{P^\star}(\theta_n^\star) \leq 3\mathcal{G}_{P^\star}(\underline{\theta}) + 540\frac{\delta_n^2}{\eta} + 540\left(C_4 + \frac{1}{\eta}\right)\frac{s + \log 2}{n}$$
$$= 3\mathcal{G}_{P^\star}(\underline{\theta}) + K_1^\star \delta_n^2 + K_2^\star(s + \log 2)/n.$$

We have shown that, for any fixed $\underline{\theta} \in \underline{\Theta}$, the above holds w.p. at least $1 - e^{-s}$. Choosing $\underline{\theta}$ such that $3\mathcal{G}_{P^\star}(\underline{\theta}) \leq 3\inf_{\theta \in \underline{\Theta}} \mathcal{G}_{P^\star}(\theta) + K_2^\star(1 - \log 2)/n$ yields the desired result. $\qquad\square$

# G  PROOFS OF GENERAL GUARANTEES FOR DOUBLEGEN (SEC. 5)

## G.1  PROOFS OF GENERALIZATION ERROR UPPER BOUNDS (THMS. S1 AND S2)

Let $\zeta_P(\theta, x) := \int \ell_{P^\star}(\theta)(\psi_P(u \mid x))\Pi(du)$ and, for any $(\alpha_\diamond, \psi_\diamond) \in [1, C_8]^{\mathcal{X}} \times \Psi$, let $\zeta_\diamond(\theta, x) := \int \ell_{P^\star}(\theta)(\psi_\diamond(u \mid x))\Pi(du)$ and

$$L_\diamond(\theta)(z) := 1(a = a^\star)\alpha_\diamond(x)\left\{\ell_{P^\star}(\theta)(y) - \zeta_\diamond(\theta, x)\right\} + \zeta_\diamond(\theta, x).$$

Similarly, let $\zeta_n^j(\theta, x) := \int \ell_{P^\star}(\theta)(\psi_n^j(u \mid x))\Pi(du)$ and

$$L_n^j(\theta)(z) := 1(a = a^\star)\alpha_n^j(x)\left\{\ell_{P^\star}(\theta)(y) - \zeta_n^j(\theta, x)\right\} + \zeta_n^j(\theta, x).$$

**Lemma S5** (Double robustness of loss). *Suppose C3, C4, C7, and C8. Let $(s_1, s_2)$ belong to the 2-simplex and*

$$B_n := C_7^{1/2} \max_{j \in [2]} \|\alpha_n^j - \alpha_P\|_{L^2(P_X)} d_\Psi(\psi_n^j, \Psi_P).$$

*For any $\theta \in \underline{\Theta}$, $\left|\mathcal{G}_{P^\star}(\theta) - \sum_{j=1}^2 s_j PL_n^j(\theta)\right| \leq B_n \mathcal{G}_{P^\star}^{1/2}(\theta)$ and also*

$$\frac{9}{10}\mathcal{G}_{P^\star}(\theta) - \frac{5}{2}B_n^2 \leq \sum_{j=1}^2 s_j PL_n^j(\theta) \leq \frac{11}{10}\mathcal{G}_{P^\star}(\theta) + \frac{5}{2}B_n^2. \qquad (S13)$$

*Proof of Lem. S5.* Let $\alpha_P(x) := 1/P(A = a^\star \mid X = x)$ and $\psi_P \in \Psi_P$. Observe that

$$\left|\mathcal{G}_{P^\star}(\theta) - \sum_{j=1}^2 s_j PL_n^j(\theta)\right| = \left|\sum_{j=1}^2 s_j \int [1 - \alpha_n^j(x)/\alpha_P(x)]\left[\zeta_n^j(\theta, x) - \zeta_P(\theta, x)\right] P_X(dx)\right|$$
$$\leq \max_{j \in [2]} \left\|1 - \alpha_n^j/\alpha_P\right\|_{L^2(P_X)} \left[\int \left[\zeta_n^j(\theta, x) - \zeta_P(\theta, x)\right]^2 P_X(dx)\right]^{1/2}$$
$$\leq C_7^{1/2} \mathcal{G}_{P^\star}^{1/2}(\theta) \max_{j \in [2]} \left\|1 - \alpha_n^j/\alpha_P\right\|_{L^2(P_X)} d_\Psi(\psi_n^j, \Psi_P)$$
$$\leq C_7^{1/2} \mathcal{G}_{P^\star}^{1/2}(\theta) \max_{j \in [2]} \left\|\alpha_P - \alpha_n^j\right\|_{L^2(P_X)} d_\Psi(\psi_n^j, \Psi_P),$$

where the consecutive inequalities hold by convexity paired with Cauchy-Schwarz, C7, and Hölder's inequality paired with the fact that $1/\alpha_P \leq 1$. This proves the first claimed inequality.

For the remaining pair of inequalities, we combine the Peter-Paul inequality $bc \leq b^2/(2\epsilon) + c^2\epsilon/2$ (with $\epsilon = 5$) with the inequality we just established, yielding

$$\sum_{j=1}^{2} s_j PL_n^j(\theta) \geq \mathcal{G}_{P^\star}(\theta) - B_n \mathcal{G}_{P^\star}^{1/2}(\theta) \geq \frac{9}{10}\mathcal{G}_{P^\star}(\theta) - \frac{5}{2}B_n^2,$$

$$\sum_{j=1}^{2} s_j PL_n^j(\theta) \leq \mathcal{G}_{P^\star}(\theta) + B_n \mathcal{G}_{P^\star}^{1/2}(\theta) \leq \frac{11}{10}\mathcal{G}_{P^\star}(\theta) + \frac{5}{2}B_n^2.$$

$\square$

**Lemma S6** (Centered losses are similar, as measured by $L^2$ norm of loss). *Fix $j \in [2]$. If C3, C4, C8, and C16, then, for any $\theta \in \underline{\Theta}$ and $(\alpha_\diamond, \psi_\diamond) \in \{(\alpha_n^j, \psi_P) : \psi_P \in \Psi_P\} \cup \{(\alpha_P, \psi_n^j)\}$, it holds that $\|L_\diamond(\theta) - L_n^j(\theta)\|_{L^2(P)}^2 \leq C_{n\diamond}^j \|\ell_{P^\star}(\theta)\|_{L^2(P^\star)}^2$ with*

$$C_{n\diamond}^j := \begin{cases} C_8 C_{16}\|1 + \alpha_n^j/\alpha_P\|_{L^\infty(P_X)} d_\Psi^2(\psi_n^j, \Psi_P), & \text{if } (\alpha_\diamond, \psi_\diamond) \in \{(\alpha_n^j, \psi_P) : \psi_P \in \Psi_P\}, \\ 2\|\alpha_n^j - \alpha_P\|_{L^\infty(P_X)}^2 \left[1 + C_{16}d_\Psi^2(\psi_n^j, \Psi_P)\right], & \text{if } (\alpha_\diamond, \psi_\diamond) = (\alpha_P, \psi_n^j). \end{cases}$$

*Proof of Lem. S6.* The proof is broken into two cases.

**Case 1:** $(\alpha_\diamond, \psi_\diamond) \in \{(\alpha_n^j, \psi_P) : \psi_P \in \Psi_P\}$. In this case $L_\diamond(\theta)(z) - L_n^j(\theta)(z) = [1(a = a^\star)\alpha_n^j(x) - 1][\zeta_n^j(\theta, x) - \zeta_P(\theta, x)]$, and so, by the law of total expectation and C8 and C16,

$$\|L_\diamond(\theta) - L_n^j(\theta)\|_{L^2(P)}^2 = \int \left\{[1(a = a^\star)\alpha_n^j(x) - 1][\zeta_n^j(\theta, x) - \zeta_P(\theta, x)]\right\}^2 P(dz)$$

$$= \int (1 - 2\alpha_n^j/\alpha_P + (\alpha_n^j)^2/\alpha_P)(x) \left[\zeta_n^j(\theta, x) - \zeta_P(\theta, x)\right]^2 P(dz)$$

$$\leq C_8\|1 + \alpha_n^j/\alpha_P\|_{L^\infty(P_X)} \int \left[\zeta_n^j(\theta, x) - \zeta_P(\theta, x)\right]^2 P(dz)$$

$$\leq C_8 C_{16}\|1 + \alpha_n^j/\alpha_P\|_{L^\infty(P_X)} d_\Psi^2(\psi_n^j, \Psi_P)\|\ell_{P^\star}(\theta)\|_{L^2(P^\star)}^2.$$

**Case 2:** $(\alpha_\diamond, \psi_\diamond) = (\alpha_P, \psi_n^j)$. In this case,

$$L_\diamond(\theta)(z) - L_n^j(\theta)(z) = 1(a = a^\star)(\alpha_P - \alpha_n^j)(x)[\ell_{P^\star}(\theta)(y) - \zeta_P(\theta, x)]$$
$$+ 1(a = a^\star)(\alpha_n^j - \alpha_P)(x)[\zeta_n^j(\theta, x) - \zeta_P(\theta, x)].$$

Applying the basic inequality $(b + c)^2 \leq 2(b^2 + c^2)$ followed by Hölder's inequality yields

$$\|L_\diamond(\theta) - L_n^j(\theta)\|_{L^2(P)}^2 \leq 2\|\alpha_P - \alpha_n^j\|_{L^\infty(P_X)}^2 \int 1(a = a^\star)[\ell_{P^\star}(\theta)(y) - \zeta_P(\theta, x)]^2 P(dz)$$
$$+ 2\|\alpha_P - \alpha_n^j\|_{L^\infty(P_X)}^2 \int [\zeta_n^j(\theta, x) - \zeta_P(\theta, x)]^2 P(dz).$$

By C16, the integral in the second term is upper bounded by $C_{16}d_\Psi^2(\psi_n^j, \Psi_P)\|\ell_{P^\star}(\theta)\|_{L^2(P^\star)}^2$. Using that conditional means are $L^2$ projections and (1), the integral in the first term upper bounds as follows:

$$\int 1(a = a^\star)[\ell_{P^\star}(\theta)(y) - \zeta_P(\theta, x)]^2 P(dz) \leq \int 1(a = a^\star)\ell_{P^\star}^2(\theta)(y) P(dz) \leq \|\ell_{P^\star}(\theta)\|_{L^2(P^\star)}^2.$$

Putting these bounds together yields the desired upper bound on $\|L_\diamond(\theta) - L_n^j(\theta)\|_{L^2(P)}^2$ for the second case. $\square$

**Lemma S7** (Centered losses are similar, as measured by generalization error). *Fix $j \in [2]$. If C3–C5, C7, and C8, then, for any $\theta \in \underline{\Theta}$ and $(\alpha_\diamond, \psi_\diamond) \in \{(\alpha_n^j, \psi_P) : \psi_P \in \Psi_P\} \cup \{(\alpha_P, \psi_n^j)\}$, it holds that $\|L_\diamond(\theta) - L_n^j(\theta)\|_{L^2(P)}^2 \leq \widetilde{C}_{n\diamond}^j \mathcal{G}_{P^\star}(\theta)$ with*

$$\widetilde{C}_{n\diamond}^j := \begin{cases} C_8 C_7\|1 + \alpha_n^j/\alpha_P\|_{L^\infty(P_X)} d_\Psi^2(\psi_n^j, \Psi_P), & \text{if } (\alpha_\diamond, \psi_\diamond) = (\alpha_n^j, \psi_P), \\ 2\|\alpha_n^j - \alpha_P\|_{L^\infty(P_X)}^2 \left[C_5 + C_7 d_\Psi^2(\psi_n^j, \Psi_P)\right], & \text{if } (\alpha_\diamond, \psi_\diamond) = (\alpha_P, \psi_n^j). \end{cases}$$

*Proof of Lem. S7.* The proof of this result is nearly identical to that of Lem. S6 and so is omitted. □

**Lemma S8** (Bounding norms of $\ell_{P^\star}$ by generalization error). *Fix $j \in [2]$. Suppose C3, C4, and C8. For any $(\alpha_\diamond, \psi_\diamond) \in \{(\alpha_n^j, \psi_P) : \psi_P \in \Psi_P\} \cup \{(\alpha_P, \psi_n^j)\}$ and $\theta \in \underline{\Theta}$, $\|\ell_{P^\star}(\theta)\|_{L^2(P^\star)}^2 \leq C_\diamond \|L_\diamond(\theta)\|_{L^2(P)}^2$, where $C_\diamond := 1 \vee \|\alpha_P/\alpha_\diamond\|_{L^\infty(P_X)} \leq C_8$.*

*Proof of Lem. S8.* Observe that

$$\|\ell_{P^\star}(\theta)\|_{L^2(P^\star)}^2 = \int 1(a = a^\star)\alpha_P(x)\ell_{P^\star}^2(\theta)(y)P(dz)$$

$$= \int 1(a = a^\star)\alpha_P(x) \left[\ell_{P^\star}(\theta)(y) - \zeta_P(\theta, x)\right]^2 P(dz) + \int \zeta_P^2(\theta, x)P(dz), \quad \text{(S14)}$$

where the second equality holds by adding and subtracting a term inside the square, expanding the square and noticing that the cross term is zero, and finally using the law of total expectation to replace the $1(a = a^\star)\alpha_P(x)$ in the second term on by 1. The remainder of the proof is broken into two cases.

**Case 1:** $\alpha_\diamond = \alpha_P$. In this case $C_\diamond = 1$. Starting with (S14), and then using that $\alpha_P = \alpha_\diamond \geq 1$ and applying the law of total expectation, we find that

$$\|\ell_{P^\star}(\theta)\|_{L^2(P^\star)}^2$$

$$\leq \int \left[1(a = a^\star)\alpha_\diamond(x)\{\ell_{P^\star}(\theta)(y) - \zeta_P(\theta, x)\}\right]^2 P(dz) + \int \zeta_P^2(\theta, x)P(dz)$$

$$= \int \left[1(a = a^\star)\alpha_\diamond(x)\{\ell_{P^\star}(\theta)(y) - \zeta_P(\theta, x)\} + \zeta_P(\theta, x)\right]^2 P(dz)$$

$$= \int \left[1(a = a^\star)\alpha_\diamond(x)\{\ell_{P^\star}(\theta)(y) - \zeta_\diamond(\theta, x)\} + \zeta_\diamond(\theta, x)\right]^2 P(dz)$$

$$+ \int \left[1 - \alpha_\diamond(x)\right]\left[\zeta_P(\theta, x) - \zeta_\diamond(\theta, x)\right]^2 P(dz)$$

$$\leq \int \left[1(a = a^\star)\alpha_\diamond(x)\{\ell_{P^\star}(\theta)(y) - \zeta_\diamond(\theta, x)\} + \zeta_\diamond(\theta, x)\right]^2 P(dz) = \|L_\diamond(\theta)\|_{L^2(P)}^2.$$

**Case 2:** $\alpha_\diamond \neq \alpha_P$. Since $(\alpha_\diamond, \psi_\diamond) \in \{(\alpha_n^j, \psi_P) : \psi_P \in \Psi_P\} \cup \{(\alpha_P, \psi_n^j)\}$, $\psi_\diamond = \psi_P \in \Psi_P$ in this case, and so $\zeta_\diamond = \zeta_P$. Recalling (S14) and applying C8, Hölder's inequality and the basic inequality $\|\alpha_P/\alpha_\diamond\|_{L^\infty(P_X)}b + c \leq (\|\alpha_P/\alpha_\diamond\|_{L^\infty(P_X)} \vee 1)(b + c) = C_\diamond(b + c)$ yields

$$\|\ell_{P^\star}(\theta)\|_{L^2(P^\star)}^2/C_\diamond \leq \int \left[1(a = a^\star)\alpha_\diamond(x)\{\ell_{P^\star}(\theta)(y) - \zeta_\diamond(\theta, x)\}\right]^2 P(dz) + \int \left[\zeta_\diamond(\theta, x)\right]^2 P(dz).$$

Using the identity $\int f^2 + \int g^2 = \int (f + g)^2 - 2\int fg$ and noticing that $\int fg$ is equal to zero when this identity is applied on the right-hand side above, this shows that $\|\ell_{P^\star}(\theta)\|_{L^2(P^\star)}^2/C_\diamond \leq \|L_\diamond(\theta)\|_{L^2(P)}^2$. Since $\alpha_\diamond \geq 1$, C8 yields $C_\diamond \leq C_8$. □

**Lemma S9** (Bounding norms of $L_\diamond$ by norms of $\ell_{P^\star}$). *Fix $j \in [2]$. Suppose $(\alpha_\diamond, \psi_\diamond) \in \{(\alpha_n^j, \psi_P) : \psi_P \in \Psi_P\} \cup \{(\alpha_P, \psi_n^j)\}$ and that C3–C5, C7, and C8 hold. Then, for any $\theta \in \underline{\Theta}$,*

$$\|L_\diamond(\theta)\|_{L^2(P)}^2 \leq C_8 \left[C_5 \|\alpha_\diamond/\alpha_P\|_{L^\infty(P_X)} + C_7 d_\Psi^2(\psi_\diamond, \Psi_P)\right] \mathcal{G}_{P^\star}(\theta).$$

*Proof of Lem. S9.* Let $\Pi_{A,X}[L_\diamond(\theta)](z) := E_P[L_\diamond(\theta)(Z) \mid A = a, X = x]$ and $\Pi_X[L_\diamond(\theta)](z) := E_P[L_\diamond(\theta)(Z) \mid X = x]$. By the law of total expectation,

$$\|L_\diamond(\theta)\|_{L^2(P)}^2 = \|L_\diamond(\theta) - \Pi_{A,X}[L_\diamond(\theta)]\|_{L^2(P)}^2 + \|\Pi_X[L_\diamond(\theta)]\|_{L^2(P)}^2$$

$$+ \|\Pi_{A,X}[L_\diamond(\theta)] - \Pi_X[L_\diamond(\theta)]\|_{L^2(P)}^2. \quad \text{(S15)}$$

We begin by bounding the sum of the first two terms on the right, and then we conclude by bounding the third. These calculations will all use that $\Pi_{A,X}[L_\diamond(\theta)](z) = 1(a = a^\star)\alpha_\diamond(x)[\zeta_P(\theta, x) - \zeta_\diamond(\theta, x)] + \zeta_\diamond(\theta, x)$ and $\Pi_X[L_\diamond(\theta)](z) = \zeta_P(\theta, x)$, where the first holds by the law of total expectation and the second by the double robustness of the loss $L_\diamond$. Studying the first term above, we apply the law of total expectation, Hölder's inequality, and C8 to show that

$$\|L_\diamond(\theta) - \Pi_{A,X}[L_\diamond(\theta)]\|_{L^2(P)}^2$$

$$= \int \left[ 1(a = a^\star)\alpha_\diamond(x) \left\{ \ell_{P^\star}(\theta)(y) - \zeta_P(\theta, x) \right\} \right]^2 P(dz)$$

$$= \iint \alpha_\diamond(x) \frac{\alpha_\diamond(x)}{\alpha_P(x)} \left[ \ell_{P^\star}(\theta)(y) - \zeta_P(\theta, x) \right]^2 P_{Y|A,X}(dy|a^\star, x) P_X(dx)$$

$$\leq C_8 \|\alpha_\diamond/\alpha_P\|_{L^\infty(P_X)} \iint \left[ \ell_{P^\star}(\theta)(y) - \zeta_P(\theta, x) \right]^2 P_{Y|A,X}(dy|a^\star, x) P_X(dx).$$

If we add the second term from (S15), $\|\Pi_X[L_\diamond(\theta)]\|_{L^2(P)}^2 = \int \zeta_P^2(\theta, x) P_X(dx)$, to both sides above and then further upper bound the right-hand side, we find that

$$\|L_\diamond(\theta) - \Pi_{A,X}[L_\diamond(\theta)]\|_{L^2(P)}^2 + \|\Pi_X[L_\diamond(\theta)]\|_{L^2(P)}^2$$

$$\leq C_8 \|\alpha_\diamond/\alpha_P\|_{L^\infty(P_X)} \iint \left\{ \left[ \ell_{P^\star}(\theta)(y) - \zeta_P(\theta, x) \right]^2 + \zeta_P^2(\theta, x) \right\} P_{Y|A,X}(dy|a^\star, x) P_X(dx).$$

The double integral above equals $\|\ell_{P^\star}(\theta)\|_{L^2(P^\star)}^2$, and so by C5 the left-hand side is bounded by $C_5 C_8 \|\alpha_\diamond/\alpha_P\|_{L^\infty(P_X)} \mathcal{G}_{P^\star}(\theta)$.

For the third term in (S15), we use the identities we derived for $\Pi_{A,X}[L_\diamond(\theta)]$ and $\Pi_X[L_\diamond(\theta)]$, expand a square, and apply the law of total expectation to find that

$$\|\Pi_{A,X}[L_\diamond(\theta)] - \Pi_X[L_\diamond(\theta)]\|_{L^2(P)}^2$$

$$= \int \left[ 1 - 1(a = a^\star)\alpha_\diamond(x) \right]^2 \left[ \zeta_\diamond(\theta, x) - \zeta_P(\theta, x) \right]^2 P(dz)$$

$$= \int \left[ 1 - 2\alpha_\diamond(x)/\alpha_P(x) + \alpha_\diamond^2(x)/\alpha_P(x) \right] \left[ \zeta_\diamond(\theta, x) - \zeta_P(\theta, x) \right]^2 P(dz).$$

We will show that the right-hand side is no more than $C_8 C_7 d_\Psi^2(\psi_\diamond, \Psi_P) \mathcal{G}_{P^\star}(\theta)$. If $\zeta_\diamond = \zeta_P$, then this claimed bound is zero and so is the right-hand side above; hence, the bound is valid. Otherwise, $\alpha_\diamond = \alpha_P$, and so $1 - 2\alpha_\diamond/\alpha_P + \alpha_\diamond^2/\alpha_P = \alpha_\diamond - 1 \leq C_8$. Combining this with C7 shows that the right-hand side is indeed no more than $C_8 C_7 d_\Psi^2(\psi_\diamond, \Psi_P) \mathcal{G}_{P^\star}(\theta)$.

Combining our bounds for the three terms in (S15) gives the desired result. $\qquad\square$

In the following lemma, we let

$$K_0^j := 2 \min \left\{ C_5 C_8 \left\| \alpha_n^j/\alpha_P \right\|_{L^\infty(P_X)} + C_7(1 + C_8) C_8 d_\Psi^2(\psi_n^j, \Psi_P), \right.$$

$$\left. C_8 \left[ C_5 + C_7 d_\Psi^2(\psi_n^j, \Psi_P) \right] + 2\|\alpha_n^j - \alpha_P\|_{L^\infty(P_X)}^2 \left[ C_5 + C_7 d_\Psi^2(\psi_n^j, \Psi_P) \right] \right\}.$$

Note that $K_0$ from (S5) is then equal to $\max_j K_0^j$.

**Lemma S10** (Bounding $L^2$ norm of loss by generalization error). *Fix $j \in [2]$ and $\theta \in \underline{\Theta}$. If C3–C5, C7, and C8, then $\|L_n^j(\theta)\|_{L^2(P)}^2 \leq K_0^j \mathcal{G}_{P^\star}(\theta)$.*

*Proof of Lem. S10.* By Lems. S7 and S9, the triangle inequality, and the basic inequality $(b + c)^2 \leq 2(b^2 + c^2)$,

$$\|L_n^j(\theta)\|_{L^2(P)}^2 \leq 2 \left( C_8 \left[ C_5 \|\alpha_\diamond/\alpha_P\|_{L^\infty(P_X)} + C_7 d_\Psi^2(\psi_\diamond, \Psi_P) \right] + \widetilde{C}_{n\diamond}^j \right) \mathcal{G}_{P^\star}(\theta)$$

for any $(\alpha_\diamond, \psi_\diamond) \in \{(\alpha_n^j, \psi_P) : \psi_P \in \Psi_P\} \cup \{(\alpha_P, \psi_n^j)\}$. In particular, $(\alpha_\diamond, \psi_\diamond)$ can be chosen to minimize the right-hand side. Upper bounding further to simplify the expression then yields $\|L_n^j(\theta)\|_{L^2(P)}^2 \leq K_0^j \mathcal{G}_{P^\star}(\theta)$. $\qquad\square$

For $j \in [2]$ and $\delta > 0$, let $\mathcal{L}_{n,\delta}^j := \{L_n^j(\theta) : \theta \in \underline{\Theta}, \|L_n^j(\theta)\|_{L^2(P)} \leq \delta\}$.

**Lemma S11** (Constant envelope for $\mathcal{L}_{n,\delta}^j$). *Fix $j \in [2]$. Suppose C3, C4, and C8. Conditionally on $\mathcal{Z}_{3-j}$, the following holds for all $z \in \mathcal{Z}$: $\sup_{\theta \in \underline{\Theta}} |L_n^j(\theta)(z)| \leq 2C_4 C_8$.*

*Proof of Lem. S11.* First applying Jensen's inequality and then applying C4 and C8 yields

$$\sup_{\theta \in \underline{\Theta}} |L_n^j(\theta)(z)|$$

$$\leq \sup_{\theta \in \underline{\Theta}} \int \left[ 1(a = a^\star)\alpha_n^j(x) |\ell_{P^\star}(\theta)(y)| + \left| 1 - 1(a = a^\star)\alpha_n^j(x) \right| |\ell_{P^\star}(\theta)(\psi_n^j(u|x))| \right] \Pi(du)$$

$$\leq C_8 C_4 + \max\{1, C_8 - 1\}C_4 \leq 2C_8 C_4.$$

$\square$

**Lemma S12** (Preliminary entropy integral bound). *Fix $j \in [2]$. Suppose C3, C4, and C8. For any $\widetilde{\Theta} \subseteq \underline{\Theta}$ and $\delta > 0$, $J(8C_8\delta, L_n^j(\widetilde{\Theta})) \leq 8C_8 J(\delta, \ell_{P^\star}(\widetilde{\Theta}))$.*

*Proof of Lem. S12.* The bound is trivial when $J(\delta, \ell_{P^\star}(\widetilde{\Theta}) \mid C_4, L^2) = \infty$, so we focus on the case where this quantity is finite. Let $Q$ be a distribution on $\mathcal{X} \times \mathcal{A} \times \mathcal{Y}$ with support $\{(x_i, a_i, y_i)\}_{i=1}^v$. Let $\mathbb{Q}^{(1)}$ be the distribution supported on $\{y_i\}_{i=1}^v$ that satisfies $\mathbb{Q}^{(1)}\{y_i\} = Q\{(x_{i'}, a_{i'}, y_{i'}) : y_{i'} = y_i\}$ and $\mathbb{Q}^{(2)}$ be the distribution on $\mathcal{Y}$ satisfying $\mathbb{Q}^{(2)}(\mathcal{B}) = \sum_{i=1}^v Q\{(x_i, a_i, y_i)\} \int 1\{\psi_n^j(u|x_i) \in \mathcal{B}\}\Pi(du)$ for all measurable $\mathcal{B} \subseteq \mathcal{Y}$. Further let $\mathbb{Q}^{(3)} = (\mathbb{Q}^{(1)} + \mathbb{Q}^{(2)})/2$.

Let $\{f_i\}_{i=1}^N$ be a minimal $4\epsilon$-cover of $\ell_{P^\star}(\widetilde{\Theta})$ with respect to $L^2(\mathbb{Q}^{(3)})$ and

$$g_i(x, a, y) := \int \left[ 1(a = a^\star)\alpha_n^j(x) \left\{ f_i(y) - f_i(\psi_n^j(u|x)) \right\} + f_i(\psi_n^j(u|x)) \right] \Pi(du).$$

We will show that $\{g_i\}_{i=1}^N$ is an $8C_8\epsilon$-cover of $L_n^j(\widetilde{\Theta})$ with respect to $L^2(Q)$. To this end, fix $\theta \in \widetilde{\Theta}$ and let $f_i$ be such that $\|\ell_{P^\star}(\theta) - f_i\|_{L^2(\mathbb{Q}^{(3)})} \leq 4\epsilon$. Observe that

$$L_n^j(\theta)(z) - g_i(z) = 1(a = a^\star)\alpha_n^j(x)[\ell_{P^\star}(\theta)(y) - f_i(y)]$$

$$+ \int \left[ 1 - 1(a = a^\star)\alpha_n^j(x) \right] \left[ \ell_{P^\star}(\theta)(\psi_n^j(u|x)) - f_i(\psi_n^j(u|x)) \right] \Pi(du).$$

Combining this with the triangle inequality, Jensen's inequality, and C8 shows that $\|L_n^j(\theta) - g_i\|_{L^2(Q)} \leq C_8(\|\ell_{P^\star}(\theta) - f_i\|_{L^2(\mathbb{Q}^{(1)})} + \|\ell_{P^\star}(\theta) - f_i\|_{L^2(\mathbb{Q}^{(2)})})$. Squaring both sides, applying the inequality $(b + c)^2 \leq 2(b^2 + c^2)$, and recalling the definitions of $\mathbb{Q}^{(3)}$ and $f_i$ then shows that $\|L_n^j(\theta) - g_i\|_{L^2(Q)}^2 \leq 2C_8^2 \|\ell_{P^\star}(\theta) - f_i\|_{L^2(\mathbb{Q}^{(1)} + \mathbb{Q}^{(2)})}^2 = 4C_8^2 \|\ell_{P^\star}(\theta) - f_i\|_{L^2(\mathbb{Q}^{(3)})}^2 = 64C_8^2\epsilon^2$. Hence, $\{g_i\}_{i=1}^N$ is an $8C_8\epsilon$-cover of $L_n^j(\widetilde{\Theta})$ with respect to $L^2(Q)$. As $\epsilon$ was arbitrary, we have shown that, for any $\gamma \in (0, \delta)$,

$$\int_\gamma^\delta \sqrt{1 + \log N(8C_8\epsilon, L_n^j(\widetilde{\Theta}), L^2(Q))}\, d\epsilon \leq \int_\gamma^\delta \sqrt{1 + \log N(4\epsilon, \ell_{P^\star}(\widetilde{\Theta}), L^2(\mathbb{Q}^{(3)}))}\, d\epsilon. \quad \text{(S16)}$$

The next part of this proof deals with the fact that the uniform entropy integral $J$ is defined using $L^2$ covering numbers with respect to finitely supported distributions, a property that $\mathbb{Q}^{(3)}$ may not satisfy. In particular, we show that there exists a finitely supported distribution $\mathbb{Q}$ on $\mathcal{Y}$ such that

$$\int_\gamma^\delta \sqrt{1 + \log N(4\epsilon, \ell_{P^\star}(\widetilde{\Theta}), L^2(\mathbb{Q}^{(3)}))}\, d\epsilon \leq \int_\gamma^\delta \sqrt{1 + \log N(\epsilon, \ell_{P^\star}(\widetilde{\Theta}), L^2(\mathbb{Q}))}\, d\epsilon. \quad \text{(S17)}$$

This argument is based on the hint given on Problem 2.5.1 of van der Vaart & Wellner (2023), which leverages the following relationship between covering numbers $N$ and packing numbers $M$: $N(\epsilon) \leq M(\epsilon) \leq N(\epsilon/2)$ (van der Vaart & Wellner, 2023, page 147). In particular, this relationship shows it suffices to exhibit a finitely supported distribution $\mathbb{Q}$ on $\mathcal{Y}$ that satisfies $M(4\epsilon, \ell_{P^\star}(\widetilde{\Theta}), L^2(\mathbb{Q}^{(3)})) \leq M(2\epsilon, \ell_{P^\star}(\widetilde{\Theta}), L^2(\mathbb{Q}))$ for all $\epsilon \in [\gamma, \delta]$. To this end, for each $m \in \mathcal{I} := \{M(4\gamma, \ell_{P^\star}(\widetilde{\Theta}), L^2(\mathbb{Q}^{(3)})), M(4\gamma, \ell_{P^\star}(\widetilde{\Theta}), L^2(\mathbb{Q}^{(3)})) + 1, \ldots, M(4\delta, \ell_{P^\star}(\widetilde{\Theta}), L^2(\mathbb{Q}^{(3)}))\}$, we let $\{f_{m,j}\}_{j=1}^m$ denote a maximal packing of $\ell_{P^\star}(\widetilde{\Theta})$ with respect to $L^2(\mathbb{Q}^{(3)})$. Let $\{Y_{i'}^\star(\omega)\}_{i'=1}^\infty$ be a sequence of iid draws from $\mathbb{Q}^{(3)}$ and denote the empirical distribution of its first $k$ draws by $\mathbb{Q}_k(\omega)$. By the strong law of large numbers, for all $\omega$ belonging to a probability one set $\Omega_\gamma$, there exists $K(\omega) < \infty$ such that, for all $m \in \mathcal{I}$, $\|f_{m,i} - f_{m,i'}\|_{L^2(\mathbb{Q}_{K(\omega)}(\omega))} \geq \|f_{m,i} - f_{m,i'}\|_{L^2(\mathbb{Q}^{(3)})} - 2\gamma$. Fixing some $\omega \in \Omega_\gamma$ and letting $\mathbb{Q}$ be the finitely supported distribution $\mathbb{Q}_{K(\omega)}(\omega)$ then shows that $M(4\epsilon, \ell_{P^\star}(\widetilde{\Theta}), L^2(\mathbb{Q}^{(3)})) \leq M(2\epsilon, \ell_{P^\star}(\widetilde{\Theta}), L^2(\mathbb{Q}))$ for all $\epsilon \in [\gamma, \delta]$, which establishes (S17).

Combining (S16) and (S17), taking a limit as $\gamma \downarrow 0$, and then taking a supremum on the right over all finitely supported distributions $\mathbb{Q}$ on $\mathcal{Y}$ followed by a supremum on the left over all finitely supported distributions $Q$ on $\mathcal{X} \times \mathcal{A} \times \mathcal{Y}$ yields

$$\sup_Q \int_0^\delta \sqrt{1 + \log N(8C_8\epsilon, L_n^j(\widetilde{\Theta}), L^2(Q))} \, d\epsilon \leq \sup_{\mathbb{Q}} \int_0^\delta \sqrt{1 + \log N(\epsilon, \ell_{P^\star}(\widetilde{\Theta}), L^2(\mathbb{Q}))} \, d\epsilon.$$

Applying the change of variables $\tilde{\epsilon} = 8C_8\epsilon$ on the left yields the result. $\qquad\square$

**Lemma S13** (Entropy integral bounds for $\mathcal{L}_{n,\delta}^j$). *Fix $j \in [2]$ and $(\alpha_\diamond, \psi_\diamond) \in \{(\alpha_n^j, \psi_P) : \psi_P \in \Psi_P\} \cup \{(\alpha_P, \psi_n^j)\}$. Both of the following bounds are valid under the stated conditions:*

$$J(\delta, \mathcal{L}_{n,\delta}^j) \leq \begin{cases} 8C_8 J(\delta/(8C_8), \ell_{P^\star}(\underline{\Theta})), & \text{if C3, C4, and C8 hold,} \\ 8C_8 J(2C_8^{1/2}\delta, \ell_{P^\star}(\underline{\Theta}_{2C_8^{1/2}\delta})), & \text{if C3, C4, C8, C15, and C16 hold.} \end{cases}$$

*Proof of Lem. S13.* We begin by proving the first bound. By Lem. S12 with $\widetilde{\Theta} = \{\theta \in \underline{\Theta} : \|L_n^j(\theta)\|_{L^2(P)} \leq \delta\}$, $J(\delta, \mathcal{L}_{n,\delta}^j) \leq 8C_8 J(\delta/(8C_8), \ell_{P^\star}(\widetilde{\Theta}))$. Since $\ell_{P^\star}(\widetilde{\Theta}) \subseteq \ell_{P^\star}(\underline{\Theta})$, $J(\delta/(8C_8), \ell_{P^\star}(\widetilde{\Theta})) \leq J(\delta/(8C_8), \ell_{P^\star}(\underline{\Theta}))$. Combining the inequalities from the preceding two sentences gives the result.

We now prove the second bound. If (i) from C15 holds we take $(\alpha_\diamond, \psi_\diamond) = (\alpha_n^j, \psi_P)$ for some $\psi_P \in \Psi_P$, and otherwise we take $(\alpha_\diamond, \psi_\diamond) = (\alpha_P, \psi_n^j)$. For any $\theta \in \underline{\Theta}$, Lem. S8, the triangle inequality, Lem. S6, and C15 yield

$$\|\ell_{P^\star}(\theta)\|_{L^2(P^\star)} \leq C_\diamond^{1/2} \|L_\diamond(\theta)\|_{L^2(P)} \leq C_\diamond^{1/2} \left( \|L_n^j(\theta)\|_{L^2(P)} + \|L_n^j(\theta) - L_\diamond(\theta)\|_{L^2(P)} \right)$$

$$\leq C_\diamond^{1/2} \left( \|L_n^j(\theta)\|_{L^2(P)} + (C_{n\diamond}^j)^{1/2} \|\ell_{P^\star}(\theta)\|_{L^2(P^\star)} \right)$$

$$\leq C_\diamond^{1/2} \|L_n^j(\theta)\|_{L^2(P)} + \|\ell_{P^\star}(\theta)\|_{L^2(P^\star)}/2.$$

Hence, $\|\ell_{P^\star}(\theta)\|_{L^2(P^\star)} \leq 2C_\diamond^{1/2} \|L_n^j(\theta)\|_{L^2(P)}$. Since $C_\diamond^{1/2} \leq C_8^{1/2}$, this yields the bound $\|\ell_{P^\star}(\theta)\|_{L^2(P^\star)} \leq 2C_8^{1/2} \|L_n^j(\theta)\|_{L^2(P)}$.

Hence, letting $\underline{\Theta}_{2C_8^{1/2}\delta} := \{\theta \in \underline{\Theta} : \|\ell_{P^\star}(\theta)\|_{L^2(P^\star)} \leq 2C_8^{1/2}\delta\}$, the following holds for any $\delta > 0$:

$$\mathcal{L}_{n,\delta}^j := \{L_n^j(\theta) : \theta \in \underline{\Theta}, \|L_n^j(\theta)\|_{L^2(P)} \leq \delta\} \subseteq L_n^j(\underline{\Theta}_{2C_8^{1/2}\delta}).$$

Hence, $J(\delta, \mathcal{L}_{n,\delta}^j) \leq J(\delta, L_n^j(\underline{\Theta}_{2C_8^{1/2}\delta}))$. Combining this with Lem. S12 with $\widetilde{\Theta} = \underline{\Theta}_{2C_8^{1/2}\delta}$ gives $J(\delta, \mathcal{L}_{n,\delta}^j) \leq 8C_8 J(\delta/(8C_8), \ell_{P^\star}(\underline{\Theta}_{2C_8^{1/2}\delta}))$. By the monotonicity of $J$ in its first argument and the fact that $1/(8C_8) < 2C_8^{1/2}$, this implies that $J(\delta, \mathcal{L}_{n,\delta}^j) \leq 8C_8 J(2C_8^{1/2}\delta, \ell_{P^\star}(\underline{\Theta}_{2C_8^{1/2}\delta}))$. $\qquad\square$

*Proof of Thm. S1.* For now fix $j \in [2]$. Let $\bar{j} := 3 - j$, $n_{\bar{j}} := |\mathcal{Z}_{\bar{j}}|$, and $P_{n,\bar{j}}$ denote the empirical distribution of the observations in $\mathcal{Z}_{\bar{j}}$. By Lems. S1, S11 and S13, the following holds for any $\delta > 0$:

$$n_{\bar{j}}^{1/2} E \|P_{n,j} - P\|_{\mathcal{L}_{n,\delta}^j} \leq K J(\delta, \mathcal{L}_{n,\delta}^j) \left[ 1 + \frac{2C_4 C_8 J(\delta, \mathcal{L}_{n,\delta}^j)}{\delta^2 n_{\bar{j}}^{1/2}} \right]$$

$$\leq 8KC_8 J(\delta/(8C_8), \ell_{P^\star}(\underline{\Theta})) \left[ 1 + \frac{16C_4 C_8^2 J(\delta/(8C_8), \ell_{P^\star}(\underline{\Theta}))}{\delta^2 n_{\bar{j}}^{1/2}} \right].$$
$$\text{(S18)}$$

By (S6) and the fact that $n_{\bar{j}} \geq \lfloor n/2 \rfloor$, the right-hand side is no more than $n_{\bar{j}}^{1/2}\delta^2$ when $\delta = \delta_n$ and, by Lem. S3, the same holds true when $\delta \geq \delta_n$.

Let $\eta^j := 1/\max\{2, 270K_0^j\} \in (0,1)$, which is deterministic conditionally on $\mathcal{Z}_n^j$ and the $j$ superscript on $\eta^j$ denotes the current fold rather than exponentiation. Applying the consequence of

Talagrand's inequality given in Lem. 3.5.9 of van der Vaart & Wellner (2023) to the class $\{-L_n^j(\theta) : \theta \in \underline{\Theta}\}$, which is bounded by Lem. S11, shows that, w.p. at least $1 - e^{-[s+\log 4]}$ conditionally on $\mathcal{Z}_n^j$,

$$PL_n^j(\theta) \leq P_{n,\bar{j}}L_n^j(\theta) + 135\eta^j\|L_n^j(\theta)\|_{L^2(P)}^2 + 135\frac{\delta_n^2}{\eta^j} + 135\left(2C_8C_4 + \frac{1}{\eta^j}\right)\frac{s+\log 4}{n_{\bar{j}}} \quad \forall\theta \in \underline{\Theta}.$$

Similarly, applying Lem. 3.5.9 of van der Vaart & Wellner (2023) to $\{L_n^j(\theta) : \theta \in \underline{\Theta}\}$ shows that, w.p. at least $1 - e^{-[s+\log 4]}$ conditionally on $\mathcal{Z}_n^j$,

$$P_{n,\bar{j}}L_n^j(\theta) \leq PL_n^j(\theta) + 135\eta^j\|L_n^j(\theta)\|_{L^2(P)}^2 + 135\frac{\delta_n^2}{\eta^j} + 135\left(2C_8C_4 + \frac{1}{\eta^j}\right)\frac{s+\log 4}{n_{\bar{j}}} \quad \forall\theta \in \underline{\Theta}.$$

Since each of the above inequalities holds w.p. at least $1 - e^{-[s+\log 4]}$ conditionally on $\mathcal{Z}_n^j$, they each also hold marginally w.p. at least $1 - e^{-[s+\log 4]}$.

Hereafter we work on the event where the above two inequalities hold for both $j \in [2]$, which occurs with marginal probability at least $1 - e^{-s}$ by a union bound. For the first of these inequalities, we are only concerned with the fact that it holds for $\theta = \theta_n$ and, for the second, with the fact that it holds for a generic fixed $\underline{\theta} \in \underline{\Theta}$. Applying Lem. S10 to the inequalities and using that $135\eta^j K_0^j \leq 1/2$ shows that, for each $j$,

$$PL_n^j(\theta_n) \leq P_{n,\bar{j}}L_n^j(\theta_n) + \frac{1}{2}\mathcal{G}_{P^\star}(\theta_n) + 135\frac{\delta_n^2}{\eta^j} + 135\left(2C_8C_4 + \frac{1}{\eta^j}\right)\frac{s+\log 4}{n_{\bar{j}}},$$

$$P_{n,\bar{j}}L_n^j(\underline{\theta}) \leq PL_n^j(\underline{\theta}) + \frac{1}{2}\mathcal{G}_{P^\star}(\underline{\theta}) + 135\frac{\delta_n^2}{\eta^j} + 135\left(2C_8C_4 + \frac{1}{\eta^j}\right)\frac{s+\log 4}{n_{\bar{j}}}.$$

Letting $\eta := 1/\max\{2, 270K_0\} = \min_j \eta^j$, the instances of $\eta^j$ on the right-hand sides above can be replaced by $\eta$, at the possible cost of looser inequalities. Moreover, since the above hold for both $j \in [2]$, $\sum_{j=1}^2 \frac{n_{\bar{j}}}{n}P_{n,\bar{j}}L_n^j(\theta) = R_n(\theta)$, and $\sum_{j=1}^2 \frac{n_{\bar{j}}}{n} = 1$, this shows that

$$\sum_{j=1}^2 \frac{n_{\bar{j}}}{n}PL_n^j(\theta_n) \leq R_n(\theta_n) + \frac{1}{2}\mathcal{G}_{P^\star}(\theta_n) + 135\frac{\delta_n^2}{\eta} + 270\left(2C_8C_4 + \frac{1}{\eta}\right)\frac{s+\log 4}{n}$$

$$R_n(\underline{\theta}) \leq \sum_{j=1}^2 \frac{n_{\bar{j}}}{n}PL_n^j(\underline{\theta}) + \frac{1}{2}\mathcal{G}_{P^\star}(\underline{\theta}) + 135\frac{\delta_n^2}{\eta} + 270\left(2C_8C_4 + \frac{1}{\eta}\right)\frac{s+\log 4}{n}.$$

Combining the above inequalities with the fact that $R_n(\theta_n) \leq R_n(\underline{\theta})$ and Lem. S5 yields

$$\frac{9}{10}\mathcal{G}_{P^\star}(\theta_n) \leq \frac{11}{10}\mathcal{G}_{P^\star}(\underline{\theta}) + \frac{1}{2}[\mathcal{G}_{P^\star}(\theta_n) + \mathcal{G}_{P^\star}(\underline{\theta})] + 270\frac{\delta_n^2}{\eta} + 540\left(2C_8C_4 + \frac{1}{\eta}\right)\frac{s+\log 4}{n} + 5B_n^2,$$

where $B_n$ is as defined in that lemma. Subtracting $\mathcal{G}_{P^\star}(\theta_n)/2$ from both sides and then multiplying both sides by $5/2$ yields

$$\mathcal{G}_{P^\star}(\theta_n) \leq 4\mathcal{G}_{P^\star}(\underline{\theta}) + \frac{5}{2}270\frac{\delta_n^2}{\eta} + 1350\left(2C_8C_4 + \frac{1}{\eta}\right)\frac{s+\log 4}{n} + \frac{25}{2}B_n^2.$$

Plugging in the value of $\eta$ and further upper bounding the right-hand side yields

$$\mathcal{G}_{P^\star}(\theta_n) \leq 4\mathcal{G}_{P^\star}(\underline{\theta}) + 1350\delta_n^2(1 \vee 135K_0) + 2700\left[C_8C_4 + (1 \vee 135K_0)\right]\frac{s+\log 4}{n} + 13B_n^2.$$

Choosing $\underline{\theta}$ such that $4\mathcal{G}_{P^\star}(\underline{\theta}) \leq 4\inf_{\theta\in\underline{\Theta}}\mathcal{G}_{P^\star}(\theta) + K_2(2 - \log 4)/n$ yields the claimed bound, (S7).

The double robustness of the final term in the bound follows directly from the fact that $\alpha_n^j = \alpha_P$ implies $\|\alpha_n^j - \alpha_P\|_{L^2(P_X)}^2 = 0$ and $\psi_n^j \in \Psi_P$ implies $d_\Psi^2(\psi_n^j, \Psi_P) = 0$. □

*Proof of Thm. S2.* The proof follows in almost exactly the same way as that of Thm. S1, except that the second half of Lem. S13 is used when establishing (S18), yielding the alternative bound

$$n_{\bar{j}}^{1/2}E\|P_{n,j} - P\|_{\mathcal{L}_{n,\delta}^j} \leq 8KC_8J(2C_8^{1/2}\delta, \ell_{P^\star}(\underline{\Theta}_{2C_8^{1/2}\delta}))\left[1 + \frac{16C_4C_8^2J(2C_8^{1/2}\delta, \ell_{P^\star}(\underline{\Theta}_{2C_8^{1/2}\delta}))}{\delta^2 n_{\bar{j}}^{1/2}}\right].$$

By (S8) and the fact that $n_{\bar{j}} \geq \lfloor n/2 \rfloor$, the right-hand side is no more than $n_{\bar{j}}^{1/2}\delta_n^2$ when $\delta = \delta_n$. The rest of the proof follows in the same way as that of Thm. S1. □

### G.2 PROOF OF MINIMAX LOWER BOUND (THM. 2)

When proving this result, we assume that every conditional and marginal probability measure of interest, such as $P$ or its conditionals, is the pushforward of $\nu = \text{Unif}[0, 1]$ with respect to some transport map, which holds under mild regularity conditions (Lemma 4.22 of Kallenberg, 2021).

*Proof of Thm. 2.* For now fix $T \in \mathcal{T}$, $P^\star \in \mathcal{P}^\star$, and $P \in \mathcal{P}(P^\star)$. Define the function $g_P$ so that, when $V \sim \nu$, $g_P(y^\star_{[n]}, V)$ follows the conditional distribution of $Z_{[n]} \mid (1[A_i = a^\star]Y_i)^n_{i=1} = (1[A_i = a^\star]y^\star_i)^n_{i=1}$ implied by $P^n$. Define $f_1 : [0, 1] \to [0, 1]$ and $f_2 : [0, 1] \to [0, 1]$ so that, when $V \sim \nu$, $f_1(V) \stackrel{d}{=} f_2(V) \stackrel{d}{=} V$ and $f_1(V) \perp\!\!\!\perp f_2(V)$ — see the proof of Lem. 4.21 in Kallenberg (2021) for a construction. By the definitions of $g_P$, $f_1$, and $f_2$, we have that $(g_P(Y^\star_{[n]}, f_1(V)), f_2(V)) \stackrel{d}{=} (Z_{[n]}, V')$ when $(Y^\star_{[n]}, V) \sim P^{\star n} \times \nu$ and $(Z_{[n]}, V') \sim P^n \times \nu$.

We now define an element of $T^\star_{P,T}$ of $\mathcal{T}^\star$, where the subscripts indicate that this choice depends on $P$ and $T$. Specifically, let $T^\star_{P,T}(y^\star_{[n]}, v) := T(g_P(y^\star_{[n]}, f_1(v)), f_2(v))$. By the conclusion of the preceding paragraph,

$$E_{(Z_{[n]}, V) \sim P^n \times \nu} \left[ D\left( P^\star, T(Z_{[n]}, V)_\sharp \Pi \right) \right] = E_{(Y^\star_{[n]}, V) \sim P^{\star n} \times \nu} \left[ D\left( P^\star, T^\star_{P,T}(Y^\star_{[n]}, V)_\sharp \Pi \right) \right].$$

Taking a supremum on both sides over $P \in \mathcal{P}(P^\star)$ followed by one over $P^\star \in \mathcal{P}^\star$ yields

$$\sup_{P^\star \in \mathcal{P}^\star} \sup_{P \in \mathcal{P}(P^\star)} E_{(Z_{[n]}, V) \sim P^n \times \nu} \left[ D\left( P^\star, T(Z_{[n]}, V)_\sharp \Pi \right) \right]$$

$$= \sup_{P^\star \in \mathcal{P}^\star} \sup_{P \in \mathcal{P}(P^\star)} E_{(Y^\star_{[n]}, V) \sim P^{\star n} \times \nu} \left[ D\left( P^\star, T^\star_{P,T}(Y^\star_{[n]}, V)_\sharp \Pi \right) \right]. \tag{S19}$$

Since $T^\star_{P,T} \in \mathcal{T}^\star$, the right-hand side is lower bounded by

$$\inf_{T^\star \in \mathcal{T}^\star} \sup_{P^\star \in \mathcal{P}^\star} \sup_{P \in \mathcal{P}(P^\star)} E_{(Y^\star_{[n]}, V) \sim P^{\star n} \times \nu} \left[ D\left( P^\star, T^\star(Y^\star_{[n]}, V)_\sharp \Pi \right) \right],$$

which is equal to the left-hand side of the display in the theorem statement since the expectation does not depend on $P$. Plugging the above lower bound into the right-hand side of (S19) and then taking an infimum over $T \in \mathcal{T}$ on the left gives the result. $\qquad\square$

## H  BEYOND EMPIRICAL RISK MINIMIZATION: A GENERIC GENERALIZATION BOUND

DoubleGen can be run with learners other than empirical risk minimizers, such as random forests or neural networks trained with stochastic gradient descent. Theoretical guarantees can also be derived in these cases. For instance, a recent work establishes convergence guarantees for stochastic gradient algorithms in the presence of nuisance parameters, showing that stochastic gradient descent can still converge under appropriate conditions (Yu et al., 2025).

Here, we focus on the general case where a black-box learner selects $\theta_n$ based on the data. This bound motivates having that learner make the population counterpart to $R_n$ small, even if done through means other than empirical risk minimization. This population counterpart is $PL_n(\theta)$, where

$$L_n(\theta)(z) := \frac{1}{2} \sum_{j=1}^2 \int \left[ 1(a = a^\star)\alpha^j_n(x) \left\{ \ell_{P^\star}(\theta)(y) - \ell_{P^\star}(\theta)(\psi^j_n(u|x)) \right\} + \ell_{P^\star}(\theta)(\psi^j_n(u|x)) \right] \Pi(du).$$

$$\tag{S20}$$

**Lemma S14** (Generic generalization bound). *If C3–C5, C7, and C8 hold with $\underline{\Theta} = \Theta$, then*

$$\mathcal{G}^{1/2}_{P^\star}(\theta) \leq [0 \vee PL_n(\theta)]^{1/2} + C^{1/2}_7 \max_{j \in [2]} \|\alpha^j_n - \alpha_P\|_{L^2(P_X)} d_\Psi(\psi^j_n, \Psi_P) \text{ for all } \theta \in \Theta.$$

The above shows that the generalization error $\mathcal{G}_{P^\star}(\theta)$ arising from the loss $\ell_{P^\star}$ will be small if the generalization error from the loss $L_n$ is small and the nuisances are estimated well.

A similar black-box generalization bound can be derived from Thm. 1 of Foster & Syrgkanis (2023). That result has the benefit of applying to more general estimation problems than the counterfactual generation problem tackled by DoubleGen, but the disadvantage of taking a more complex form and not yielding a doubly robust remainder term.

*Proof of Lem. S14.* For any $\theta \in \Theta$, adding and subtracting terms and applying Lem. S5 yields

$$\mathcal{G}_{P^\star}(\theta) = \mathcal{G}_{P^\star}(\theta) - PL_n(\theta) + PL_n(\theta)$$
$$\leq C_7^{1/2} \mathcal{G}_{P^\star}^{1/2}(\theta) \max_{j \in [2]} \|\alpha_n^j - \alpha_P\|_{L^2(P_X)} d_\Psi(\psi_n^j, \Psi_P) + 0 \vee PL_n(\theta).$$

When combined with the below Lem. S15, this yields the claimed bound. $\square$

**Lemma S15** (Basic inequality). *If $b, c, d \geq 0$ satisfy $b \leq cb^{1/2} + d$, then $b^{1/2} \leq d^{1/2} + c$.*

*Proof.* The condition on $b, c, d$ can be rewritten as $(b^{1/2} - c/2)^2 \leq d + c^2/4$, which is only possible if $b^{1/2} \leq c/2 + (d + c^2/4)^{1/2}$. By the triangle inequality, $b^{1/2} \leq d^{1/2} + c$. $\square$

# I  SUFFICIENT CONDITIONS FOR MIXED LIPSCHITZ CONDITIONS (C7 AND C16)

For any $\psi \in \Psi$, let $P_{\psi|x}(\cdot) := \psi(\cdot \,|\, x)_\sharp \Pi$. For distributions $\nu_1, \nu_2$ defined on the same probability space, define the chi-squared divergence as

$$\chi^2(\nu_1 \,\|\, \nu_2) = \begin{cases} \int \left(\frac{d\nu_1}{d\nu_2} - 1\right)^2 d\nu_2, & \text{if } \nu_1 \ll \nu_2 \\ \infty, & \text{otherwise.} \end{cases} \tag{S21}$$

**Lemma S16.** *If $d_\Psi^2(\psi, \Psi_P) := \operatorname{ess\,sup}_{X \sim P_X} \chi^2(P_{\psi|X} \,\|\, P_{Y|A=a^\star,X})$ for all $\psi \in \Psi$, then C16 holds with $C_{16} = 1$. If C5 also holds, then C7 holds with $C_7 = C_5$.*

By the definition of $\Psi_P$, $P_{\psi_P|X} = P_{Y|A=a^\star,X}$ $P_X$-a.s. for all $\psi_P \in \Psi_P$. Hence, an equivalent expression for $d_\Psi$ from the above lemma is $d_\Psi^2(\psi, \Psi_P) := \operatorname{ess\,sup}_{X \sim P_X} \chi^2(P_{\psi|X} \,\|\, P_{\psi_P|X})$ for a generic $\psi_P \in \Psi_P$.

*Proof of Lem. S16.* Let $\psi_P \in \Psi_P$, so that $P_{\psi_P|X} = P_{Y|A=a^\star,X}$. For the first claim, we apply the definition of $P_{\psi|x}, P_{\psi_P|x}$ and Cauchy-Schwarz to show that, for $P_X$-almost all $X$,

$$\left\{\int [\ell_{P^\star}(\theta)(\psi(u|x)) - \ell_{P^\star}(\theta)(\psi_P(u|x))] \, \Pi(du)\right\}^2 = \left\{\int \ell_{P^\star}(\theta)(y) \, (P_{\psi|x} - P_{\psi_P|x})(dy)\right\}^2$$
$$\leq \chi^2(P_{\psi|x} \,\|\, P_{\psi_P|x}) \int \ell_{P^\star}^2(\theta)(y) P_{\psi_P|x}(dy).$$

Taking a $P_X$-essential supremum of the first term on the right and then integrating both sides against $P_X$ yields that C16 holds with $d_\Psi^2(\psi, \Psi_P) := \operatorname{ess\,sup}_{X \sim P_X} \chi^2(P_{\psi|X} \,\|\, P_{\psi_P|X})$ and $C_{16} = 1$.

For the second claim, we use the general result that C16 and C5 together imply C7 with $C_7 = C_5 C_{16}$. Since we have already established that C16 holds with $C_{16} = 1$ when $d_\Psi^2(\psi, \Psi_P) := \operatorname{ess\,sup}_{X \sim P_X} \chi^2(P_{\psi|X} \,\|\, P_{\psi_P|X})$, C5 implies C7 holds with $C_7 = C_5$ when $d_\Psi$ takes this form. $\square$

# J  TOTAL VARIATION GUARANTEE FOR DOUBLEGEN DIFFUSION

## J.1  SCORE NETWORK CLASS

We establish a rate of convergence for DoubleGen diffusion modeling (Example 2) when an empirical risk minimizer is evaluated over a particular neural network class $\Theta$. This class is selected to

be large enough to make the approximation error small. In the following lemma, $\| \cdot \|_0$ denotes the sparsity 'norm' that counts the number of nonzero entries of a Euclidean vector and $w :=$ $(d+1, W, W, \ldots, W, 1) \in \mathbb{R}^{D+1}$. Here and throughout this appendix we let $\mathcal{P}_1^\star$ denote the collection of all $P^\star$ satisfying C10–C12 for fixed constants $C_{10}$, $C_{11}$, and $C_{12}$. Unless otherwise specified, we use '$\lesssim$' to denote inequalities up to constants that do not depend on $P^\star \in \mathcal{P}_1^\star$ or $n$, including through $n$-dependent quantities like $\underline{t}$ or $\underline{\Theta}$.

**Lemma S17** (Score network for diffusion model, Oko et al., 2023). *Suppose C9–C13. There exists a depth $D \lesssim \log^4 n$, width $W \lesssim n \log^6 n$, sparsity level $S \lesssim n^{d/(2s+d)} \log^8 n$, weight bound $B_{\mathrm{wgt}}$ satisfying $\log B_{\mathrm{wgt}} \lesssim \log^4 n$, and output bound $B_{\mathrm{out}} \lesssim \log^{1/2} n$ such that, for all $n$ large enough,*

$$\underline{\Theta} := \{\theta \in (\mathbb{R}^d)^{\mathbb{R}^d \times [\underline{t}, \bar{t}]} : \|\theta(\cdot, t)\|_\infty \leq B_{\mathrm{out}}/\sigma_t \; \forall t \in [\underline{t}, \bar{t}]\} \cap \quad \text{(bounded $d$-dimensional output)}$$

$$\Big\{ (M_D \, \mathrm{ReLU}(\cdot) + v_D) \circ \ldots \circ (M_2 \, \mathrm{ReLU}(\cdot) + v_2) \circ (M_1(\cdot) + v_1) : \quad$$

$$\text{(sparse ReLU network)}$$

$$M_j \in [-B_{\mathrm{wgt}}, B_{\mathrm{wgt}}]^{w_{j+1} \times w_j}, v_j \in [-B_{\mathrm{wgt}}, B_{\mathrm{wgt}}]^{w_{j+1}}, \sum_{j=1}^{D} (\|M_j\|_0 + \|v_j\|_0) \leq S \Big\}$$

*satisfies the approximation error bound $\inf_{\theta \in \underline{\Theta}} \mathcal{G}_{P^\star}(\theta) \lesssim n^{-2s/(2s+d)} \log^2 n$.*

The above lemma above is a restatement of Thm. 3.1 from Oko et al. (2023), and so the proof is omitted. We require the condition that $n$ is sufficiently large so that our C12 implies Assumption 2.6 from Oko et al. (2023), which is a slightly weaker version of our boundary smoothness condition that allows $\varepsilon$ to decay with $n$. Our results would also hold under their weaker condition—we use the stronger condition because it is simpler to state.

### J.2 VERIFYING THE CONDITIONS OF THM. S1

Under the conditions of Thm. 3, we show that DoubleGen diffusion satisfies the conditions of Thm. S1. We suppose throughout that C8 holds, since this standard causal condition (Hernán & Robins, 2024) is directly assumed in Thm. 3. We further assume that $n$ is large enough so the conclusion of Lem. S17 holds.

**Condition C3:** Applying Thm. 4.1.15 in (Durrett, 2019) coordinatewise shows $\theta_{P^\star} \in \mathrm{argmin}_{\theta \in \Theta} E_{P^\star}[\ell(\theta, Y)]$.

**Condition C4:** This condition follows by the following lemma.

**Lemma S18** (Bounds on denoising score matching loss, Oko et al., 2023). *Suppose C9, C11, and C13 and $n \geq 2$. For $\underline{\Theta}$ chosen as in Lem. S17, $\sup_{\theta \in \underline{\Theta}, y \in \mathcal{Y}} |\ell(\theta, y)| \lesssim \log^2 n$ and $\sup_{y \in \mathcal{Y}} |\ell(\theta_{P^\star}, y)| \lesssim \log^2 n$. As a consequence, C4 holds with $C_4 \lesssim \log^2 n$.*

The proof of the above relies on the following helper lemma.

**Lemma S19.** *If C13, then there exists $C > 0$ such that $\int_{\underline{t}}^{\bar{t}} \sigma_t^{-2} dt \leq C \log n$ for all $n \geq 2$ and $t \geq \underline{t}$.*

*Proof of Lem. S19.* For all $x \geq 0$, $1 - \exp(-x) \geq x/(1+x)$, and so $1/[1 + \exp(-x)] \leq 1 + 1/x$. Plugging in $2 \int_0^t \beta_v dv$ for $x$ shows $\sigma_t^{-2} \leq 1 + 1/(2 \int_0^t \beta_v dv) \leq 1 + 1/(2\underline{\beta}t)$. Hence,

$$\int_{\underline{t}}^{\bar{t}} \sigma_t^{-2} dt \leq \int_{\underline{t}}^{\bar{t}} [1 + 1/(2\underline{\beta}t)] dt = \bar{t} - \underline{t} + \frac{1}{2\underline{\beta}} \int_{\underline{t}}^{\bar{t}} \frac{1}{t} dt = \bar{t} - \underline{t} + \frac{1}{2\underline{\beta}} \left( \log \bar{t} - \log \underline{t} \right).$$

By C13 and the fact that $n \geq 2$, the right-hand side upper bounds by a constant times $\log n$. $\square$

*Proof of Lem. S18.* Lem. C.1 from Oko et al. (2023) shows $\sup_{\theta \in \underline{\Theta}, y \in \mathcal{Y}} |\ell(\theta, y)| \lesssim \log^2 n$ provided $\underline{\Theta}$ satisfies the bound $\sup_{\theta \in \underline{\Theta}, t \in [\underline{t}, \bar{t}], y_t \in \mathbb{R}^d} |\sigma_t \theta(y_t, t)| \lesssim \log^{1/2} n$, which it does by the choice of $B_{\mathrm{out}}$ Lem. S17.

It remains to show that $\sup_{y_0 \in \mathcal{Y}} |\ell(\theta_{P^\star}, y_0)| \lesssim \log^2 n$. This fact was used in the proof of Thm. C.4 in Oko et al. (2023), though the details for deriving it were omitted. We give them here. We start by following similar arguments to those used in the proof of Lem. C.1 in Oko et al. (2023), yielding the following for each $y_0 \in [-1, 1]^d$:

$$
\begin{aligned}
\frac{1}{2} \ell(\theta_{P^\star}, y_0) &= \frac{1}{2} \int_{\underline{t}}^{\overline{t}} \int_{\mathbb{R}^d} \left\| \frac{\mu_t y_0 - y_t}{\sigma_t^2} - \theta_{P^\star}(y_t, t) \right\|^2 p(y_t \mid y_0) dy_t \, dt \\
&\leq \int_{\underline{t}}^{\overline{t}} \int_{\mathbb{R}^d} \left\| \frac{\mu_t y_0 - y_t}{\sigma_t^2} \right\|^2 p(y_t \mid y_0) dy_t \, dt + \int_{\underline{t}}^{\overline{t}} \int_{\mathbb{R}^d} \|\theta_{P^\star}(y_t, t)\|^2 p(y_t \mid y_0) dy_t \, dt \\
&\lesssim \log n + \int_{\underline{t}}^{\overline{t}} \int_{\mathbb{R}^d} \|\theta_{P^\star}(y_t, t)\|^2 p(y_t \mid y_0) dy_t \, dt,
\end{aligned}
\tag{S22}
$$

where the suppressed multiplicative constant on the right-hand side of '$\lesssim$' does not depend on $y_0$ and the bound on the first term follows from $Y_t \mid Y_0 \sim N(\mu_t Y_0, \sigma_t^2 I_d)$ and Lem. S19. By the bound on the score $\theta_{P^\star}$ given in Eq. 17 of Lem. A.3 in Oko et al. (2023), the latter term above satisfies

$$
\begin{aligned}
\int_{\underline{t}}^{\overline{t}} \int_{\mathbb{R}^d} \|\theta_{P^\star}(y_t, t)\|^2 p(y_t \mid y_0) dy_t \, dt &\lesssim \int_{\underline{t}}^{\overline{t}} \sigma_t^{-2} \int_{\mathbb{R}^d} \left( 1 \vee \sigma_t^{-2} (\|y_t\|_\infty - \mu_t)_+^2 \right) p(y_t \mid y_0) dy_t \, dt \\
&\leq \int_{\underline{t}}^{\overline{t}} \sigma_t^{-2} \int_{\mathbb{R}^d} \left( 1 + \sigma_t^{-2} (\|y_t\|_\infty - \mu_t)_+^2 \right) p(y_t \mid y_0) dy_t \, dt \\
&\lesssim \log^2 n + \int_{\underline{t}}^{\overline{t}} \sigma_t^{-4} \int_{\mathbb{R}^d} (\|y_t\|_\infty - \mu_t)_+^2 p(y_t \mid y_0) dy_t \, dt,
\end{aligned}
\tag{S23}
$$

where the final inequality used Lem. S19. For the latter term, we use that $(\|b\|_\infty - c)_+ = \|(b - c)_+\|_\infty \vee \|(b + c)_-\|_\infty$ for all $b \in \mathbb{R}^d$ and $c \in \mathbb{R}$, with the positive and negative part functions applied elementwise. Combining this with the fact that $y_0 \in [-1, 1]^d$ yields

$$
\begin{aligned}
\int_{\mathbb{R}^d} (\|y_t\|_\infty - \mu_t)_+^2 p(y_t \mid y_0) &\leq \sup_{\tilde{y}_0 \in [-1,1]^d} \int_{\mathbb{R}^d} \|(y_t - \mu_t)_+\|_\infty^2 p(y_t \mid Y_0 = \tilde{y}_0) \, dy_t \\
&\quad + \sup_{\tilde{y}_0 \in [-1,1]^d} \int_{\mathbb{R}^d} \|(y_t + \mu_t)_-\|_\infty^2 p(y_t \mid Y_0 = \tilde{y}_0) \, dy_t.
\end{aligned}
$$

Since $Y_t \mid Y_0 \sim N(\mu_t Y_0, \sigma_t^2 I_d)$, it is straightforward to verify that the first supremum is achieved at $\tilde{y}_0 = (1, 1, \ldots, 1)$ and the second at $\tilde{y}_0 = (-1, -1, \ldots, -1)$. Hence, for $V \sim N(0_d, I_d)$,

$$
\int_{\mathbb{R}^d} (\|y_t\|_\infty - \mu_t)_+^2 p(y_t \mid y_0) \leq E\left[\|(\sigma_t V)_+\|_\infty^2\right] + E\left[\|(\sigma_t V)_-\|_\infty^2\right] = \sigma_t^2 E[\|V\|_\infty^2].
$$

By properties of Gaussian random variables, $E[\|V\|_\infty^2] \leq C \log d$, with $C$ a universal constant. Hence, the second term on the right-hand side of (S23) is upper bounded by $C \log(d) \int_{\underline{t}}^{\overline{t}} \sigma_t^{-2} dt \lesssim \log n$ (Lem. S19). Plugging this into (S23) and then returning to (S22) shows that $\sup_{y_0 \in \mathcal{Y}} |\ell(\theta_{P^\star}, y_0)| \lesssim \log^2 n$.

Since $\ell_{P^\star}(\theta)(\cdot) := \ell(\theta, \cdot) - \ell(\theta_{P^\star}, \cdot)$ and we have shown that $\sup_{\theta \in \underline{\Theta}, y \in \mathcal{Y}} |\ell(\theta, y)| \vee \sup_{y \in \mathcal{Y}} |\ell(\theta_{P^\star}, y)| \lesssim \log^2 n$, the triangle inequality yields C4 with $C_4 \lesssim \log^2 n$. $\qquad \square$

**Condition C5:** For any $\theta \in \underline{\Theta}$ and $y \in \mathcal{Y}$, Cauchy-Schwarz yields that

$$
\begin{aligned}
\ell_{P^\star}^2(\theta)(y) &= \left( \int_{\underline{t}}^{\overline{t}} E\left[ \left\{ \frac{2(\mu_t y - Y_t)}{\sigma_t^2} - \theta(Y_t, t) - \theta_{P^\star}(Y_t, t) \right\}^\top \{\theta_{P^\star}(Y_t, t) - \theta(Y_t, t)\} \,\middle|\, Y_0 = y \right] dt \right)^2 \\
&\leq \left( \int_{\underline{t}}^{\overline{t}} E\left[ \left\| \frac{2(\mu_t y - Y_t)}{\sigma_t^2} - \theta(Y_t, t) - \theta_{P^\star}(Y_t, t) \right\|^2 \,\middle|\, Y_0 = y \right] dt \right) \\
&\quad \times \int_{\underline{t}}^{\overline{t}} E\left[ \|\theta_{P^\star}(Y_t, t) - \theta(Y_t, t)\|^2 \,\middle|\, Y_0 = y \right] dt.
\end{aligned}
$$

The leading integral above is upper bounded by $2[\ell(\theta, y) + \ell(\theta_{P^\star}, y)]$. Using Lem. S18 to bound this quantity uniformly over $(\theta, y) \in \underline{\Theta} \times \mathcal{Y}$, plugging this bound into the above, and finally integrating both sides against $P^\star$ yields that $\|\ell_{P^\star}(\theta)\|_{L^2(P^\star)}^2 \lesssim \log^2(n)\mathcal{G}_{P^\star}(\theta)$, and so C5 holds with $C_5 \lesssim \log^2 n$.

**Condition C6:** We will provide a finite bound on $J(\delta, \ell_{P^\star}(\underline{\Theta}))$ when $\underline{\Theta}$ is the neural network class from Lem. S17. The entropy of $\ell(\underline{\Theta})$ depends on $n$ through the architecture of the neural nets in $\underline{\Theta}$ and the truncation times $\underline{t}$ and $\overline{t}$ used in the loss.

**Lemma S20** (Entropy integral bound for score network loss class). *If C9–C13 and $\underline{\Theta}$ is chosen as in Lem. S17, then there exists $C_6 < \infty$ such that, for all $n \geq 2$, $P^\star \in \mathcal{P}$, and $\delta \leq c$,*

$$J(\delta, \ell_{P^\star}(\underline{\Theta})) \leq C_6 \log^8(n) n^{d/(4s+2d)} \delta \sqrt{\log \delta^{-1}}.$$

Since the above entropy integral bound is finite, C6 holds. Our proof of Thm. 3 will make more precise use of this bound to obtain a valid choice of $\delta_n$.

We provide the proof of Lem. S20 at the end of this appendix. That proof will rely on a bound on the metric entropy of $\ell(\underline{\Theta})$, which we state and prove now. In what follows, $\|\cdot\|_\infty$ denotes the usual norm on $L^\infty([-1, 1]^d)$.

**Lemma S21** (Metric entropy bound for score network loss class). *If C9–C13 and $n \geq 2$, then there exist constants $c \in (0, 1)$ and $C \in (0, \infty)$ that do not depend on $n$ such that, for all $\epsilon \leq c$,*

$$\log N(\epsilon, \ell(\underline{\Theta}), \|\cdot\|_\infty) \leq Cn^{d/(2s+d)}[\log^{16}(n) + \log^{12}(n) \log(1/\epsilon)].$$

*Proof of Lem. S21.* We imitate the proof of Lem. C.2 from Oko et al. (2023), but modify it slightly so that it gives a covering number bound even when $\epsilon$ is arbitrarily small.

We will show there exists a constant $C' < \infty$ such that, for all $\varepsilon \leq 1/2$ and a constant $c_3$ that we will specify later,

$$\log N(c_3 \varepsilon \log^{7/2} n, \ell(\underline{\Theta}), \|\cdot\|_\infty) \leq C' n^{d/(2s+d)}[\log^{16}(n) + \log^{12}(n) \log(1/\varepsilon)]. \tag{S24}$$

Applying the change of variables $\epsilon := c_3 \varepsilon \log^{7/2} n$ then gives the desired result with $c := c_3 \log^{7/2}(n)/2$ and an appropriately specified $C$ that depends only on $c, C'$.

Fix $\varepsilon \leq 1/2$. Below we let $\tilde{Y}_t := (\mu_t Y_0 - Y_t)/\sigma_t^2$. For any $\theta_1, \theta_2 \in \underline{\Theta}$ and $y \in \mathcal{Y}$, we have that

$$|\ell(\theta_1, y) - \ell(\theta_2, y)| \tag{S25}$$

$$\leq \left| \int_{\underline{t}}^{\overline{t}} E\left[ \left( \left\| \tilde{Y}_t - \theta_1(Y_t, t) \right\|_2^2 - \left\| \tilde{Y}_t - \theta_2(Y_t, t) \right\|_2^2 \right) 1\{\|\sigma_t \tilde{Y}_t\|_\infty \geq \log^{1/2} \varepsilon^{-1}\} \middle| Y_0 = y \right] dt \right|$$

$$+ \left| \int_{\underline{t}}^{\overline{t}} E\left[ \left( \left\| \tilde{Y}_t - \theta_1(Y_t, t) \right\|_2^2 - \left\| \tilde{Y}_t - \theta_2(Y_t, t) \right\|_2^2 \right) 1\{\|\sigma_t \tilde{Y}_t\|_\infty < \log^{1/2} \varepsilon^{-1}\} \middle| Y_0 = y \right] dt \right|.$$

The bounds on $\theta_1, \theta_2$ from Lem. S17, Cauchy-Schwarz, and Lem. F.12 from Oko et al. (2023) show that the first term is no more than a constant $c_1$ that may depend on $d$ times $\varepsilon \log^2 n$, where we used that $\int_{\underline{t}}^{\overline{t}} \sigma_t^{-2} dt \lesssim \log n$ by Lem. S19. For the second term, we use that, for any $\tilde{y}_t \in [-\sigma_t^{-1} \log^{1/2} \varepsilon^{-1}, \sigma_t^{-1} \log^{1/2} \varepsilon^{-1}]^d$,

$$\left| \|\tilde{y}_t - \theta_1(y_t, t)\|_2^2 - \|\tilde{y}_t - \theta_2(y_t, t)\|_2^2 \right| = |\langle 2\tilde{y}_t - (\theta_1 + \theta_2)(y_t, t), (\theta_1 - \theta_2)(y_t, t)\rangle|$$

$$\leq \frac{2d^{1/2}}{\sigma_t} \left( \log^{1/2} \varepsilon^{-1} + B_{\text{out}} \right) \|(\theta_1 - \theta_2)(y_t, t)\|_2.$$

Plugging this into the second term in (S25) shows that term is no more than a constant $c_2$ times

$$2d^{1/2} \log^2(n)(\sqrt{\log \varepsilon^{-1}} + B_{\text{out}}) \int_{\underline{t}}^{\overline{t}} E\left[ \|(\theta_1 - \theta_2)(Y_t, t)\|_2 I\{\|\sigma_t \tilde{Y}_t\|_\infty \leq \sqrt{\log \varepsilon^{-1}}\} \middle| Y_0 = y \right] dt.$$

The expectation above is upper bounded by $\sup_{y_t : \|\mu_t y - y_t\|_\infty \leq \sigma_t \log^{1/2} \varepsilon^{-1}} \|(\theta_1 - \theta_2)(y_t, t)\|_2$, which by the bounds on $y$ (C9) is further bounded by $\sup_{y_t : \|y_t\|_\infty \leq \mu_t + \sigma_t \log^{1/2} \varepsilon^{-1}} \|(\theta_1 - \theta_2)(y_t, t)\|_2 =$

$\|\|\|(\theta_1 - \theta_2)(\cdot, t)\|_2\|_{L^\infty([-\mu_t - \sigma_t \log^{1/2} \varepsilon^{-1}, \mu_t + \sigma_t \log^{1/2} \varepsilon^{-1}]^d)}$. Hence, the second term in (S25) is no more than $2c_2 d^{1/2} \log^2(n) \left( \log^{1/2} \varepsilon^{-1} + B_{\text{out}} \right)$ times

$$\int_{\underline{t}}^{\overline{t}} \|\|\|(\theta_1 - \theta_2)(\cdot, t)\|_2\|_{L^\infty([-\mu_t - \sigma_t \log^{1/2} \varepsilon^{-1}, \mu_t + \sigma_t \log^{1/2} \varepsilon^{-1}]^d)} dt.$$

Combining our bounds of the two terms in (S25) shows that, if

$$\|\|\|(\theta_1 - \theta_2)(\cdot, t)\|_2\|_{L^\infty([-\mu_t - \sigma_t \log^{1/2} \varepsilon^{-1}, \mu_t + \sigma_t \log^{1/2} \varepsilon^{-1}]^d)} \leq \varepsilon / \log^{1/2} \varepsilon^{-1} \text{ for all } t \in [\underline{t}, \overline{t}], \tag{S26}$$

then, for all $y$,

$$|\ell(\theta_1, y) - \ell(\theta_2, y)| \leq \left[ c_1 + 2c_2 d^{1/2}(\overline{t} - \underline{t}) \left( 1 + \log^{-1/2} \varepsilon^{-1} B_{\text{out}} \right) \right] \varepsilon \log^2 n.$$

By the bounds on $\overline{t}$ and $B_{\text{out}}$ from C13 and Lem. S17 and the fact that $\varepsilon \leq 1/2$, there exists a constant $c_3$ that does not depend on $n$ such that, for all $\theta_1, \theta_2$ satisfying (S26),

$$|\ell(\theta_1, y) - \ell(\theta_2, y)| \leq c_3 \varepsilon \log^{7/2} n \text{ for all } y \in [-1, 1]^d. \tag{S27}$$

The bound on $\overline{t}$ and the fact that $\mu_t, \sigma_t \in [0, 1]$ further imply there exists a constant $c_4$ such that, for $b(n, \varepsilon) := c_4 (\log^{1/2} \varepsilon^{-1} \vee \log n)$,

$$[\underline{t}, \overline{t}] \cup [-\mu_t - \sigma_t \log^{1/2} \varepsilon^{-1}, \mu_t + \sigma_t \log^{1/2} \varepsilon^{-1}] \subseteq [-b(n, \varepsilon), b(n, \varepsilon)],$$

and so a sufficient condition for (S26) is that

$$\|\|\|(\theta_1 - \theta_2)(\cdot)\|_2\|_{L^\infty([-b(n,\varepsilon), b(n,\varepsilon)]^{d+1})} \leq \varepsilon / \log^{1/2} \varepsilon^{-1}.$$

Hence, the above implies (S27), which yields that

$$\log N(c_3 \varepsilon \log^{7/2} n, \ell(\underline{\Theta}), L^\infty([-1, 1]^d)) \leq \log N(\varepsilon / \log^{1/2} \varepsilon^{-1}, \underline{\Theta}, \|\| \cdot \|_2\|_{L^\infty([-b(n,\varepsilon), b(n,\varepsilon)]^{d+1})}).$$

We apply Eq. 57 from Lem. C.2 of Oko et al. (2023) (see also Suzuki (2018)) and the architecture of $\underline{\Theta}$ from Lem. S17 to bound the right-hand side above by a constant times

$$n^{d/(2s+d)} \log^{12}(n) \left[ \log^4 n + \log \left( \varepsilon^{-1} \log^{1/2}(\varepsilon^{-1}) \log(n) n b(n, \varepsilon) \right) \right].$$

This is in turn upper bounded by a constant $C'$ times $n^{d/(2s+d)} [\log^{16}(n) + \log^{12}(n) \log(1/\varepsilon)]$. Hence, (S24) holds, completing the proof. □

*Proof of Lem. S20.* Let $\| \cdot \|_\infty$ denote the supremum norm on $\mathcal{Y}$. Combining the definition of the entropy integral in (S1) with the fact that $\| \cdot \|_{L^2(Q)} \leq \| \cdot \|_\infty$ for any distribution $Q$ on $\mathcal{Y}$ shows that

$$J(\delta, \ell_{P^\star}(\underline{\Theta})) \leq \int_0^\delta \sqrt{1 + \log N(\epsilon, \ell_{P^\star}(\underline{\Theta}), \| \cdot \|_\infty)} \, d\epsilon.$$

Since the uncentered loss class $\ell(\underline{\Theta}) := \{\ell(\theta, \cdot) : \theta \in \underline{\Theta}\}$ equals the centered loss class $\ell_{P^\star}(\underline{\Theta}) := \{\ell(\theta, \cdot) - \ell(\theta_{P^\star}, \cdot) : \theta \in \underline{\Theta}\}$ up to the translation $\ell(\theta_{P^\star}, \cdot)$, the two classes have the same sup-norm covering number. Moreover, by Lem. S21, there exists $C > 0$ such that $\log N(\epsilon, \ell(\underline{\Theta}), \| \cdot \|_\infty) \leq C n^{d/(2s+d)} [\log^{16} n + \log^{12}(n) \log \epsilon^{-1}]$ for all $\epsilon \in (0, c]$. Applying this fact and then the triangle inequality shows that, whenever $\delta \leq c$,

$$J(\delta, \ell_{P^\star}(\underline{\Theta})) \leq \int_0^\delta \sqrt{1 + C n^{d/(2s+d)} [\log^{16} n + \log^{12}(n) \log \epsilon^{-1}]} \, d\epsilon$$

$$\leq \frac{\delta}{2} (1 + C^{1/2} n^{d/(4s+2d)} \log^8 n) + C^{1/2} n^{d/(4s+2d)} \log^6(n) \int_0^\delta \sqrt{\log \epsilon^{-1}} \, d\epsilon. \tag{S28}$$

For the remaining integral, we apply the change of variables $\varepsilon = \epsilon/\delta$ and the triangle inequality to find that, for all $\delta \leq c < 1$,

$$\int_0^\delta \sqrt{\log \epsilon^{-1}} \, d\epsilon = \delta \int_0^1 \sqrt{\log \delta^{-1} + \log \varepsilon^{-1}} \, d\varepsilon \leq \delta \log^{1/2} \delta^{-1} + \delta \int_0^1 \log^{1/2} \varepsilon^{-1} \, d\varepsilon.$$

The remaining integral equals $\pi^{1/2}/2$. Hence, there exists a constant $k_1$ such that, for all $\delta \leq c$, $\int_0^\delta \sqrt{\log \epsilon^{-1}} \, d\epsilon \leq k_1 \delta \log^{1/2} \delta^{-1}$. Combining this with (S28) then shows that there exists a constant $C_6$ such that, for all $\delta \leq c$, $J(\delta, \ell_{P^\star}(\underline{\Theta})) \leq \log^8(n) n^{d/(4s+2d)} \delta \sqrt{\log \delta^{-1}}$, as desired. □

**Condition C7:** Since C5 holds, Lem. S16 shows this condition holds with $C_7 = C_5$ and $d_\Psi^2(\psi, \Psi_P) := \text{ess sup}_{X \sim P_X} \chi^2(P_{\psi|X} \| P_{\psi_P|X})$.

### J.3 Proof of total variation bound (Thm. 3)

*Proof of Thm. 3.* Throughout this proof, we write '$\lesssim$' to denote inequality up to a multiplicative constant that does not depend on the value of $n \geq 2$ or $P^\star \in \mathcal{P}_1^\star$. We suppose throughout that $n$ is large enough so the conclusion of Lem. S17 holds. In Appx. J.2, we show that C8 and C9–C13 imply that the conditions of Thm. S1 hold with $d_\Psi$ as in Lem. S16 and, for all $\delta$ small enough,

$$J(\delta, \ell_{P^\star}(\underline{\Theta})) \lesssim \log^8(n) n^{d/(4s+2d)} \delta \sqrt{\log \delta^{-1}}.$$

Plugging this into (S6) shows that $\delta_n \lesssim \log^{17/2}(n) n^{-s/(2s+d)}$. Hence, Thm. S1 shows that, w.p. at least $1 - e^{-r}$,

$$\mathcal{G}_{P^\star}(\theta_n) \lesssim \inf_{\theta \in \underline{\Theta}} \mathcal{G}_{P^\star}(\theta) + \log^{17}(n) \, n^{-2s/(2s+d)} + r/n + \max_{j \in [2]} \|\alpha_n^j - \alpha_P\|_{L^2(P_X)}^2 d_\Psi^2(\psi_n^j, \Psi_P).$$

Here, we used that the constant $K_0$ from Thm. S1 is a.s. bounded by a constant not depending on $n$ since, by assumption, C8 holds and $\max_j d_\Psi(\psi_n^j, \Psi_P)$ is a.s. uniformly bounded. By Lem. S17, the leading approximation term above is upper bounded by $n^{-2s/(2s+d)} \log^2 n$, uniformly over $P^\star \in \mathcal{P}_1^\star$, and so, w.p. at least $1 - e^{-r}$,

$$\mathcal{G}_{P^\star}(\theta_n) \lesssim \log^{17}(n) \, n^{-2s/(2s+d)} + r/n + \max_{j \in [2]} \|\alpha_n^j - \alpha_P\|_{L^2(P_X)}^2 d_\Psi^2(\psi_n^j, \Psi_P).$$

By Prop. 1—which is applicable since the above implies C2 and C1 holds by the arguments in Appx. A—the above implies that the following holds w.p. at least $1 - e^{-r}$:

$$\mathrm{TV}(P^\star, \tau(\theta_n)_\sharp \Pi) \lesssim \log^{17/2}(n) \, n^{-s/(2s+d)} + \sqrt{r/n} + \max_{j \in [2]} \|\alpha_n^j - \alpha_P\|_{L^2(P_X)} d_\Psi(\psi_n^j, \Psi_P) + \epsilon,$$

with $\epsilon = O(n^{-s/(2s+d)})$ as defined in the study of Example 2 in Appx. A. Since $\epsilon$ decays faster than the first term on the right, it can be absorbed into the leading constant of the first term above. $\square$

## K Wasserstein guarantee for DoubleGen flow matching

### K.1 Statement of guarantee

We now present a Wasserstein guarantee for DoubleGen flow matching (Example 1) for an empirical risk minimizer $\theta_n$ over some class $\underline{\Theta}$. This guarantee will rely on the following conditions.

**C17)** *Bounded support:* $\mathrm{support}(P^\star) \subseteq [-1, 1]^d$.

**C18)** *Uniformly bounded candidate vector fields:* $\sup_{\theta \in \underline{\Theta}} \|\theta\|_{L^\infty(\mathbb{R}^d \times [0,1]; \|\cdot\|_2)} < \infty$, where $\|\theta\|_{L^\infty(\mathbb{R}^d \times [0,1]; \|\cdot\|_2)} := \sup_{(y,t) \in \mathbb{R}^d \times [0,1]} \|\theta(y, t)\|_2$.

**C19)** *Set of candidate vector fields has finite sup-norm entropy integral:*

$$\int_0^\delta \sqrt{1 + \log N(\epsilon, \underline{\Theta}, \|\cdot\|_{L^\infty(\mathbb{R}^d \times [0,1]; \|\cdot\|_2)})} \, d\epsilon < \infty.$$

The latter two conditions are satisfied by appropriately specified neural network classes (Suzuki, 2018; Fukumizu et al., 2024).

When stating the guarantee, we directly suppose C1 holds. This is reasonable since, as noted in Appx. A, Fukumizu et al. (2024) gives conditions under which it holds when the vector field is sufficiently smooth. We also directly assume C8, which is standard in causal inference—see Chap. 3.3 of Hernán & Robins (2024) for a discussion.

**Corollary S1** (Wasserstein guarantee for DoubleGen flow matching). *Suppose there exists an empirical risk minimizer $\theta_n \in \arg\min_{\theta \in \underline{\Theta}} R_n(\theta)$ and C1 holds with $D = W_2$, $b = 1/2$, $\epsilon = 0$ and $C_1$ a finite constant. Further suppose C8, C17, and C18. Under these conditions, the conditions of Prop. 1 and Thm. S1 hold for appropriately specified constants. Hence, letting $\delta_n$ be as defined in (S6), the following holds w.p. at least $1 - e^{-s}$:*

$$W_2(P^\star, \phi_{n\sharp}\Pi) \lesssim \inf_{\theta \in \underline{\Theta}} \mathcal{G}_{P^\star}^{1/2}(\theta) + \delta_n + (s^{1/2} + 1)/n^{1/2} + \max_{j \in [2]} \|\alpha_n^j - \alpha_P\|_{L^2(P_X)} d_\Psi(\psi_n^j, \Psi_P),$$

$$\tag{S29}$$

*where $d_\Psi$ is as in Lem. S16 and '$\lesssim$' denotes inequality up to a multiplicative constant that does not depend on $n$ or $s$.*

*Proof of Cor. S1.* In Appx. K.2, we show that C8, C17, and C18 imply that the conditions of Thm. S1 hold. This, in turn, implies that C2 holds with $r$ equal to the right-hand side of (S7). Hence, the conditions of Prop. 1 hold, and combining the result with the triangle inequality yields (S29).  $\square$

For ease of presentation we have suppressed the explicit constants on the right-hand of (S29), but it is straightforward to compute them using the arguments in Appx. K.2. The argument that establishes C6 in Appx. K.2 relates the uniform entropy integral of $\ell_{P^\star}(\underline{\Theta})$ to the entropy integral of $\underline{\Theta}$ from C19. This, in turn, makes it possible to compute the value of $\delta_n$ by directly considering the size of $\underline{\Theta}$.

In future work, it would be interesting to establish that the bound in (S29) is minimax rate optimal. One possible approach would involve adapting the arguments in Fukumizu et al. (2024).

### K.2 VERIFYING THE CONDITIONS OF THM. S1

Under the conditions of Cor. S1, we show that DoubleGen flow matching (Example 1) satisfies the conditions of Thm. S1.

**Condition C3:** Applying Thm. 4.1.15 in (Durrett, 2019) coordinatewise shows $\theta_{P^\star} \in \arg\min_{\theta\in\Theta} E_{P^\star}[\ell(\theta, Y)]$.

**Condition C4:** Since the functions in $\underline{\Theta}$ are uniformly bounded (C18), the facts that $U \sim \Pi = N(0_d, I_d)$ and support$(P^\star) \subseteq [-1,1]^d$ (C17) imply $\sup_{\theta\in\underline{\Theta}} \|\ell(\theta, \cdot)\|_{L^\infty(P^\star)} < \infty$. The function $\ell(\theta_{P^\star}, \cdot)$ is also $P^\star$-a.s. bounded, since

$$\ell(\theta_{P^\star}, y) = \min_{\theta:\mathbb{R}^d\to\mathbb{R}} \int_0^1 E_\Pi[\|y - U - \theta([1-t]U + ty, t)\|_2^2]dt \leq \int_0^1 E_\Pi[\|y - U\|_2^2]dt.$$

By the triangle inequality, the two established bounds imply that $\ell_{P^\star}(\theta)(y) := \ell(\theta, y) - \ell(\theta_{P^\star}, y)$ satisfies C4.

**Condition C5:** For any $\theta \in \underline{\Theta}$ and $y \in [-1,1]^d$, Cauchy-Schwarz yields that

$$\ell_{P^\star}^2(\theta)(y) = \left( \int_0^1 E_\Pi[\{2(y - U) - (\theta + \theta_{P^\star})([1-t]U + ty, t)\}^\top (\theta_{P^\star} - \theta)([1-t]U + ty, t)]dt \right)^2$$

$$\leq \left( \int_0^1 E_\Pi\left[ \|2(y - U) - (\theta + \theta_{P^\star})([1-t]U + ty, t)\|_2^2 \right] dt \right)$$

$$\times \left( \int_0^1 E_\Pi\left[ \|(\theta_{P^\star} - \theta)([1-t]U + ty, t)\|_2^2 \right] dt \right).$$

The leading integral above is upper bounded by $2[\ell(\theta, y) + \ell(\theta_{P^\star}, y)]$. By the arguments we used to establish C4, this term is bounded uniformly over $(\theta, y)$. Integrating both sides above against $P^\star$ and using that $\mathcal{G}_{P^\star}(\theta) = \int_0^1 E_{P^\star\times\Pi}\left[ \|(\theta_{P^\star} - \theta)([1-t]U + tY^\star, t)\|^2 \right] dt$ shows that $\|\ell_{P^\star}(\theta)\|_{L^2(P^\star)}^2 \leq C_5 \mathcal{G}_{P^\star}(\theta)$ for some constant $C_5$ that does not depend on $\theta \in \underline{\Theta}$.

**Condition C6:** We will show that $\ell_{P^\star}$ is Lipschitz and then use properties of covering numbers to relate the uniform entropy integral of $\ell_{P^\star}(\underline{\Theta})$ to one measuring the size of $\underline{\Theta}$, which is finite by C19.

To establish the Lipschitz property, we use that, for any $\theta_1, \theta_2 \in \underline{\Theta}$ and $y$,

$$\ell_{P^\star}(\theta_1)(y) - \ell_{P^\star}(\theta_2)(y)$$
$$= \int_0^1 E_\Pi[\{2(y - U) - (\theta_1 + \theta_2)([1-t]U + ty, t)\}^\top (\theta_1 - \theta_2)([1-t]U + ty, t)]dt,$$

and so, by Hölder's inequality and the triangle inequality,

$$\|\ell_{P^\star}(\theta_1) - \ell_{P^\star}(\theta_2)\|_\infty \leq C\|\theta_1 - \theta_2\|_{L^\infty(\mathbb{R}^d\times[0,1];\|\cdot\|_2)},$$

where $C := 2(\sup_{\theta\in\underline{\Theta}} \|\theta\|_{L^\infty(\mathbb{R}^d\times[0,1];\|\cdot\|_2)} + \sup_{y\in[-1,1]^d} E_\Pi[\|y - U\|_2])$ and $\|f\|_\infty := \sup_{y\in[-1,1]^d} |f(y)|$. The constant $C$ is finite since functions in $\underline{\Theta}$ are uniformly bounded (C18), and so the above shows $\ell_{P^\star}(\cdot)$ is $C$-Lipschitz continuous on $\underline{\Theta}$ when its domain is equipped with $\|\cdot\|_{L^\infty(\mathbb{R}^d\times[0,1];\|\cdot\|_2)}$ and codomain is equipped with the supremum norm.

When $\ell_{P^\star}(\underline{\Theta})$ is $C$-Lipschitz, properties of covering numbers yield that $N(\epsilon, \ell_{P^\star}(\underline{\Theta}), \|\cdot\|_\infty) \leq N(\epsilon/C, \underline{\Theta}, \|\cdot\|_{L^\infty(\mathbb{R}^d \times [0,1];\|\cdot\|_2)})$. Combining this with the fact that $\sup_Q N(\epsilon, \ell_{P^\star}(\underline{\Theta}), L^2(Q)) \leq N(\epsilon, \ell_{P^\star}(\underline{\Theta}), \|\cdot\|_\infty)$, with the supremum taken over all finitely supported measures $Q$ on $[-1,1]^d$, and then subsequently plugging these bounds into (S1), we find that, for all $\delta > 0$,

$$J(\delta, \ell_{P^\star}(\underline{\Theta})) \leq \int_0^\delta \sqrt{1 + \log N(\epsilon/C, \underline{\Theta}, \|\cdot\|_{L^\infty(\mathbb{R}^d \times [0,1];\|\cdot\|_2)})} \, d\epsilon. \tag{S30}$$

Applying a change of variables to the right-hand side and using C19 shows that the right-hand side is finite, which establishes C6.

**Condition C7:** Since C5 holds, Lem. S16 shows this condition holds with $C_7 = C_5$ and $d_\Psi^2(\psi, \Psi_P) := \operatorname{ess\,sup}_{X \sim P_X} \chi^2(P_{\psi|X} \| P_{\psi_P|X})$.

## L    KULLBACK-LEIBLER GUARANTEE FOR DOUBLEGEN AUTOREGRESSIVE LANGUAGE MODELS

### L.1    STATEMENT OF GUARANTEE

We now present a Kullback-Leibler guarantee for DoubleGen autoregressive language models (Example 3) for an empirical risk minimizer $\theta_n$ over some class $\underline{\Theta}$, such as one based on a transformer architecture (Vaswani, 2017). This guarantee will rely on the following condition:

**C20)** *Non-trivial probability of any possible next token:* For all $j \in [d]$, there exists $\delta_j > 0$ such that, for all $\theta \in \underline{\Theta} \cup \{\theta_{P^\star}\}$,

$$P^\star\big\{\delta_j \leq \theta_{Y^\star(j)}(1, 1, \ldots, 1, Y^\star(1), Y^\star(2), \ldots, Y^\star(j-1))\big\} = 1.$$

Because C20 only involves token sequences that are generated under $P^\star$, it allows some tokens to have zero probability conditional on earlier tokens. However, for token sequences that occur with positive probability, no hypothesis $\theta$ can assign tokens vanishing probability given past tokens.

**Corollary S2** (Kullback-Leibler guarantee for DoubleGen autoregressive language models)**.** *If there exists an empirical risk minimizer $\theta_n \in \operatorname{argmin}_{\theta \in \underline{\Theta}} R_n(\theta)$ and C6, C8, and C20 hold, then the conditions of Prop. 1 and Thm. S1 hold for appropriately specified constants. Hence, letting $\delta_n$ be as defined in (S6), the following holds w.p. at least $1 - e^{-s}$:*

$$\mathrm{D}_{\mathrm{KL}}(P^\star \| \phi_{n\sharp}\Pi) \lesssim \inf_{\theta \in \underline{\Theta}} \mathcal{G}_{P^\star}(\theta) + \delta_n^2 + (s+1)/n + \max_{j \in [2]} \|\alpha_n^j - \alpha_P\|_{L^2(P_X)}^2 d_\Psi^2(\psi_n^j, \Psi_P),$$

*where $d_\Psi$ is as in Lem. S16 and '$\lesssim$' denotes inequality up to a multiplicative constant that does not depend on $n$ or $s$.*

*Proof of Cor. S2.* In Appx. L.2, we show that C3–C5 and C7 hold. Combining this with the assumed C6 and C8 implies the conditions of Thm. S1 hold. This, in turn, implies that C2 holds with $r$ equal to the right-hand side of (S7). Moreover, by Appx. A, C1 holds with $D = \mathrm{D}_{\mathrm{KL}}$, $b = C_1 = 1$, and $\epsilon = 0$. Hence, the conditions of Prop. 1 hold, and applying that proposition gives the result. $\square$

### L.2    VERIFYING THE CONDITIONS OF THM. S1

Under the conditions of Cor. S2, we show that DoubleGen autoregressive language models (Example 3) satisfy the conditions of Thm. S1. This corollary directly assumes C6 and C8, and so we only need to establish the remaining conditions.

**Condition C3:** This follows from Thm. 2.6.3 in Cover (1999).

**Condition C4:** This follows from the definition of the cross-entropy loss and C20.

**Condition C5:** Let $\bar{\theta}(y) = \prod_{j=1}^d \theta_{y(j)}(1, \ldots, 1, y(1), \ldots, y(j-1))$, and define $\bar{\theta}_{P^\star}$ similarly. Observe that $\bar{\theta}_{P^\star}$ is the probability mass function of $P^\star$. Let $P_\theta^\star$ be the distribution of $[k]^d$ with

probability mass function $\bar{\theta}$. By C20, $\bar{\theta}(y) > 0$ if $y \in \mathcal{Y}_{P^\star} := \{y : \bar{\theta}_{P^\star}(y) > 0\}$. Combining this with the fact that $\log^2 b \leq (b-1)^2 (b^{-1} \vee 1)^2$ for all $b > 0$ yields

$$\|\ell_{P^\star}(\theta)\|_{L^2(P^\star)}^2 = 4 \sum_{y \in \mathcal{Y}_{P^\star}} \log^2 \left[\frac{\bar{\theta}^{1/2}(y)}{\bar{\theta}_{P^\star}^{1/2}(y)}\right] \bar{\theta}_{P^\star}(y) \leq 4 \sum_{y \in \mathcal{Y}_{P^\star}} \left[\frac{\bar{\theta}_{P^\star}^{1/2}(y)}{\bar{\theta}^{1/2}(y)} \vee 1\right]^2 \left[\bar{\theta}^{1/2}(y) - \bar{\theta}_{P^\star}^{1/2}(y)\right]^2 .$$

By C20, the squared maximum on the right is upper bounded by $\prod_{j=1}^d \delta_j^{-1}$, and so the right-hand side is upper bounded by $C_5 := 4 \prod_{j=1}^d \delta_j^{-1}$ times $\sum_{y \in \mathcal{Y}_{P^\star}} [\bar{\theta}^{1/2}(y) - \bar{\theta}_{P^\star}^{1/2}(y)]^2$, the squared Hellinger distance between $P^\star$ and $P_\theta^\star$. This squared distance is upper bounded by $\mathrm{D}_{\mathrm{KL}}(P^\star \parallel P_\theta^\star)$ (Tsybakov, 2009, Lemma 2.4), which in turn is equal to the generalization error $\mathcal{G}_{P^\star}(\theta)$. Hence, C5 holds with constant $C_5$.

While we have shown C5 holds, the corresponding constant grows exponentially with $d$. Given this, alternative bounds that do not rely on C5 may be preferred in this scenario. Section 3.2 of Foster & Syrgkanis (2023) provides one approach for deriving such bounds. While those bounds tend to have a slower rate in $n$ than the one in Thm. S1, they may be much better behaved in terms of $d$. In future work, it would be interesting to apply such results to derive generalization bounds for DoubleGen autoregressive language models.

**Condition C7:** Since C5 holds, Lem. S16 shows this condition holds with $C_7 = C_5$ and $d_\Psi$ as specified in that lemma.

## M  FURTHER DETAILS ON NUMERICAL EXPERIMENTS

### M.1  GENERATING COUNTERFACTUAL SMILING FACES

#### M.1.1  EXPERIMENTAL SETUP

**Code availability.**  The code will be placed on GitHub for the camera-ready version of this manuscript. Please see the attached zip files for this blinded peer-review version.

**Preprocessing.**  Training and test images were cropped to remove artifacts and resized to $64 \times 64$ pixels.

**Features.**  Each image was accompanied by 39 binary features indicating attributes such as hair color, age, and presence of accessories like hats and eyeglasses. We removed 8 features judged to plausibly be caused by whether someone smiles (raised eyebrows, open mouth, etc.), leaving 31 features in $X$. The names of these 31 features were: `5_o_Clock_Shadow`, `Bald`, `Bangs`, `Big_Lips`, `Big_Nose`, `Black_Hair`, `Blond_Hair`, `Brown_Hair`, `Bushy_Eyebrows`, `Chubby`, `Double_Chin`, `Eyeglasses`, `Goatee`, `Gray_Hair`, `Heavy_Makeup`, `Male`, `Mustache`, `Narrow_Eyes`, `No_Beard`, `Pale_Skin`, `Pointy_Nose`, `Receding_Hairline`, `Sideburns`, `Straight_Hair`, `Wavy_Hair`, `Wearing_Earrings`, `Wearing_Hat`, `Wearing_Lipstick`, `Wearing_Necklace`, `Wearing_Necktie`, and `Young`. The names of the 8 features excluded from $X$ because they could plausibly be caused by whether someone smiles are: `Arched_Eyebrows`, `Attractive`, `Bags_Under_Eyes`, `Blurry`, `High_Cheekbones`, `Mouth_Slightly_Open`, `Oval_Face`, `Rosy_Cheeks`.

**Nuisance estimation.**  For nuisance estimation, the inverse propensity was estimated using lightgbm (Ke et al., 2017) with the Riesz regression loss (Chernozhukov et al., 2021), with hyperparameters tuned via Optuna (Akiba et al., 2019). The outcome model was estimated using $k$-nearest neighbors, with $k = 200$ and the Euclidean distance used to compare feature vectors. A draw from this outcome model was obtained by sampling from $\mathrm{Unif}\{1, 2, \ldots, 200\}$ and then returning the corresponding neighboring image.

Sensitivity to nuisance misspecification was assessed by fitting each nuisance twice. In the first, well-specified, scenario, the nuisances were fit using all available data. In the second, misspecified, scenario, the outcome model was trained only using dark-haired instances and the inverse propensity weights for lighter-haired instances was scaled down by a factor of 4. By misspecifying the nuisances in this way, MSGM and plug-in estimation overrepresent dark-haired individuals when their relevant nuisance is misspecified (Tab. S2) .

Table S2: Assessment of the outcome and propensity nuisance estimators in well-specified and misspecified settings, via cross-fitted estimates of the probability (in percent) of having each attribute.[*] In both cases, the misspecified models underrepresent the Blonde attribute relative to the test set.

|  |  | Lipstick | Makeup | Female | Earrings | No Beard | Blonde |
|---|---|---|---|---|---|---|---|
|  | **Test set** | **48%** | **40%** | **59%** | **20%** | **84%** | **14%** |
| Outcome | Well-spec. | 48% | 42% | 58% | 17% | 86% | 14% |
|  | Misspec. | 49% | 44% | 56% | 18% | 86% | 0% |
| Propensity | Well-spec. | 47% | 38% | 58% | 19% | 83% | 15% |
|  | Misspec. | 44% | 36% | 54% | 19% | 80% | 9% |

[*] To assess the propensity estimator, an IPW cross-fitted estimator is used for each attribute. To assess the outcome model, a cross-fitted plug-in estimator is used for each attribute.

**Diffusion model architecture and training.** The diffusion models were trained using Diffusers and Pytorch (von Platen et al., 2022; Paszke et al., 2019), with code adapted from Hugging Face (2025) with the help of a Claude coding assistant. A U-Net score network was used (Ronneberger et al., 2015), with block output channels of (128, 384, 768), 4 layers per block, and attention head dimension 16. Training images were segmented to isolate faces (Kirillov et al., 2023; Lugaresi et al., 2019), and the training loss upweighted face pixels by a factor of 8. AdamW was used to update network weights over 500 epochs, with minibatches of size 192 and an initial $10^{-4}$ learning rate decaying exponentially with a half-life of 250 epochs (Loshchilov & Hutter, 2017). To improve model stability, we kept an exponential moving average of the model weights during training (decay rate 0.999) and used these averaged weights to generate images at inference time (Polyak & Juditsky, 1992).

**Performance metrics.** Performance was evaluated using several metrics. Fréchet and kernel inception distances were computed (Heusel et al., 2017; Bińkowski et al., 2018), based on an embedding tailored for human faces (ArcFace) (Deng et al., 2019) and the classical Inception-v3 embedding (Szegedy et al., 2016). Precision and recall were also computed (Kynkäänniemi et al., 2019), using these same embeddings. All of the aforementioned metrics were evaluated using torch-fidelity with its default hyperpameter values (Obukhov et al., 2020).

The above metrics were evaluated by comparing synthetically generated images to those from the test set. This test set consists of 39,829 draws from $P$, rather than from the target counterfactual distribution $P^\star$. To account for this, those data were reweighted in a doubly robust fashion. Concretely, test data were used to estimate the propensity score and outcome model with the same lightgbm and $k$-nearest neighbors nuisance estimation strategies described earlier. This yielded doubly robust weights, such that the weighted empirical distribution of the test data approximates $P^\star$. Observations were then resampled with replacement according to these weights, yielding 10,000 test images that should be approximately distributed according to $P^\star$.

In addition to the above metrics, we assessed the distribution of the attributes listed in Tab. 1 among synthetically generated faces by training a model to predict these attributes from images of faces. Because this predictor is used to evaluate diffusion models trained on the training set, we trained it on the test set, avoiding data leakage in subsequent diffusion model evaluation. To train the model, we added a new head to ResNet-50 and finetuned all weights over 20 epochs using AdamW and a multi-label classification cross-entropy loss (He et al., 2016; Loshchilov & Hutter, 2017). The model attained good predictive performance on the data it did not see during training, attaining the highest area under the curve of 0.995 for Female, the lowest of 0.90 for Earrings, and 0.96-0.98 for the other attributes. It was also well calibrated: the mean predicted probability of having each attribute differed by at most 0.02 from the actual proportion of instances with that attribute. To mitigate domain shift in image sharpness between real and synthetic faces (synthetic Laplacian variances were roughly 30% lower on average), we applied a Pillow's default sharpening filter to synthetic images before supplying them to the prediction model (Clark et al., 2025). For each attribute we report the mean predicted percentage across synthetic images (= mean predicted probability $\times 100\%$).

Table S3: Diffusion model performance as in Tab. 3, but using Inception-v3 rather than ArcFace embeddings.

|  |  | FID ↓ | KID ↓ | Prec. ↑ | Rec. ↑ |
|---|---|---|---|---|---|
|  | Naïve | 1.00 | 1.00 | 0.66 | 0.49 |
| *Both right* | Plug-in | 0.90 | 0.78 | 0.64 | 0.49 |
|  | MSGM | 0.88 | 0.76 | 0.64 | 0.49 |
|  | DoubleGen | **0.88** | 0.79 | **0.65** | **0.50** |
| *Outcome wrong* | Plug-in | 1.76 | 1.83 | 0.67 | 0.25 |
|  | DoubleGen | **0.89** | **0.81** | 0.63 | **0.49** |
| *Propensity wrong* | MSGM | 0.92 | 0.76 | 0.64 | 0.47 |
|  | DoubleGen | 0.99 | 0.92 | **0.64** | 0.45 |
| *Both wrong* | DoubleGen | 1.28 | 1.27 | 0.66 | 0.40 |

*↓/↑ denote whether smaller/larger values are preferred.*
*Fréchet/Kernel inception distances (FID/KID) rescaled so Naïve takes value 1.00.*
Values where DoubleGen is better than Naïve are marked in blue, and—within a given nuisance misspecification category—those where it **performs at least as well as the best alternative method** are marked in bold.

Table S4: Predicted percentages of synthetic counterfactual smiling faces possessing the attributes from Tab. 1. Also displayed are the **target attribute distribution**, calculated as the percentages of all faces (both smiling and nonsmiling) in the test set with these attributes, and naïve diffusion model attribute distribution, calculated as the percentages of synthetic faces generated by a traditional diffusion model trained on only smiling faces.
*Caution: Test-set attributes are human-labeled. Synthetic-image attributes are supplied by a classifier trained on real data and may be biased by human-imperceptible artifacts in the diffusion model output; treat them as approximate.*

|  |  | Lipstick | Makeup | Female | Earrings | No Beard | Blonde |
|---|---|---|---|---|---|---|---|
|  | **Test set** | **48%** | **40%** | **59%** | **20%** | **84%** | **14%** |
|  | Naïve | 57% | 41% | 69% | 26% | 86% | 15% |
| *Both right* | Plug-in | 49% | 35% | 61% | 22% | 81% | 12% |
|  | MSGM | 48% | 34% | 61% | 21% | 82% | 13% |
|  | DoubleGen | 49% | 35% | 62% | 22% | 83% | 13% |
| *Outcome wrong* | Plug-in | 47% | 34% | 54% | 23% | 75% | 0% |
|  | DoubleGen | 50% | 36% | 62% | 23% | 82% | 13% |
| *Propensity wrong* | MSGM | 43% | 30% | 55% | 20% | 78% | 7% |
|  | DoubleGen | 47% | 34% | 60% | 20% | 78% | 11% |
| *Both wrong* | DoubleGen | 45% | 31% | 57% | 19% | 76% | 4% |

### M.1.2 RESULTS

Tab. S3 shows results for this diffusion model experiment, but using the Inception-v3 embedding rather than ArcFace embeddings. Fig. S2 displays a random sample of 200 faces generated by the naïve model and DoubleGen when both nuisances were well specified.

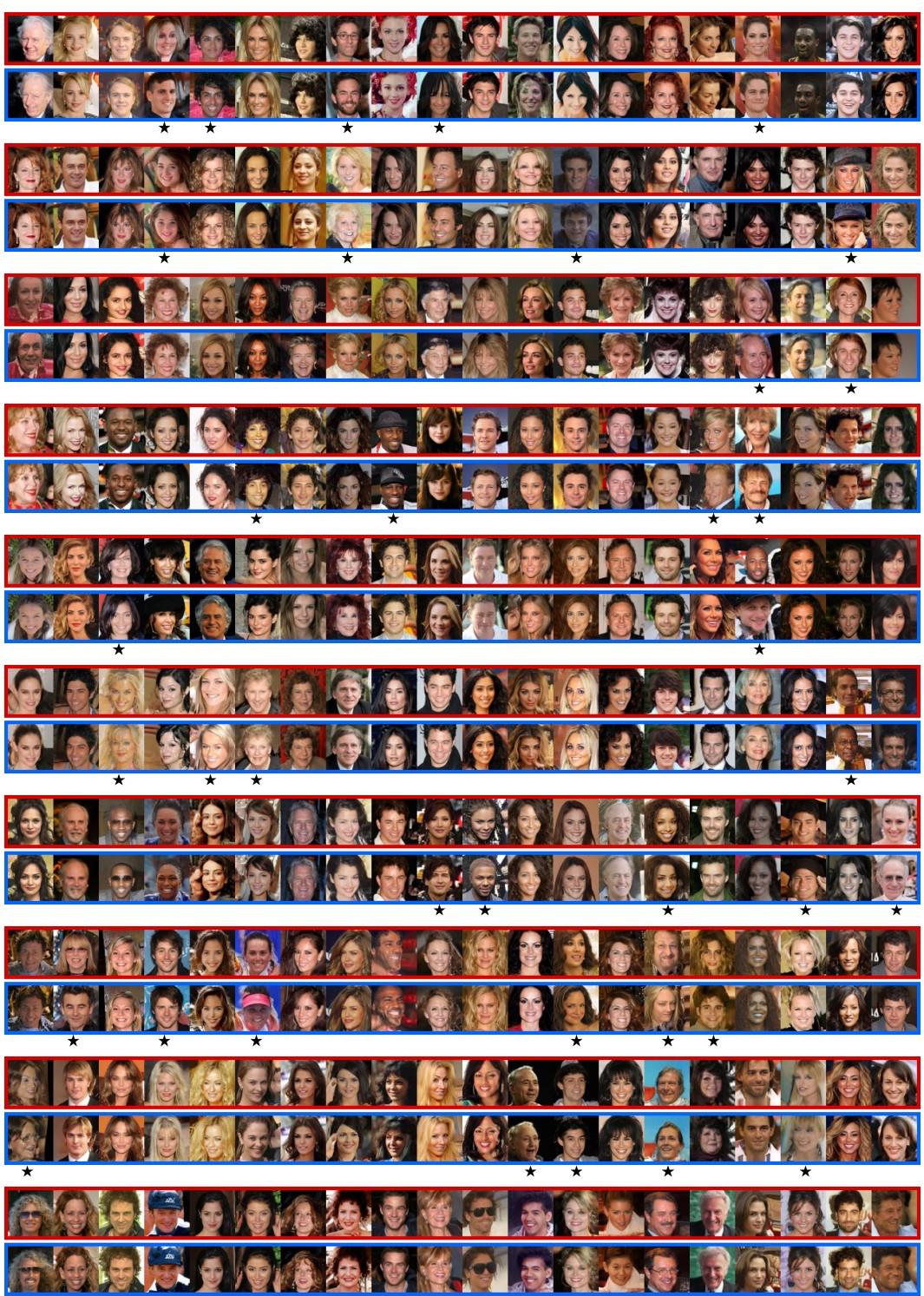

Figure S2: Batch of 200 random samples from a **traditional diffusion model** (left columns) and **DoubleGen diffusion model** (right columns), under the same setup as in Fig. 1. The ★'s mark the 20% of pairs whose ArcFace embeddings (Deng et al., 2019) have the largest cosine dissimilarity.

## M.2 GENERATING COUNTERFACTUAL PRODUCT REVIEWS

### M.2.1 EXPERIMENTAL SETUP

**Preprocessing.** Approximately 63.8 million instances with unknown product categories were removed from the dataset. Reviews were preprocessed by first prepending the rating (e.g., '5 stars: ' or '1 star: '), then tokenizing with the Llama 3 tokenizer, and finally truncating to at most 192 tokens.

**Features.** A total of 38 total features were considered. Of these, 33 were one-hot encoded product categories extracted from the dataset's metadata (books, electronics, etc.) The other 5 were the number of product images, number of product videos, and the length (in characters) of the product title, description, and details.

**Creating the semi-synthetic dataset.** To make the synthetic intervention depend on the features in a realistic way, we used 10% of the data to estimate a propensity model $\pi(a \,|\, x)$ based on an actual variable in the dataset, verified purchase status (see next paragraph for details). The remaining 90% was restricted to verified reviews only. From this subset, 30% was held out for evaluation, while 60% was modified for training. We drew a synthetic intervention indicator for each instance conditionally on $X$ using $\pi$, and replaced review text for instances not following the intervention with that of randomly drawn unverified reviews. Letting $Q$ denote the distribution of the original observations, $P$ the synthetic training data, and $A = a^\star$ denote following the synthetic intervention, a draw from $P$ can be sampled by drawing $X \sim Q_{X \,|\, A=a^\star}$, $A \,|\, X$ from $\pi$, $Y$ given $A = a^\star$ and $X$ from $Q_{Y \,|\, A=a^\star, X}$, and $Y$ given $A \neq a^\star$ and $X$ from $Q_{Y \,|\, A \neq a^\star}$. Crucially, the G-computed counterfactual distribution under $P$, $P^\star$, equals $Q_{Y \,|\, A=a^\star}$, which lets us evaluate the ground truth using our test set.

**Synthetic propensity $\pi$.** The propensity $\pi$ for the synthetic intervention $a^\star$ was designed to behave similarly to the propensity to have a purchase be verified, but with two key modifications. First, to make training computationally feasible on an academic budget, we reduced the proportion of instances receiving the synthetic intervention from approximately 90% to 3.5%. This reduction also demonstrates our method's performance in settings where receiving the target intervention is rare. Second, to ensure sufficient confounding for distinguishing between causal and non-causal methods in this illustrative experiment, we amplified how strongly the propensity depended on baseline features. Specifically, we used 10% of the data to obtain an estimate $\widehat{\pi}$ via linear-logistic regression, then defined the synthetic propensity on the remaining data as $\mathrm{expit}[-8.5 + 2\log it\,\widehat{\pi}(x)]$. This transformation maintains the relative ordering of propensities while increasing variance and shifting the overall values they take downward.

Large absolute coefficients in the logistic regression fit $\widehat{\pi}$ include dummies for the following categories: books ($-0.32$), Kindle Store ($-0.31$), movies and TV ($-0.15$), automotive ($0.15$), CDs and vinyl ($-0.14$), and clothing, shoes, and jewelry ($0.13$). Video count also had a coefficient of $0.07$ and title length had a coefficient of $0.03$.

**Nuisance estimation.** Nuisances were estimated similarly to as in the smiling faces experiment, so here we only highlight the differences. For the propensity, we used the SynapseML implementation of lightgbm, since this can be run on a dataset with hundreds of millions of observations (Hamilton et al., 2018). Because that implementation does not allow custom loss functions, we used a cross-entropy loss instead of a Riesz regression loss. We also used standard default tuning parameter values, rather than selecting them via Optuna, and truncated inverse propensities at 1000. For the outcome model, we stratified the search for the 200 nearest neighbors by product category, using the other 5 features for matching within each category.

As for the smiling faces experiment, sensitivity to nuisance misspecification was assessed by fitting each nuisance model twice. The well-specified nuisances used all baseline features. The poorly specified propensity omitted the product category features. In contrast, the poorly specified outcome model ignored the 5 non-category features. A draw from this model was obtained by equiprobability sampling of a training instance that received the intervention within the category defined by $x$.

**Training.** We used low-rank adaptation (LoRA) to finetune the 1 billion parameter Llama 3.2 base model (Hu et al., 2022; Dubey et al., 2024). The LoRA layers had rank 8, resulting in about 5.5M trainable parameters. AdamW was used to update the weights, with a learning rate decaying from $2 \times 10^{-4}$ according to a cosine decay over 1 epoch. Training was implemented using the

```
5 stars: My son loves to use the game and can play for hours. Thanks for a fantastic
app purchase!
```

```
5 stars: My son loves to use the game and can play for hours. Thanks for a fantastic
game purchase.
```

```
5 stars: Bought this to use as a basecoat to put on over makeup so that it doesn't
leave me with white patches under my eyes. Works wonderfully
```

```
5 stars: Bought this to use as a basecoat to clear stains from a glass countertop.
Worked fine, and was reasonably priced for doing the job
```

```
1 star: Purchased this adapter for using a Fire tablet with a cable tv box and it
doesn't work I can not get the picture to show up and it just says input doesn't exist
```

```
3 stars: Good for the price, not perfect in my opinion. Bought in October and one
stopped working in December. You get what you pay for.
```

```
1 star: This was a total bummer to start with I was very excited it got here 4 days
later and then when I started putting it together the legs don't line up so I have to
put in so many screws to basically take it apart but no longer use for anything
```

```
1 star: This was a total bungle to put together I just didn't understand the
instructions
```

```
5 stars: These are a must if you want to look great in a skirt. They are very durable.
Will save me months and months of having to go buy new ones.
```

```
5 stars: These are a must if you want to look great in your shorts. They are very
durable. Will save you money and time when it's time to order more.
```

Figure S3: Reviews generated by a **traditional language model** trained on only instances that follow the target intervention (top) and **DoubleGen language model** (bottom). Samples are coupled, with the random seed set to the same value for both models before generation. While the models often produce similar reviews, certain product categories are over- or underrepresented by the traditional language model—see Fig. S4 for an example with book reviews.

Transformers and PEFT libraries, with a PyTorch backend (Wolf et al., 2019; Mangrulkar et al., 2022; Paszke et al., 2019). A Claude coding assistant was used to help write, edit, and comment the code.

**Performance metrics.** Performance was evaluated using four metrics. Mean perplexity was computed using 20k randomly selected test samples, as computed in the HuggingFace evaluate library (Jelinek et al., 1977; Von Werra et al., 2022). MAUVE and frontier integrals were used to assess divergences between 10k synthetic generated samples and 20k randomly selected test samples, using the default settings from the software in (Pillutla et al., 2021). Finally, the 1-Wasserstein distance was computed between the distributions of the star ratings appearing at the beginning of the generated samples (e.g., a review that begins "3 stars: ..." is converted to a numeric rating of 3) and the star ratings of the roughly 150M test samples.

> 5 stars: This tool was great and really helped make my wedding cake. The metal is very sturdy and the wood is also of good quality. The handles are long enough to put more weight on as well.

> 5 stars: This book was great! I had to read the last half in a day. The first half was slow to get going and I did not know much about the subject, but it was interesting and eventually I understood the characters a little. The last half was fantastic! Will definitely read more of Anna Lee's books. I recommend it to all mystery readers!

> 5 stars: This is a very cute, simple but not unattractive case. I used the screen protector that came with it- although I had to replace my own screen protector. I also bought a new plastic screen protector when it came time for it to be replaced- both lasted for a few years. It's very slim, so I'm confident it will last a very long time to come, and the price was reasonable. I highly recommend.

> 5 stars: This is a very cute, simple but not childish, book. The photos are gorgeous and the illustrations are wonderfully colorful- bright and cheerful! I have ordered quite a few of this author's titles recently and am so very happy I purchased it.

> 3 stars: It's an OK quality mask. The design and the eye holes are nice. However, the straps on the back are not adjustable at all so it's hard to keep it on your face or to get the bottom part on straight.

> 3 stars: It's an OK book. The first and the last chapters are rather repetitive. The characters are interesting and likable.

> 5 stars: This is amazing!!!! It's durable, easy to use, I love it and it came with all the batteries

> 5 stars: This book was amazing. The author took the time get to know and truly connect with both the characters.

Figure S4: Coupled reviews from the same models as in Fig. S3, filtered to show only pairs where at least one model's review contains the word 'book' or 'books'. **DoubleGen** generates reviews with this word at roughly the same frequency as the test set (4.4% vs. 4.2%), while the **traditional model** severely underrepresents this content (0.24%).

## M.3   RESULTS

Fig. S3 displays reviews generated by DoubleGen and the naïve approach. Most of these reviews are similar to one another. This suggests there was relatively little confounding in this experiment. However, there are certain types of reviews that are underrepresented by the naïve approach. This includes those that contain the word 'book' or 'books' (Fig. S4). This is not surprising given that the propensity to receive the synthetic intervention is lower for reviews written for items in the Amazon's Books product category.

