# OpenReview forum: "DoubleGen: Debiased Generative Modeling of Counterfactuals"
_ICLR.cc/2026/Conference — Submitted to ICLR 2026_

### Official Review · Reviewer_RQds · 2025-10-14

**Soundness:** 2
**Presentation:** 3
**Contribution:** 3
**Rating:** 6
**Confidence:** 3

**Summary:**

The paper introduces a new counterfactual generation framework that ensures doubly-robustness. Theoretical properties of the robustness and optimality of the estimates were derived. Empirical experiments on the framework were presented and analyzed.

**Strengths:**

The paper is the first to propose a generative counterfactual framework that is doubly robust. Previous works typically only focus on at most one aspect.

The framework is completely general-purpose and can be used with any generative models, making it highly flexible to deploy in different practical tasks.

The theories derived in the paper, including finite-sample PAC bounds and minimax rates, are technically highly nontrivial. The proof techniques are valuable for existing statistical learning and semiparametric theory literature.

The theories provide a clear interpretation of the role of "doubly robustness" in the framework.

**Weaknesses:**

Framework novelty: the proposed framework (Algorithm 1 and Algorithm 2) seems to take the form of a common framework for doubly-robust estimation. The novelty of the framework is unclear.

Assumptions: the theories hinge on a series of assumptions (C3-C8), involving bounded losses, curvature constants, entropy integrals, and Lipschitz properties of transport maps. These assumptions seem to be stylized, and their practical verifiability remains unknown.

Baselines: the baselines used in the experiments are naive (plug-in, IPW), and no state-of-the-art baselines were used for comparison.

**Questions:**

Framework novelty: how is the algorithm novel compared to existing doubly robust frameworks?

Assumptions: could the authors justify the reasonability and verifiability of the assumptions that were used in the paper?

Baselines: I suggest that the authors should do an ablation study by comparing different types of generative models other than the diffusion model used in DoubleGen.

---

> ### Author Response · Authors · 2025-11-20
> **Thank you - reply 1/2**
>
> Thank you for your review and for highlighting the generality and theoretical contributions of our piece. We respond to your questions in what follows.
>
> > Framework novelty: how is the algorithm novel compared to existing doubly robust frameworks?
>
> The novelty of our proposed method stems in identifying how to incorporate ideas from causal inference and semiparametric statistics into generative modeling. While some previous works have done this (see lit review in Section 2), they have all either solved a harder problem than necessary (generating features and outcomes rather than just outcomes) and/or apply only to specific generative modeling frameworks. They are also not doubly robust and lack accompanying generalization error or divergence guarantees.
>
> In the present work, we identified the key components of popular generative modeling (risk minimization + subsequent transformation of the risk minimizer for sampling), and then systematically studied how to derive a causal estimation strategy. We identified the risk minimization portion as the key component where prior methods developed by the causal inference community could be employed. Specifically, we use an augmented inverse probability weighted estimator of the risk (Robins et al., 1994), which ultimately provided our generative modeling proposal with double robustness. We emphasize that this framework has, to our knowledge, never previously been used in generative modeling contexts.
>
> Beyond proposing a method, we provide a rigorous theoretical study of it. We show how existing results from the generative modeling literature can be ported over to reduce establishing a divergence guarantee to establishing a generalization bound (Appendix A). We then provide a novel generalization bound in Theorem 1, the derivation of which is highly nontrivial, requiring handling the fact that our risk minimization step relies on cross-fitted nuisance estimators. This theorem is written in such a way that, to the extent possible, the conditions overlap with those for standard generative modeling frameworks (Prop. S1), even though the proof is much more involved. Having a theorem that shares conditions with those for standard generative models is helpful because it then makes it possible to port over existing results from the literature to establish some of the conditions of our theorem. We do this in Appendix J when we study DoubleGen diffusion.
>
> So to conclude, our novelty is as follows:
> We are the first to provide a doubly robust counterfactual generative modeling framework.
> We are the first to provide high-probability divergence or generalization error guarantees for a counterfactual generative modeling framework, doubly robust or otherwise.
> We show our general framework can be used to provide guarantees for three popular generative modeling frameworks, including to establish the near minimax rate optimality of DoubleGen diffusion.
> We provide supporting numerical experiments suggesting that our theoretical guarantees are informative at realistic sample sizes.

---

> > ### Author Response · Authors · 2025-11-20
> > **Thank you - reply 2/2**
> >
> > > Assumptions: could the authors justify the reasonability and verifiability of the assumptions that were used in the paper?
> >
> > Thank you for raising this point. In the revision, we have added a figure (Fig. S1) that broadly summarizes our results and the conditions they rely on. Moving from left to right in the figure, results go from high-level guarantees that can be applied broadly to more specialized results that verify the conditions of those to their left in specific settings. Importantly, we have provided sufficient conditions for C3-C8 in all three of our examples (Appxs. J, K, and L). Moreover, in the context of diffusion models, we provide a specific neural network instantiation of the hypothesis class $\\Theta$ and show that using it yields a near minimax counterfactual distribution estimator.
> >
> > > Baselines: I suggest that the authors should do an ablation study by comparing different types of generative models other than the diffusion model used in DoubleGen.
> >
> >
> > We have experimentally evaluated DoubleGen on two distinctly different generative modeling problems: images and text. For images, we applied diffusion models and compared to IPW (a state-of-the-art method published in Wu, 2024) and a direct approach, a plug-in estimator. These comparators use the same generative modeling framework (diffusion) and nuisance estimators as DoubleGen. By doing so, we are able to systematically evaluate its robustness properties and performance within our computational budget (see our reply to Reviewer bHio for a discussion of the computational costs of evaluating DoubleGen and competitors in a variety of nuisance misspecification scenarios).
> >
> > In future work, it would be interesting to compare to methods that use different nuisances, but then for comparison purposes these same nuisances should be supplied to DoubleGen. We could also compare to an approach built on a different generative modeling framework, but then DoubleGen should also be trained using that framework. While we agree that such a systematic benchmarking exercise is worth doing, it is beyond the scope of the current work.

---

> > > ### Comment · Reviewer_RQds · 2025-11-20
> > >
> > > I appreciate the author's responses. I have one further comment.
> > >
> > > The author's statement "...IPW (a state-of-the-art method published in Wu, 2024..." is misleading. IPW was proposed a long time ago and not by Wu et al. (2024). My understanding is that this is an ablation model created from DoubleGen by retaining the IPW component, and it happens to coincide with Wu et al. (2024). So I think the author should clarify this in the paper, where the simplest way is to change "IPW" to the name of the model proposed by Wu et al. (2024).
> > >
> > > Overall, I believe the paper is good work. I will consider raising my score in favor of its publication.

---

> ### Author Response · Authors · 2025-11-20
>
> We agree that, similarly to AIPW (see cited Robins, 1994), IPW as a method for estimating counterfactual means/distributions was introduced a long time ago. To clarify this, we have uploaded a new version that cites Robins (1986) when discussing IPW methods on line 107. However, to our knowledge its use for **generative modeling of counterfactual distributions** is more recent - and indeed, appears to have first been used in Wu et al. (2024). Earlier counterfactual generative modeling approaches approaches (e.g., Kocaoglu et al., 2017) did not use IPW.
>
> Regarding naming, we agree it would be clearer to update to the name used in Wu et al. (2024). To this end, we have updated "IPW" in the new version to "MSGM" (marginal structural generative model).

---

### Official Review · Reviewer_AaJB · 2025-10-26

**Soundness:** 4
**Presentation:** 3
**Contribution:** 3
**Rating:** 8
**Confidence:** 2

**Summary:**

The paper proposes a counterfactual generation framework/loss called DoubleGen with doubly robustness properties. That is, the framework can properly estimate the counterfactual risk if either the inverse propensity or the conditional transport map is estimated consistently. The authors then give some examples of adapting DoubleGen to different types of generative models, such as diffusion and autoregressive models. They then provide a theoretical analysis that shows that the resulting generalization error between the estimated and true risk is small with high probability. Finally, the authors conduct numerical experiments on two datasets, demonstrating that DoubleGen has more robust performance under misspecification settings.

**Strengths:**

•	The proposed framework can be generalized to a number of generative models with different architectures, highlighting its flexibility and potential applicability across diverse domains.

•	The authors provide rigorous proofs to validate the generalization and minimax lower bounds for the counterfactual generation problem.

•	The conditions/assumptions used in Section 5.2 mostly make sense and allow for nonparametric nuisance estimation while still providing finite-sample guarantees.

•	The celebA example serves as an intuitive and straightforward illustration for confounding bias.

**Weaknesses:**

•	The notation/symbols in the introduction section needs more clarification to improve readability. For instance, if $P$ represents the factual distribution, then it would be better to use $P^*$ rather than $\mathbb{P}$ to represent the counterfactual distribution.

•	Similarly, the symbols should be better clarified in Sections 4 and 5. For example, it should be noted that $Law(Y)$ means “the distribution of $Y$”. Also, it is recommended to use consistent fonts for variables (e.g., $X$, $Y$, $A$), distributions (e.g., $\mathbb{P}$), and spaces (e.g., $\mathcal{Y}$, $\mathcal{U}$).

**Questions:**

Is it possible to extend the framework to situations where there exist unmeasured confounding and/or interference (i.e., the treatment on one unit can affect the outcome of another)?

---

> ### Author Response · Authors · 2025-11-20
> **Thank you**
>
> Thank you for these recommendations. In the revision, we have followed your suggestion and updated the notation for the counterfactual distribution from $\\mathbb{P}$ to $P^\\star$. To avoid double super/subscripting, we have in turn modified the notation $\\phi_{P^\\star}$ to $\\phi^\\star$. We agree that using stars throughout for counterfactual distributions is clearer and we hope that this improves readability.
>
> We also appreciate your other suggestions regarding symbols for fonts. We generally have tried to follow them, using \\mathcal for spaces and capital letters for random variables. After experimenting with using \\mathbb notation for all distributions, we found that extensive use of blackboard-bold made many of the equations visually heavy. Therefore, we would prefer to use standard font for distributions. If you feel strongly about using \\mathbb for distributions, we are happy to revisit this.
>
> Regarding your question: we agree that it would be interesting to extend this framework to situations where there is unmeasured confounding and/or interference. Our hope is that such extensions would borrow from the existing causal inference literature on these topics. For example, instrumental variable methods might provide a way to make progress when there are unmeasured confounders. While we agree this is an interesting area for future work, it is beyond the scope of this piece.

---

### Official Review · Reviewer_bHio · 2025-10-31

**Soundness:** 2
**Presentation:** 2
**Contribution:** 3
**Rating:** 4
**Confidence:** 1

**Summary:**

The paper address the problem of training generative models of counterfactuals using observational data, in which the intervention is not assigned randomly, but rather depends on some underlying attributes of the population. The authors define a general class of generative models (OracleGen) and then propose how the loss function for these models can be adjusted to allow for doubly robust generation using samples from a confounded distribution. The authors provide some results about the theoretical guarantees of their proposed approach, although verifying whether these are correct is beyond my expertise. The authors then showcase some empirical results showing that their approach works better than naive approaches and IPW.

The theoretical results presented in this paper falls outside of my area of expertise, hence in what follows I will focus my comments only on the proposed setup and experimental evaluation.

**Strengths:**

The most important contributions of this paper seem to lie in the theoretical guarantees it derives for the consider counterfactual generation setting. However, I do not feel qualified to evaluate whether these results are correct and/or novel.

**Weaknesses:**

- The experimental results in Table 3 are presented for a single dataset in each modality only. Further, the results seem to be run over a single seed only, making it difficult to establish any notion of statistical significance of the presented results.
- For section 6.1, from the description of the performance metrics in the appendix (l. 2089) it seems to me that the test set was obtained from the confounded dataset in a way very analogous to the way that the outcome model was constructed (using doubly robust generation, under the same models). If this is the case, one could expect some notion of "congeniality bias" in the results (there results are good because the model was fitted using the same principles and assumptions used to construct the test set). Have the authors considered other methods for constructing the test set?
- For the experiments in section 6.2, I am not sure to what extent PPL is a good evaluation metric in this context, as it seems to measure how "confident" the LLM was in its own prediction (unless an external LLM was used to evaluate the quality of the generations, which is not stated in the paper). That does not seem to account for whether the generated samples match the counterfactual distribution or not.
- It would be great if the descriptions of the evaluation metrics used in the paper were a bit more self-contained.

**Questions:**

Some things which were not clear to me when reading the paper include:
- What is the space of the interventions $A$? Are the authors only considering binary interventions $A \in \{0, 1\}$? In line 144, why do we consider $A=1$?
- In Algorithm 2, why is the empirical risk computed over two disjoint sets $\mathcal{Z}_n^1$ and $\mathcal{Z}_n^2$? In a standard doubly-robust estimators we usually want to estimate the propensity score function and the outcome model using disjoint subsets of the data, however it does not seem to me from the formulation of $R_n(\theta)$ which of $\mathcal{Z}_n^j$ is used for estimating the propensity and which is used for estimating the outcome model. Also, why do we need two outcome models $\psi_n^1$ and $\psi_n^2$?

---

> ### Author Response · Authors · 2025-11-20
> **Thank you - reply 1/2**
>
> Thank you for your review. We very much appreciate your feedback on our experiments, and respond to that in what follows.
>
> > The experimental results in Table 3 are presented for a single dataset in each modality only. Further, the results seem to be run over a single seed only, making it difficult to establish any notion of statistical significance of the presented results.
>
> Thank you for this comment. We emphasize that our experimental results are meant to support our substantial theoretical findings, which show that DoubleGen is doubly robust (Thm. 1) and satisfies a high-probability divergence guarantee (Prop. 1) under conditions. We further evaluate the plausibility of these conditions for three popular generative modeling frameworks: flow matching, diffusion models, and autoregressive language models (Appendices J, K, L). And finally, our theory shows that DoubleGen diffusion models satisfy a high-probability maximal risk guarantee similar to the one satisfied by traditional diffusion models.
>
> Our experimental results complement these theoretical findings, showing that they do indeed support DoubleGen’s strong performance for realistic data-generating processes and sample sizes. Evaluating these properties involved fitting 9 image diffusion models (DoubleGen and comparators across different nuisance misspecification settings) and finetuning 9 language models. Unfortunately, reproducing our full experimental suite with multiple datasets and many seeds is beyond the computational budget that is available to us for this conference submission (each diffusion-model replica alone takes ≈300 GPU-hours).  We have therefore chosen to keep the experiments focused, and to be completely transparent about their scope and their purpose. To this end, we have added the following to the Discussion section:
>
> > **Scope of empirical study.** Our experiments were designed to evaluate the applicability of our theoretical guarantees to realistic data-generating processes and sample sizes, not to provide an exhaustive benchmark. Given that training a single diffusion model in our setup required roughly 300 NVIDIA A6000 GPU hours, we leave systematically exploring performance on additional datasets and random seeds to future work.

---

> > ### Author Response · Authors · 2025-11-20
> > **Thank you - reply 2/2**
> >
> > > For section 6.1, from the description of the performance metrics in the appendix (l. 2089) it seems to me that the test set was obtained from the confounded dataset in a way … Have the authors considered other methods for constructing the test set?
> >
> > Thank you for raising this. In the revision, we have added a second, complementary assessment of the performance of the diffusion model that does not require estimating the nuisances used by DoubleGen. This new assessment evaluates the percentage of synthetic images with the attributes from Table 1 for DoubleGen and other approaches. To do this, we finetuned a ResNet-50 model to predict the 6 attributes in Table 1 based on an image (see Appx. M.1.1 for details). We then used this fitted model to predict the attributes on 12k synthetic images generated by each diffusion model (Table S4). These new results support that DoubleGen corrects the confounding of the naïve model and is doubly robust. Importantly, evaluating the attributes in the test set does not require estimating the same nuisances as DoubleGen. The only nuisance estimated for evaluation here is a separate attribute‑prediction model, and so there should be no concern of “congeniality bias” in this setting.
> >
> > We also emphasize that, in our other experiment (generating Amazon review text), the ground truth is known and so there is no need to estimate any nuisances during evaluation.
> >
> > > For the experiments in section 6.2, I am not sure to what extent PPL is a good evaluation metric in this context, as it seems to measure how "confident" the LLM was in its own prediction (unless an external LLM was used to evaluate the quality of the generations, which is not stated in the paper). That does not seem to account for whether the generated samples match the counterfactual distribution or not.
> >
> > To clarify, PPL compares the next-token probabilities predicted by the language model to out-of-sample ground truth text via a cross-entropy loss. Therefore, it is not measuring how confident an LLM is in its own predictions, but rather, how well its predictions line up with actual text. Since the data-generating process was semi-synthetic in this experiment, that ground truth text corresponded to holdout data drawn from the true counterfactual distribution.
> >
> > Given that PPL is a standard measure of language model performance, we feel it is important to include here. However, we also include several other metrics. These include other standard metrics (MAUVE, Frontier Integral) of how well synthetic text lines up with a ground-truth distribution. They also include a sentiment measure more specialized to this Amazon Reviews context, namely the Wasserstein distance between review distributions. Finally, in Fig. S4, we also assessed confounding by measuring the proportion of instances that included the word “book(s)”, which is interesting since the synthetic intervention was less likely to be given to book reviews than reviews for other products.
> >
> > > It would be great if the descriptions of the evaluation metrics used in the paper were a bit more self-contained.
> >
> > Thank you for raising this. In the revision, we have added clear references in the footnotes to Table 3 to show readers where they can find a self-contained presentation of the performance metrics used (Appendices M.1.1 and M.2.1). Unfortunately the tight space constraints preclude us from moving the information in these appendices to the main text.
> >
> >
> >
> > We now address your questions.
> >
> > **Q1**: Thank you for catching the typo on line 144. We have corrected it in the revision, replacing $Y|A=1,X$ with $Y|A=a^\\star,X$. In general, we are considering deterministic interventions that fix $A$ to some specific value, which we label $a^\\star$. So, while $A$ could take more than 2 values (e.g., “treatment 1, treatment 2, …, treatment 10”), our intervention fixes $A$ to one of them (e.g., $a^\\star=2$ so that the intervention is to assign treatment 2 to everyone).
> >
> > **Q2**: Algorithm 2 uses cross-fitting, which is an approach commonly used in double/debiased machine learning and semiparametric estimation. The general idea is to use separate data to estimate nuisances and evaluate empirical means (in this case, of loss functions). When studying a cross-fitted estimator, it is possible to condition on the data used to train the nuisances to ensure that the empirical means behave well (e.g., satisfy concentration inequalities). This is what we do in our proof of Theorem 1, for example. In the revision, we have made this explicit when we first introduce Alg. 2 and added a reference to a paper on the topic (Newey and Robins, 2018).

---

> > > ### Comment · Reviewer_bHio · 2025-11-26
> > >
> > > Thank you for all your clarifications and extended discussion, I have updated my score accordingly. While I understand that running experiments can be expensive, the performance differences in Table 3. are quite small in some settings (e.g. Language model section, where the cost of experiments should be much smaller than 300 GPU hours), and hence I think it would be great to establish some notion of statistical significance of the presented results, to validate to what extent the proposed method leads to actual performance improvements.

---

### Official Review · Reviewer_YFqG · 2025-11-01

**Soundness:** 3
**Presentation:** 4
**Contribution:** 4
**Rating:** 6
**Confidence:** 3

**Summary:**

This paper studies confounding and misspecification in generative models that aim to sample from counterfactual distributions. The authors propose DoubleGen, a doubly robust training framework that learns to generate counterfactual outcomes from observational data with confounding. The paper provides theoretical support and demonstrates the method on diffusion models and language models.

**Strengths:**

1.The paper proposes a doubly robust training framework for counterfactual generation called DoubleGen and claims that the generator remains consistent as long as either the propensity model or the outcome model is correct, without requiring both to be correct.

2.The paper provides theoretical support, including a finite sample generalization bound for the DoubleGen risk, a guarantee that links excess risk to closeness to the target interventional distribution, and a result that in the diffusion setting the method achieves a near minimax optimal rate.

3.The paper motivates the task using clear counterfactual scenarios such as what would each person look like if everyone smiled and frames this as causal counterfactual generation rather than ordinary attribute editing or style transfer.

**Weaknesses:**

1. In the CelebA experiment, the paper highlights specific confounding attributes such as lipstick, female, and no beard, and argues that naive generation entangles these attributes with smiling. However, the quantitative evaluation later focuses on global realism and diversity metrics such as FAD, KID, precision, and recall. The paper does not report whether the attribute distributions themselves were corrected toward the intended counterfactual target, so it is unclear if the method actually removes the identified confounding.


2. The theoretical results rely on a large set of conditions (C1-C20), which are distributed across the main paper and the appendix. The paper does not sufficiently discuss the practical meaning of these assumptions or how a practitioner could verify them. For example, the localized bound depends on condition C15, which assumes that at least one of the nuisance models is not estimated poorly.

**Questions:**

Q1. The theory shows bounds in terms of the error of the estimated propensity model and the distance between the estimated outcome model and the oracle outcome model, but in experiments you only present two regimes described as well specified and misspecified. Could the authors quantify nuisance quality more explicitly, for example by reporting calibration error, KL divergence, mean squared error, conditional log-likelihood, FID, or perplexity for each nuisance model?

Q2. The method assumes positivity, namely that every covariate profile has some non-zero chance to receive the intervention of interest. CelebA is relatively balanced for many attributes and the text experiment is semi-controlled, so violations of this assumption may be limited in the reported settings. Could the authors investigate what happens when the treatment probability is near zero for some subpopulation?

Q3. Algorithm 2 involves first fitting the nuisance models, then performing a cross-fit style data split, and then sampling minibatches together with latent noise in order to obtain unbiased gradient estimates of the proposed objective. This appears more expensive than training a single diffusion model or a single language model with a standard loss. Could the authors quantify the additional wall-clock time and GPU memory required by DoubleGen compared to naive finetuning and compared to an IPW-only baseline? Could the authors clarify whether the dominant cost comes from training the nuisance models or from optimizing the DoubleGen objective itself? For the language modeling setup with LoRA finetuning, could the authors report the approximate GPU-hours?

Q4. Table 1 highlights explicit confounding attributes such as lipstick, makeup, and female. However, Table 3 reports global distribution-quality metrics such as FAD and KID rather than directly measuring whether those specific confounding attributes were corrected in the generated counterfactual samples. Could the authors provide a more direct evaluation by reporting, for Naive, IPW, Plug-in, and DoubleGen, the marginal frequencies of the attributes listed in Table 1 within the generated samples, and comparing those frequencies to the target counterfactual distribution estimated via reweighting on the test set?

---

> ### Author Response · Authors · 2025-11-20
> **Thank you - reply 1/2**
>
> Thank you for your review. We are glad that you found the proposed DoubleGen framework for counterfactual generation, the accompanying theoretical guarantees, and the causal motivation and examples to be strengths of the paper. In particular, we appreciate your positive assessment of the contribution and presentation, and your comments on the doubly robust consistency and the near minimax-optimal rate in the diffusion setting.
>
> We now reply to your comments and suggestions.
>
> > In the CelebA experiment, the paper highlights specific confounding attributes such as lipstick, … The paper does not report whether the attribute distributions themselves were corrected toward the intended counterfactual target, so it is unclear if the method actually removes the identified confounding.
>
> In the revision, we have added an assessment of the attributes from Table 1 for DoubleGen and other approaches. We did this by finetuning a ResNet-50 model to predict the 6 attributes in Table 1 based on animage. Details on this training procedure are given in Appendix M.1.1. We then used this fitted model to predict the attributes on 12k synthetic images generated by each diffusion model - see the newly added Table S4. Overall, we find that DoubleGen generates faces that better match the attribute distributions in the test set than does the naïve method, and is more robust to nuisance misspecification than alternative methods. This assessment complements the more comprehensive ones given in Tables 1 and S2, which instead evaluate how more general distributional characteristics compare between the estimated and true counterfactual distributions.
>
> > The theoretical results rely on a large set of conditions…
>
> The generalization bound in Theorem 1 is our main result, and this only relies on 6 conditions (C3-C8). These conditions were carefully selected so that they would be similar to the ones typically used to establish generalization bounds in non-causal problems (Prop. S1 in Appendix F). Indeed, the existence of a risk minimizer, bounded loss, and curvature of the risk (C3-C5) are both required in causal and non-causal problems, and both results rely on an entropy condition. The large overlap between these conditions is important—and intentional!—because it makes it possible to borrow results from the non-causal literature when establishing guarantees for DoubleGen.
>
> Many of the other conditions that appear in this piece actually serve as sufficient conditions for verifying the ones needed for our main generalization bound. To make this clearer, we have added Fig. S1 to the revision, which broadly summarizes our results and the conditions they rely on. Moving from left to right in the figure, results go from high-level guarantees that can be applied broadly to more specialized results that verify the conditions of those to their left in specific settings. This figure covers 17 of the 20 conditions in the paper, and shows that many of them (C9-C13, C17-C20) are used specifically to discuss and verify our main conditions in the context of our specific examples.
>
> The only three conditions that do not appear in the figure (C14-C16) are used to provide the benchmark guarantee for OracleGen and an alternative, localized variant of our generalization bound that may be of interest to some specialized readers. However, we emphasize that our main generalization bound (Thm. 1) was sufficient to provide generalization bounds and divergence guarantees in all three of our examples. This includes our guarantee for DoubleGen diffusion models, which shows that using a specific neural network class for the hypothesis class $\\Theta$ yields a near minimax counterfactual distribution estimator.
>
> We also thank you for pointing out the lack of discussion of C15, which is used only for the localized variant of our main generalization bound. We have added this discussion below Theorem S2. We further added two references below C14 to provide readers with some context on settings where bounds based on the local entropy may yield sharper guarantees than those based on the global entropy.

---

> > ### Author Response · Authors · 2025-11-20
> > **Thank you - reply 2/2**
> >
> > Answers to questions:
> >
> > **Q1:** Thank you for raising this. In the revision, we have added Tab. S2, which quantifies the quality of the propensity and outcome model nuisance estimators. It does this by using these nuisances to derive cross-fitted estimates of the probability of having each attribute. If these probabilities differ heavily from those in the test set, then a nuisance model must be poorly specified. This table shows that (as expected because of the form of misspecification - see the *Nuisance estimation* section of M.1.1), under misspecification both nuisance models underrepresent Blonde individuals to a great extent and most other attributes from Tab. 1 to a lesser extent. We would be happy to add additional assessments of the nuisances used in our experiments in the camera-ready version of our paper.
> >
> > **Q2:** Thank you for raising this point. Positivity is a commonly required condition in causal inference (see, e.g., the cited Hernán and Robins reference). Without it, structural assumptions (e.g., linearity) would be needed to make it possible to extrapolate to values of the feature X where receipt of the intervention wouldn’t be possible. As is typically the case in settings that require positivity, when the overlap is poor enough, our estimator will become unstable. This can be seen by looking at our loss, which weights by the inverse of the propensity score: if the propensity to receive the intervention is very small for some individuals, these weights become highly variable.
> >
> > One of our experiments assessed the behavior of DoubleGen in a setting of poor overlap, though admittedly with a large sample size (n$\\approx$500M). Concretely, in our Amazon Reviews text experiment, some instances (e.g., certain book reviews) had a less than 1% chance of receiving the intervention conditional on their baseline covariates. As can be seen in Table 3, DoubleGen still performed well relative to alternative approaches in this setting.
> >
> > **Q3:** Thank you for this question. The relative cost of using a DoubleGen loss versus a typical diffusion model loss will depend on how the nuisances are estimated. In our experiments, we fit the propensity score via lightgbm and the outcome model via k-nearest neighbors. As such, the cost of estimating the nuisances was significantly less than the cost of training the diffusion model. For example, in our CelebA example, training the correctly specified nuisances took less than 1 hour total (using 48 CPU cores), compared to about 4 days to fit the diffusion model on 3 NVIDIA RTX A6000 GPUs. This 4-day duration was comparable to the length of time it took to train the traditional diffusion model using only Smiling faces over the same number of epochs.
> >
> > While the runtimes were comparable in our experiments, there are naturally settings where training DoubleGen will take longer. For example, if the outcome model had itself been fitted with a diffusion model conditioned on $X$ (rather than k-nearest neighbors), then we would expect the time to train the nuisances to be comparable to twice the time it would take to fit a traditional diffusion model, plus a little additional time to generate synthetic outcomes from these nuisance diffusion models while training DoubleGen and evaluate the loss $\\ell$ at these new synthetic outcomes. Nevertheless, broadly speaking, the runtime should be on the same order of magnitude as that of the traditional diffusion model in these settings, while avoiding the biases that this traditional approach suffers from.
> >
> > **Q4:** We have added Table S4 to the revision - see the earlier part of our reply for a discussion.

---

### Author Response · Authors · 2025-11-20
**Revised paper uploaded**

We thank the reviewers for their helpful comments and suggestions on our paper. We have uploaded a revised version of our paper. Edits are indicated by a grey bar in the right column (e.g., lines 144-146). Of course, we would remove these in the camera-ready version.

We will reply to each reviewer individually below.

---

### Author Response · Authors · 2025-12-01
**Summary of discussion for new area chair**

We thank the reviewers and ACs for the time that you have put into considering our paper. Since we understand that all papers have been assigned new ACs, we want to briefly summarize the discussion and rebuttals below.

**Reviewer AaJB (original score 8)** has consistently viewed the paper as a good contribution, highlighting the flexibility of the framework, the strength of the theoretical guarantees (including minimax bounds), and the clarity of the CelebA example; their main suggestions were for cleaner notation and symbol consistency, which we implemented in the revision.

**Reviewer YFqG (original score 6)** was positive about the core DoubleGen framework, its theory, and the causal framing, and raised concerns mainly about how we assessed whether the CelebA experiment mitigated confounding and about the interpretability of our many conditions; we addressed these by adding nuisance-quality diagnostics (Tab. S2), a direct attribute-level evaluation of confounding correction (Tab. S4), a clearer discussion of assumptions including C15 and positivity, and runtime comparisons, but they did not have a chance to respond before the rebuttal period was cut short.

**Reviewer RQds (original score 6, indicated they'd consider raising score)** emphasized the importance and technical depth of the theoretical results and the generality of the framework, but initially questioned the degree of novelty relative to existing doubly robust setups, the practicality/verifiability of assumptions C3–C8, and the strength of baselines. Our replies clarified how DoubleGen brings semiparametric doubly robust ideas into generative modeling in a way that is new (including the cross-fitted AIPW-style risk and generalization/divergence guarantees), explained how Fig. S1 and Appxs. J–L provide concrete sufficient conditions for the main assumptions in several standard generative frameworks, and updated the naming of the “MSGM” (IPW-based) baseline, after which they wrote that they believe the paper is good work and stated that they would consider raising their score in favor of publication.

**Reviewer bHio (original score 4, raised to 6)** focused primarily on experimental scope and evaluation, expressing concern about single datasets/seeds, potential “congeniality bias” in one diffusion evaluation, and the use and explanation of metrics. In response, we clarified that the experiments are designed as focused support for substantial theory under realistic compute limits, added an evaluation based only on a separately trained attribute-prediction model (Tab. S4), explained the language-model metrics (including how PPL is computed against ground-truth text), and pointed to more self-contained metric descriptions in the appendices. After this, they thanked us and noted that they had updated their score upward to reflect our clarifications.

---

### Meta-Review · Area_Chair_myxe · 2026-01-07

**Summary:**

Despite the high scores, in my opinion, this is a borderline paper because the reviewers are not confident, especially the reviewer with an overly supportive score. I have read the paper myself to form a more informed opinion and have some additional feedback for the authors. First, I will cover reviewer concerns, then discuss comments that I have based on my own reading.

Summary of Reviewers' Comments:

Reviewer YFqG is concerned with the computational/memory/cost overhead of the method and a lack of sufficient justification of the many assumptions distributed throughout the paper.

Reviewer bHio mainly had minor clarification questions about experiments and had a confidence of 1.

Reviewer AaJB was also not confident with a confidence of 2. They seem very supportive, but their review does not contain anything substantial, as they mainly asked clarification questions about symbols.

Reviewer RQds had concerns about novelty and also a lack of baselines.


AC Comments:

The authors' notion of "counterfactual" is a term borrowed from the potential outcomes framework, which actually is an interventional query in Pearl's SCM framework, not a true counterfactual with abduction steps. This distinction should be added to the paper. X is assumed to be sufficient for adjustment as we infer from equation (1).

Based on my own reading, I believe the authors could have done a better job connecting the objectives of interventional sampling and risk minimization. The concept of risk is kept too vague and generic at the beginning, which is confusing.

I find the paper in its current form not accessible to a significant majority of the causality community. The topic is super interesting. The contribution seems valid to a degree. But the presentation makes it very difficult to believe this paper will be as impactful as it could have been, had it been executed and written better.

It is not clearly explained how the double-robustness of some risk function translates to estimating the causal effect, which is not available, and one cannot directly define a risk function on. The examples don't help much: Example 1 requires access to P*, which is not available, similarly, for other examples. The authors simply show how the loss functions work in these settings.

Algorithm 1 is too brief. How is \tau trained/calculated?  What is its relation to R? Maybe the authors meant to write l(\tau(\theta), Y^*) instead of l(\theta, Y^*)?

Algorithm 2 is also confusing and not explained properly. \psi(.|x) is supposed to sample from Y. Then, below, why do you write \psi_n(u|x)?

Experiments, especially baselines, are very limited. The authors do make their point, but for ICLR, I would expect to see a more extensive baseline comparison since there are many recent works at the intersection of generative models and causal inference/interventional sampling using diffusion models.

I strongly urge the authors to showcase the approach on a toy SCM X->T->Y, X->Y, with numerical values. Showcase the double robustness. Tell us how to choose loss l and risk R for simple discrete variables. The presentation is suffering from being overly general.

**Reviewer Concerns:**

The authors' reply to Reviewer YFqG seems satisfactory to me.

The authors reply to Reviewer RQds did not address their main concerns in my opinion. The authors did not provide additional experiments.

**Reviewer Scores:**

I believe the scores would be similar after rebuttal. It is likely that the reviewer with a score of 8 and confidence 2 would have reduced their score based on the criticism raised by others.

---

### Decision · Program_Chairs · 2026-01-26

Reject